# Translocated *Legionella pneumophila* small RNAs mimic eukaryotic microRNAs targeting the host immune response

Tobias Sahr[1], Pedro Escoll [1], Christophe Rusniok[1], Sheryl Bui [2], Gérard Pehau-Arnaudet[3], Gregory Lavieu [2] & Carmen Buchrieser [1✉]

*Legionella pneumophila* is an intracellular bacterial pathogen that can cause a severe form of pneumonia in humans, a phenotype evolved through interactions with aquatic protozoa in the environment. Here, we show that *L. pneumophila* uses extracellular vesicles to translocate bacterial small RNAs (sRNAs) into host cells that act on host defence signalling pathways. The bacterial sRNA RsmY binds to the UTR of *ddx58* (RIG-I encoding gene) and *cRel*, while tRNA-Phe binds *ddx58* and *irak1* collectively reducing expression of RIG-I, IRAK1 and cRel, with subsequent downregulation of IFN-β. Thus, RsmY and tRNA-Phe are bacterial trans-kingdom regulatory RNAs downregulating selected sensor and regulator proteins of the host cell innate immune response. This miRNA-like regulation of the expression of key sensors and regulators of immunity is a feature of *L. pneumophila* host-pathogen communication and likely represents a general mechanism employed by bacteria that interact with eukaryotic hosts.

[1] Institut Pasteur, Université de Paris, Biologie des Bactéries Intracellulaires and CNRS UMR 6047, 75724 Paris, France. [2] Université de Paris, INSERM ERL U1316, UMR 7057/CNRS, Paris, France. [3] Unité de Technologie et Service BioImagerie Ultrastructurale and CNRS UMR 3528, Paris, France. ✉email: cbuch@pasteur.fr

ntracellular bacterial pathogens communicate with their hosts to allow pathogen entry, persistence, and replication within the host cells. This is mainly achieved by translocating bacterial protein effectors into the host cell through dedicated secretion systems[1]. The translocated proteins manipulate different host cell processes and signalling pathways. The Gram-negative bacterium *Legionella pneumophila*, a parasite of aquatic protozoa and a feared pathogen when reaching the human lungs, is one of these pathogens[2]. In the environment *L. pneumophila* replicates intracellularly in freshwater protozoa and during human disease, a severe pneumonia called Legionnaires' disease, it replicates in alveolar macrophages. The capacity to replicate in these immune cells is thought to stem from the ability to replicate within protozoan cells, as intracellular replication is similar in both hosts[3,4]. Genome sequence analysis has revealed a unique feature of this pathogen, namely the presence of many proteins similar to eukaryotic proteins suggesting that mimicry of eukaryotic functions allows *L. pneumophila* to subvert host pathways and to replicate in these cells[5,6]. Indeed, many studies thereafter showed that these proteins are secreted effectors of the Dot/Icm type IV secretion system[7] and are part of the over 330 translocated effector proteins of *L. pneumophila*[8]. Evolutionary analyses strongly suggested that they had been acquired by horizontal gene transfer from their protozoan hosts during co-evolution[9–13]. The translocation of these different bacterial proteins in the host cell is a vital part of a successful infection. They are subverting numerous host pathways such as ubiquitin signalling[14,15], the sphingolipid metabolism and autophagy[16–18], Rab proteins[19–25] and target different organelles such as the nucleus or mitochondria[26–29]. Given the high number of protein effectors that mimic eukaryotic functions that have been identified in the *Legionella* genomes[30] it is tempting to hypothesize that *L. pneumophila* might also mimic eukaryotic RNAs such as miRNAs to interfere with eukaryotic regulatory mechanism. If it is the case, this raises another question, how the bacterial RNA is transported into the eukaryotic cell? One possibility could be that it is mediated by extracellular vesicles (EVs) shed from bacteria.

EVs produced by bacteria, archaea, and eukaryotic cells, are increasingly recognized as important mediators of intercellular communication via transfer of a wide variety of molecular cargoes[31]. They have been implicated in many aspects of cell physiology such as stress response, intercellular competition, lateral gene transfer (via RNA or DNA), pathogenicity, and detoxification[32]. The pathophysiological roles of EVs are beginning to be recognized in diseases including cancer, infectious diseases, and neurodegenerative disorders, highlighting potential novel targets for therapeutic intervention[33]. Furthermore, recent studies highlight that the role of EVs in intercellular communication and in pathogenicity might have been largely undervalued so far[32]. Recently, EVs have gained great attention for their proposed roles in cell-to-cell communication, and as biomarkers for disease. Indeed, EVs have been implicated in many functions as mediators of near and long distance communication between eukaryotic cells[31,34] and as mediators of cell–cell and trans-kingdom communication in host-pathogen interactions[35].

*L. pneumophila* is well known to release EVs when grown in laboratory media[36]. These *L. pneumophila*-derived EVs (*Lp*-EVs) can be internalized by macrophages during cellular co-incubation and are able to fuse with eukaryotic membranes in vitro, as suggested by analyses of *Lp*-EVs with model-membranes[37]. Furthermore, when mouse macrophages were infected with such *Lp*-EVs the fusion of the phagosome with the lysosome was inhibited, like during infection of human macrophages with *L. pneumophila* bacteria[38]. Recently, it has been shown that *Lp*-EVs, when incubated with THP-1 macrophage-like cells are potent pro-inflammatory stimulators of macrophages[39]. At later time points, EVs seem to facilitate *L. pneumophila* replication by miR-146a-dependent IRAK-1 suppression[39]. It was suggested that EVs might thereby promote the spreading of *L. pneumophila* in the host. A first characterization of the proteome content of *Lp*-EVs identified 33 proteins enriched in the EVs, some of which possess proteolytic and lipolytic enzyme activities, which may contribute to the destruction of the alveolar lining during infection[40]. However, it is not known whether *Lp*-EVs contain also RNAs.

Here we show that *L. pneumophila* releases EVs in vitro and *in cellulo* during infection of the human U2OS cell line, human THP-1 monocytic cells, and human monocyte-derived macrophages (hMDMs). These *Lp*-EVs contain bacterial RNAs that are transported into the host cell where they downregulate IRAK1 and RIG-I, most likely by mimicking eukaryotic miRNAs.

## Results

**L. pneumophila extracellular vesicles are enriched in bacterial small RNAs.** Like many bacteria, *L. pneumophila* produces EVs during extracellular and intracellular growth[38,40,41]. To analyze the shape and structure of these *Lp*-EVs we purified them from broth culture and imaged them by uranyl acetate negative staining transmission electron microscopy and Cryo-TEM. On average, we purified $0.6–1.0 \times 10^8$ *Lp*-EVs from a 500 ml *L. pneumophila* culture grown to post-exponential phase. As shown in Fig. 1A, the *Lp*-EVs are mostly spherical structures harbouring single membrane bilayers ranging from around 20 to 200 nm in diameter. Moreover, the Cryo-TEM analyses revealed the presence of tube-shaped vesicles and double membrane bilayer vesicles (Fig. 1B). To analyze whether they contain RNA molecules, we used Vybrant™ DiD solution, a lipid dye to stain the membranes of the EVs, and Syto®RNA-Select to label the RNA content if present. Indeed, *Lp*-EVs stained with the DiD membrane stain (red) and the selective RNA dye (green) could be visualized (Fig. 1C).

To quantify the amount of purified *Lp*-EVs and estimate their size nanoparticle tracking analyses (NTA) was used after staining the putative *Lp*-EVs with Vybrant™ DiD to stain the membranes of the EVs. Although we used a size filtration column to remove excess of free dye, the presence of aggregated dye within our samples, or the presence of other large protein/lipid aggregates emanating from *L. pneumophila* could not be completely ruled out. Thus, we added a well-established sucrose floatation step to the isolation procedure[42,43]. We first analyzed the size distribution and the number of particles pre- and post-floatation, through light scattering mode revealing that the number of particles was moderately decreased after floatation likely due to the three additional ultracentrifugation steps required for the floatation procedure (Fig. 1D, E). Particles in both samples showed a median size of ~130 nm (Fig. 1E). In addition, we compared particles size and concentration when measured in fluorescence mode to analyze *Lp*-particles labelled with the red-lipophilic dye. Size distribution was similar (Fig. 1F, right panel), and ~85% of the particles were positive for the membrane dye, consistently with our FACS data (Supplementary Fig. 1) and previous studies. Possible differences in size between the analyses are probably due to the different accuracy of the various methods used here, a phenomenon described in detail by Bachurski and colleagues who have compared the different methods[44]. Together these results suggest that most of the *Lp*-derived nanoparticles considered in our study are indeed *Lp*-EVs.

However, we could not use the NTA to measure the percentage of *Lp*-EVs positive for the RNA dye as the set-up of the machine was not compatible with the fluorescent properties of the RNA-dye. Thus, we used conventional flow cytometry analyses, as the NTA results with respect to the size distribution and the

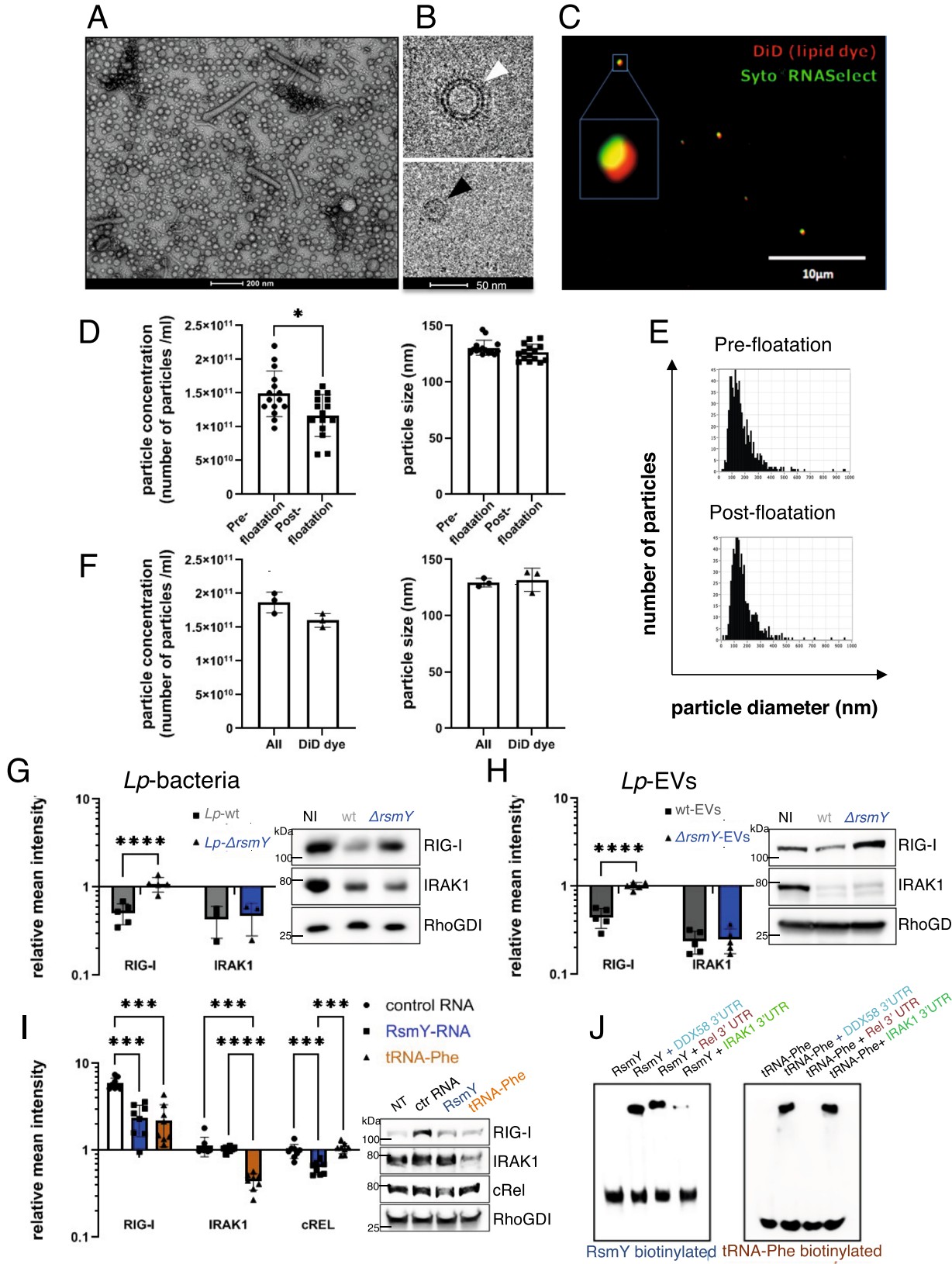

quantification were comparable to determine the percentage of *Lp*-EVs that contains RNA molecules (Supplementary Fig. 1A). For standardization, we used Megamix-Plus SSC (BioCytex) beads, a mix of fluorescent beads of varied diameters selected to cover a major part of the theoretical microparticle size range (0.1–0.5 μm), as a size-related parameter. Bead acquisition

allowed setting the cytometer to study microparticles within a constant size region and obtaining reproducible microparticle counts by flow cytometry (Supplementary Fig. 1B). Quantification showed that approximately 30% of the examined *Lp*-EVs contained detectable amounts of RNA (Supplementary Fig. 1C). We thus isolated the RNA from the *Lp*-EVs and analyzed the size

**Fig. 1** *Lp*-**EVs are a population of single and double membrane vesicles that contain small RNAs modulating RIG-I and IRAK protein levels. A** Negative staining transmission electron microscopy of *Lp*-EVs ($n = 1$). **B** Cryogenic transmission electron microscopy ($n = 1$), unilamellar EVs (black arrow), two lipid bilayer EVs (white arrow). **C** Fluorescence microscopy, DiD (red) and Syto-RNAselect (green) labelled *Lp*-EVs, ($n = 5$). **D** Absolute particle concentration (left) and median size (right) of purified *Lp*-EVs determined by ZetaView. Each dot represents a single measurement performed in triplicates and the (mean) SD of $n = 5$. Statistical analysis performed using unpaired t-test (two-tailed, $p < 0.05$ significant). Red Square, representative size distribution of particles for each sample. *$p = 0.0112$ (Wilcoxon). ns: $p > 0.05$ (Wilcoxon). Source data provided as Source data file. **E** Size distribution pre- and post-floatation was unchanged. **F** Absolute particle concentration (left) and median size (right) measured through light scatter (All) and fluorescence (DiD) mode. Each dot is one measurement and the (mean) SD of $n = 3$, ns: $p > 0.05$ (Wilcoxon). Source data provided as Source data file. **G** THP-1 cells infected for 8 h with *L. pneumophila* wt or *ΔrsmY*. Data are presented as (mean) SD of $n \geq 3$ independent, biological replicates ($p < 0.0001$) H) THP-1 incubated 3 h with wt or *ΔrsmY Lp*-EVs. Data are presented as (mean) SD of $n = 5$ independent biological replicates. ($p < 0.0001$). (**G** + **H**) Protein quantities of RIG-I and IRAK analyzed by western blot, intensities relative to the non-infected control (NI) and RhoGDI loading control. A two-way ANOVA for statistical analysis was performed. Right, representative western blots. Source data provided as Source data file. **I** Quantification of RIG-I, IRAK1 and cRel protein levels after transfection of THP-1 with RsmY and tRNA-Phe. Relative mean intensities normalized to non-transfected cells (NT) and RhoGDI loading control. Control RNA (ctrlRNA) average of random *L. pneumophila* DNA, anti-sense of RsmY or tRNA-Phe. Left, representative blot. Data are presented as (mean) *SD* of $n = 8$ independent biological replicates. For statistical analysis a two-way ANOVA was performed with $p < 0.05$ significant (*), $p < 0.01$ very significant (**), $p < 0.001$ extremely significant (***). Source data provided as Source data file. **J** Representative EMSA ($n = 3$) of in vitro transcribed RNA.

**Table 1 The twenty most enriched RNA molecules in the *Lp*-EVs as determined by RNAseq analyses.**

| Label/DNA region[a] | Annotation[a] | log2FC[b] | norm *Lp*-EV reads[c] | Similarity[d] |
|---|---|---|---|---|
| *lpp0294* (f): 328282-328335 | UTR/start | 13 | 13 184 | hsa-miR 329-3p |
| *lppnc0374* | sRNA (anti lpp1558) | 12 | 77 201 | - |
| *lppt29* | tRNA-Phe | 12 | 304 066 | hsa-miR 5001-3p + 4451 |
| *lppt25* | tRNA-Pro | 12 | 280 164 | - |
| *lppt08* | tRNA-Pro | 10 | 29 498 | hsa-miR 6813-5p + 152-5p |
| *lppt37* | tRNA-Met | 9 | 33 524 | - |
| *lpp0956* (f): 1063203-1063309 | UTR | 9 | 6 110 | hsa-miR 4775 |
| *lppnc0001* | sRNA RsmY | 9 | 349 673 | hsa-miR 144-3p |
| *lppt04* | tRNA-Tyr | 9 | 81 954 | hsa-miR 4687-3p + 6819-5p |
| *lppt39* | tRNA-Val | 8 | 104 797 | hsa-miR 323a-5p + 619-3p |
| *lppt38* | tRNA-Pro | 8 | 44 788 | hsa-miR 6816-3p |
| *lppt12* | tRNA-Leu | 8 | 26 718 | - |
| *lppnc0692* | sRNA (intergenic lpp2966/67) | 8 | 25 277 | hsa-miR 6807-5p + 4773 |
| *lppt34* | tRNA-Leu | 8 | 327 716 | - |
| *lppt43* | tRNA-Leu | 7 | 134 92 | hsa-miR 6742-3p |
| *lpp5011* (f): 2105717-2105648 | CDS | 7 | 17 070 | hsa-miR 3619-5p |
| *lpp5013* (f): 2578647-2578735 | CDS | 7 | 9 863 | hsa-miR 4637 + 6751-3p |
| *lppt06* | tRNA-Thr | 6 | 648 296 | hsa-miR 302b-5p + 6077 |
| *lppnc0047* | sRNA | 6 | 6 946 | hsa-miR 1290 + 329-5p |
| *lppt41* | tRNA-Met | 6 | 845 116 | hsa-miR 769-5p |

[a]Label-DNA region; sRNA label as defined in Sahr et al., 2017.
[b]Log2FC, Enrichment of RNA molecules in the Lp-EVs as compared to the bacterial culture defined as log 2 fold change (FC).
[c]norm Lp-EV reads, Number of reads detected for each sRNA within the Lp-EVs after normalization.
[d]Similarity, similarity found to a human sapiens micro RNAs (has-miR), 5p, 5′ end, 3p, 3′end, inverted, similarity with the inverted sequence; CDS coding sequence.

and RNA quality, revealing that mainly short nucleotide sequences of around 50–150 nucleotides (nts) were contained in the *Lp*-EVs (Supplementary Fig. 1D).

**L. pneumophila sRNAs contained in Lp-EVs show similarity to human microRNAs.** To characterize the RNA molecules present, and to identify those enriched in *Lp*-EVs we performed RNAseq analyses. Four independent RNAseq libraries were constructed from RNA isolated from purified *Lp*-EVs and as control from the bacterial pellets from which the *Lp*-EVs had been shed and deep sequenced using an Illumina platform. The sequences obtained from the *Lp*-EV RNAseq libraries were compared to those obtained from the RNA extracted from the bacterial pellets (Supplementary Fig. 1E). The *Lp*-EV-sRNA cargo was defined as RNAs for which after normalization at least 1000 reads were sequenced and that showed an enrichment of a log2FC > 5 as compared to the RNAs from the bacterial pellets. Using these parameters, the analysis identified 39 different sRNAs enriched in

the *Lp*-EVs (Supplemental Data 1). The 20 highest enriched sRNA that were present in all four biological replicates comprised segments of 12 tRNAs, 4 sRNAs, and 4 fragments of mRNA located either in the coding region or the untranslated region (UTR) of genes (Table 1).

To predict possible functions in the host cells we investigated whether certain of these sRNA might show similarities to human microRNA (hsa-miR) sequences using the miRBase database (http://www.mirbase.org/). We conducted the search with a cut-off *E*-value <12[45]. This revealed that segments of 15 of the 20 *L. pneumophila* RNAs showed some similarity to different hsa-miRs (Table 1). Most interestingly, the nucleotides between 48 and 65 of RsmY showed similarity to the hsa-miR144-3p, a microRNA predicted to interact with the UTR of mRNAs coding for proteins implicated in the host immune response to pathogens, such as *ddx58* mRNA (coding for RIG-I), *rel* mRNA (coding for c-Rel, a NFκB subunit) or *mapk8* mRNA (coding JNK1). The nucleotides between 59 and 73 of the tRNA-Phe showed similarity to the hsa-miR5001-3p, predicted to bind among others to the UTR of *irak1*,

*mavs*, *ago1*, and *mapk9* mRNAs (coding for IRAK1, RIG-I, MAVS, AGO1, and JNK2, respectively). Several of these human target mRNAs, such as the retinoic-acid inducible gene I (RIG-I, encoded by the *ddx58* gene) IRAK1, or MAVS are key players in sensing pathogen-associated molecular patterns and their downstream immune signalling[46]. Thus, RsmY and tRNA-Phe are promising candidate RNAs contained within *Lp*-EVs that might impact the innate immune response of host cells.

**RIG-I and IRAK1 protein levels are suppressed in the host cell in an Lp-EV and RsmY- dependent manner.** To determine whether *Lp*-EVs and specifically the sRNA RsmY modulates the RIG-I-like receptor (RLR) and the Toll-like (TLR) receptor signalling pathway of infected host cells we analyzed the levels of RIG-I and IRAK1 proteins after 8 h of infection of THP-1 cells with *L. pneumophila* wt or its isogenic *rsmY* mutant strain (Δ*rsmY*). In parallel we incubated THP-1 cells with *Lp*-EVs purified from wt *L. pneumophila* or from the Δ*rsmY* strain. We choose RIG-I because of the similarity of RsmY with hsa-miR144-3p, which is predicted to influence RIG-I expression, and IRAK1 because it has been described previously, that *L. pneumophila* infection leads to a suppression of IRAK1 protein levels[39] and, according to the prediction described above, IRAK1 would be affected by tRNA-Phe but not by RsmY. Indeed, RIG-I protein levels were significantly downregulated in THP-1 cells infected with the wt bacteria, but not when infected with the Δ*rsmY* strain. IRAK1 levels were also down regulated as reported previously, but this regulation was independent of RsmY as supposed (Fig. 1G). As the tRNA-Phe gene has only a single locus in the *L. pneumophila* genome, it was not possible to construct a viable knock out mutant to study the impact of tRNA-Phe in this way. Most importantly, RIG-I and IRAK1 protein levels were downregulated in a similar manner when THP-1 cells were incubated with purified *Lp*-EVs only and the downregulation of RIG-I but not IRAK1 was lost when the *Lp*-EVs had been purified from the Δ*rsmY* strain (Fig. 1H). This phenotype was also observed when incubating human monocyte derived macrophages (hMDM) with *Lp*-EVs purified from wt *L. pneumophila* or the Δ*rsmY* strain (Supplementary Fig. 2A). Taken together, the observed downregulation of RIG-I seems to be at least partially RsmY-dependent during infection as well as after incubation with *Lp*-EVs. In contrast, although IRAK1 levels were lowered after 8 h post infection with *L. pneumophila* bacteria and after 3 h incubation with *Lp*-EVs, this was independent of RsmY (Fig. 1G, H).

To better understand the impact of the sRNA contained within the *Lp*-EVs on RIG-I and IRAK1 we aimed to exclude the effects of other factors on the surface of EVs, such as LPS, lipopeptides, or effector proteins that might be inside the *Lp*-EVs that may influence immune signalling regulated by RIG-I and IRAK1. Thus, we transfected THP-1 cells either with in vitro transcribed RsmY, or in vitro transcribed tRNA-Phe and as control (ctrlRNA) with an unrelated RNA or alternatively, with the sequences complementary to RsmY or tRNA-Phe using lipofectamine or electroporation, respectively. Both methods yielded comparable results. When transfecting with either of the sRNA, the RIG-I levels were up-regulated, which is expected as RIG-I is a major cytosolic RNA sensor in the cell sensing the presence of non-self RNAs. However, when comparing ctrlRNA transfection to RsmY and tRNA-Phe transfection, it becomes evident that RsmY and tRNA-Phe transfection led to a significantly reduced induction of the RIG-I protein level, as predicted (Fig. 1I and Supplementary Fig. 2B). Strikingly, IRAK1 protein levels were suppressed only after transfection with tRNA-Phe-RNA, but not with RsmY-RNA (Fig. 1I and Supplementary Fig. 2B). The above-described results indicated that RsmY and tRNA-Phe are

implicated in regulating RIG-I and IRAK1, respectively. Indeed, RNA interaction assays of RsmY and tRNA-Phe with the UTR of *ddx58* (RIG-I) or *irak1* mRNA revealed that in vitro RsmY interacts with the UTR of *ddx58* and tRNA-Phe with the UTR of *irak1* and *ddx58* (Fig. 1J) further suggesting that RsmY-RNA is not responsible for the *Lp*-EV-dependent IRAK1 suppression. Interestingly, interaction between RsmY and the UTR cRel used as control, was also observed but not of tRNA-Phe. This goes in hand, with the analyses of the protein levels of cRel as they were only slightly reduced after transfection of RsmY, but not after transfection of tRNA-Phe or ctrlRNA (Fig. 1I and Supplementary Fig. 2B) indicating that RsmY might have additional targets besides RIG-I, such as cRel.

**Lp-EVs shed during infection contain the sRNA RsmY.** Recently it has been shown that bacterial RNA bound to an RNA binding protein might potentiate interferon-β production by binding to RIG-I[47], however direct action of bacterial sRNAs on host genes is not known yet. Our results strongly suggest that RsmY may act as a transkingdom RNA regulator in the host cell, as it interacts with RIG-I and c-Rel mRNA and is involved in the regulation of their expression (Fig. 1G–J) and was also found enriched in *Lp*-EVs. To perform its regulatory actions in the host cell, RsmY should be present in *Lp*-EVs shed from *L. pneumophila* during infection of human macrophages. Using fluorescent in situ hybridization (FISH) we analyzed if *Lp*-EVs can enter hMDM, and if we can detect RsmY within these *Lp*-EVs after internalization. After 5 h of incubation of hMDM with *Lp*-EVs cells were fixed and hybridized with two RsmY-specific FISH probes (Supplementary Table 1). As shown in Fig. 2A, RsmY packed in *Lp*-EVs were identified in the host cytoplasm. We then infected hMDMs with the *L. pneumophila* wt strain, fixed them after 5 h of infection, and stained with the RsmY specific FISH probe. Indeed, we also could detect the RsmY signal in *Lp*-EV-like structures (Supplementary Fig. 2C), suggesting that RsmY is transported during infection in EVs into the host cell.

**Lp-EVs are taken up by the host cell and seem to escape from the Legionella containing vacuole (LCV) to the cytosol.** Using electron microscopy previous studies have shown that *L. pneumophila* produces EVs when grown in broth culture as well as during intracellular growth after 24 h of infection of the amoeba *Dictyostelium discoideum*[40]. Here we used fluorescent confocal microscopy and imaging in living cells to follow internalized *Lp*-EVs from early time points of uptake as well as during infection of human cells. First, we incubated U2OS cells stably expressing Sec61β-GFP, an ER marker, with purified *Lp*-EVs stained with DiD dye as described above. As expected, the *Lp*-EVs entered the U2OS Sec61β-GFP cells (Supplementary Fig. 2D). To understand, by which pathway the *Lp*-EVs are taken up, we followed the entry of the purified EVs through the plasma membrane in presence or absence of cytochalasin D an inhibitor of actin polymerization and phagocytosis or in presence or absence of dynasore, a dynamin inhibitor[48] acting on clathrin and lipid-raft mediated entry. Cytochalasin D and to a lesser extent also dynasore reduced the entry of *Lp*-EVs significantly (Supplementary Fig. 2E). Thus *Lp*-EVs seem to be taken up preferentially via actin-dependent phagocytosis or macropinocytosis. However, also some dynamin-dependent internalization occurs indicating that *Lp*-EV internalization may involve multiple pathways as proposed for eukaryotic EVs[48].

To determine the localization of *Lp*-EVs shed from *L. pneumophila* during infection we used confocal microscopy of fixed cells as well as confocal imaging of living cells. *L. pneumophila*

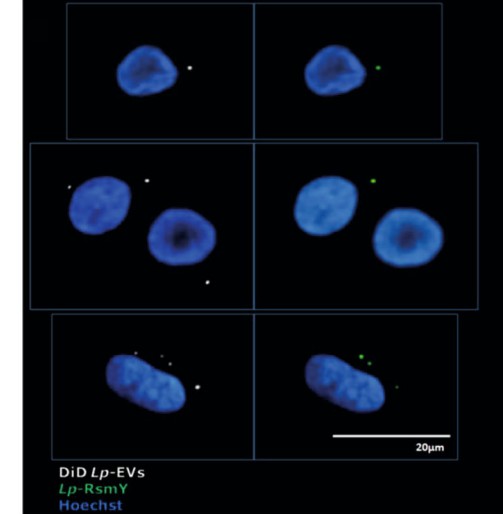

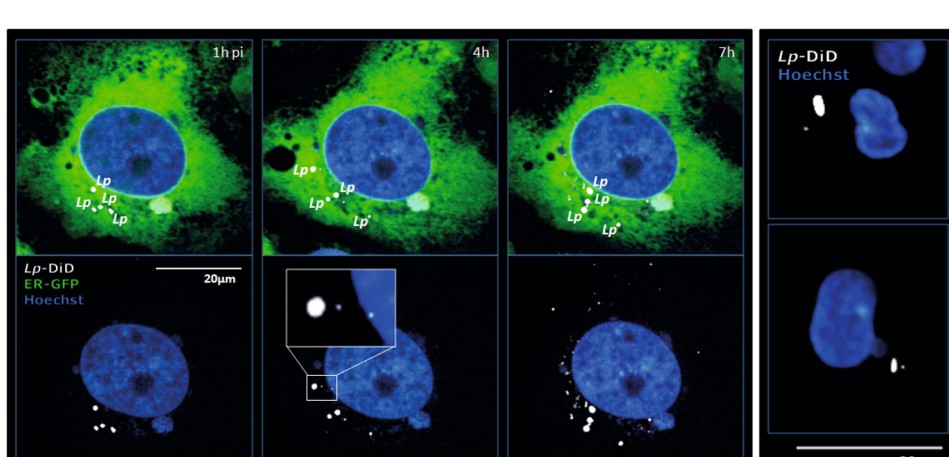

**Fig. 2 Lp-EVs are shed during infection and contain RsmY. A** Representative images of three independent experiments from FISH analyses using probes specific for RsmY. hMDM cells incubated with purified DiD-labelled Lp-EVs. Internalized, DiD-labelled Lp-EVs (white) 5 h post-infection. Blue, nucleus fluorescently stained using Hoechst 33342. **B** Human bone osteosarcoma epithelial cells stably expressing Sec61β-GFP for ER labelling (U2OS- Secβ61β cells; first three panels) and human monocyte-derived macrophages (hMDM) (right panel) were infected with DiD-labelled *L. pneumophila* grown until post-exponential phase (OD4.2). Images of living infected U2OS- Secβ61β cells after 1, 4, and 7 h of infection at 37 °C (5% $CO_2$) taken with an automated confocal microscope (Opera) are shown. Upper panel shows the cells with the green florescence labelling of the ER, lower panel shows the same cells without the green channel. Representative images of three independent experiments. White: DiD-labelled *L. pneumophila* and DiD-labelled Lp-EVs shed during infection; Green: ER; Blue: Nucleus (Hoechst 33342). Right panel shows fixed hMDM cells, 5 h post-infection, analyzed by confocal microscopy. White: *L. pneumophila* and Lp-EVs; Blue: Nucleus (Hoechst 33342).

was stained with DiD and washed carefully to avoid any dye excess. These bacteria were used to infect the U2OS Sec61β-GFP cells or hMDMs in 96 well plates. If Lp-EVs were shed during infection they should also be stained with DiD and thus their budding off from the bacteria and their localization in the host cell should be visible. To record such events, we used an automated confocal microscope to acquire images every 30 min after infection with DiD-labelled *L. pneumophila* thus imaging hundreds of living cells up to 7 h post-infection. As shown in Fig. 2B, after >3 h, little vesicle-like DiD-labelled structures, the Lp-EVs, were visible outside of the predicted LCVs that are closely surrounding the bacteria. The number of Lp-EVs in the host cell cytosol increased over time with *L. pneumophila* remaining intact during the infection followed over 7 h. We repeated this experiment by infecting hMDMs with DiD-labelled *L. pneumophila* and fixing the cells at 5 h post-infection. Again, we observed DiD labelled vesicles, indicating that *L. pneumophila* produces Lp-EVs in both, U2OS cells and primary macrophages.

We then used 3D confocal time-lapse imaging of living, infected U2OS-Sec61β-GFP cells, where we acquired Z-stack images every 30 s for 18 min to capture the event of Lp-EV "release" from the bacteria in real time. Indeed, as shown in Movie 1 and Supplementary Fig. 3A we observed the release of DiD-labelled EVs from *L. pneumophila*, showing that this happens during infection in living, human cells. Taken together, we show that *L. pneumophila* shed EVs during infection of human host cells of which about 30% contain specific sRNAs that are transported within these Lp-EVs across the plasma and LCV membranes in the host cell cytosol.

**Lp-EVs follow the endosomal pathway and interact with the ER.** The localization of Lp-EVs in the host cell may further indicate their role during infection. Thus, purified DiD-labelled Lp-EVs were incubated with U2OS-Sec61β-GFP cells, which allow the precise visualization of ER membranes. In parallel, these

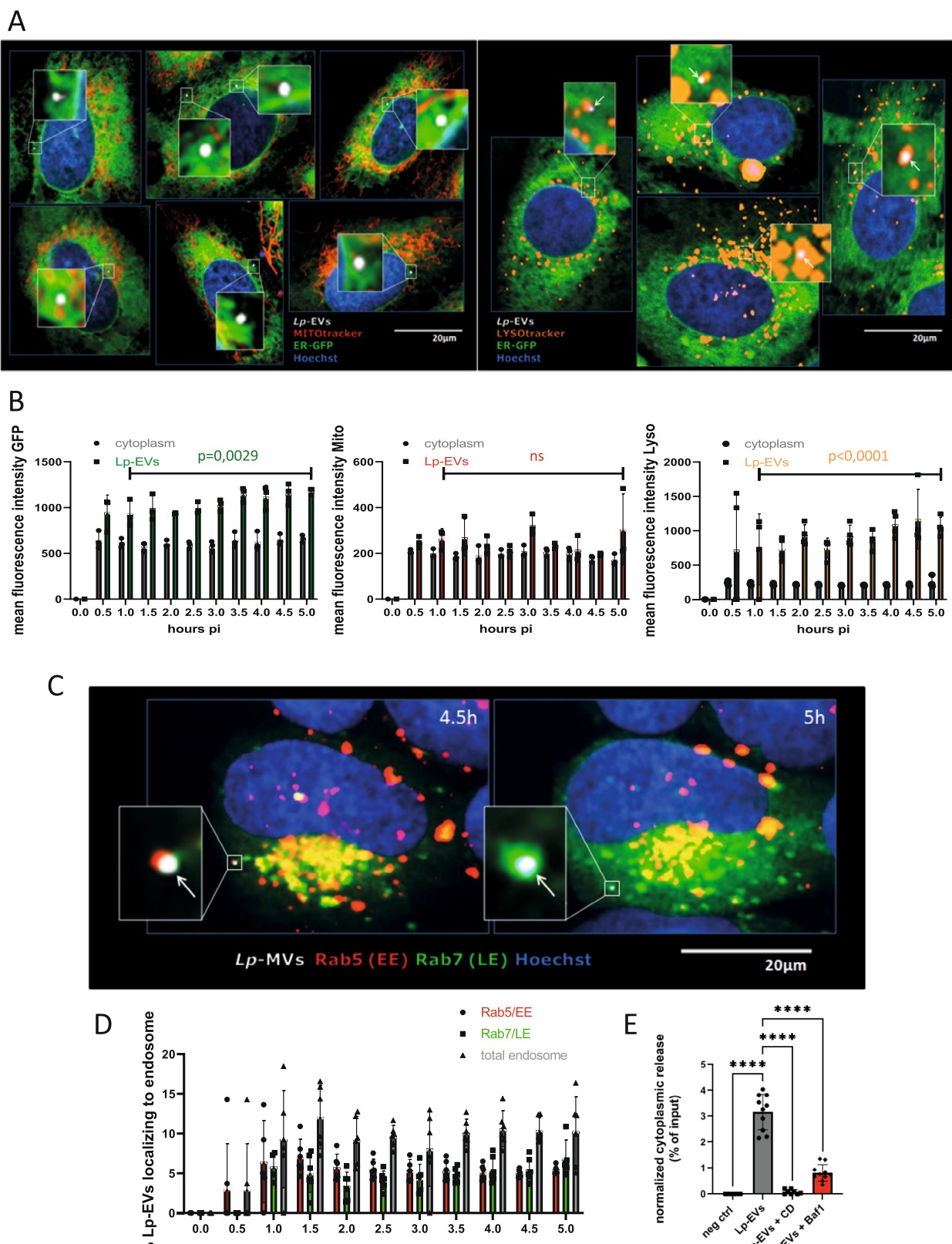

cells were stained with Hoechst 33342 to visualize the DNA/nucleus, with MitoTracker to visualize the mitochondria, or with LysoTracker to reveal acidic organelles. We then used fluorescence-based high content dynamic imaging to follow the internalized *Lp*-EVs in living cells over time (Fig. 3A). The *Lp*-EVs seemed to co-localize with the ER and to acidic organelles,

but not with mitochondria (Fig. 3A). To quantify the events, we applied High-Content Analyses (HCA) of single cells. The DiD signal was used to detect *Lp*-EVs in the cytoplasm of U2OS-Sec61β-GFP cells and the fluorescence of Mitotracker, Lyso-tracker, Sec61β-GFP were measured to analyze in hundreds of living cells at the single cell level if *Lp*-EVs co-localize with

**Fig. 3 Internalized *Lp*-EVs follow the endosomal pathway. A** U2OS-Sec61β cells incubated with DiD-labelled *Lp*-EVs. Live cell image shows co-localization of *Lp*-EVs with ER and mitochondria (left and magnifications) and with acidic organelles (right and magnifications). Representative images of $n = 3$. White: DiD-labelled *Lp*-EVs; Green: ER; Blue: Nucleus; Red: Mitochondria; Orange: acidic organelles. **B** Quantification of co-localization events over time. Live cell image acquisition of multiple fields per well of infected U2OS-Sec61β cells was performed on an automated confocal microscope and the mean fluorescence intensity (MFI) of GFP, Mitotracker and Lysotracker, and *Lp*-EVs was analyzed, suggesting that they co-localize with ER and lysosomes. Data are presented as (mean) SD with each dot representing the mean of up to 600 independently analyzed cells. For statistical analysis an unpaired t-test was performed (two-tailed; $p < 0.05$ significant). Source data provided as Source data file. **C** U2OS cells labelled with Rab5 and Rab7 CellLight incubated with DiD-labelled *Lp*-EVs. Co-localization of *Lp*-EVs with Rab5+ early endosomes (EE) 4.5 h post-infection and with Rab7+ late endosomes (LE) 5 h post-infection. Live cell image acquisition of multiple fields per well of infected U2OS cells was performed over the time at 37 °C (5% $CO_2$) on an automated confocal microscope. Representative images of five independent experiments. White: DiD-labelled *Lp*-EVs; Blue: Nucleus; Red: Rab5; Green: Rab7. **D** Quantification of *Lp*-EVs co-localizing with EE and LE were determined during 5 h of infection. Data are presented as (mean) SD with each dot representing the mean of around 500 independently analyzed cells. Source data provided as Source data file. **E** Quantification of purified HiBiT-GroEL EV content delivery within LgBiT-THP-1 acceptor cells, $+/-$ Bafilomycin A1 (BA1) or Cytochalasin D (CD) treatment. Negative control, LgBiT-THP-1 cells incubated with purified *Lp*-EVs (no HiBiT). The luminescence in the negative control was defined as 0%. Bafilomycin A1 (loss of endosomal acidification) and Cytochalasin D (inhibition of actin polymerization) significantly reduces the content delivery into the host cell. Error bars represent the (mean) SD of at least $n = 8$ biological repeats. For statistical analysis, a two-way ANOVA analysis was performed with $p < 0.0001$ (****). Source data provided as Source data file.

mitochondria, acidic organelles, or the ER. As shown in Fig. 3B the quantitative analyses confirmed that starting from 1.5 h post incubation, automatically detected *Lp*-EVs have statistically significantly enriched mean fluorescence intensity overlapping with GFP and lysotracker signal, but not with Mitotracker, suggesting that *Lp*-EVs co-localize with the ER and acidic organelles, but not with mitochondria. Similarly, when using hMDM cells and pHrodo to label acidic structures and tracking the *Lp*-EVs within these cells for 17 h the *Lp*-EVs clearly co-localize with acidic subcellular structures (Supplementary Fig. 3B), but overall intensities and differences were less pronounced than in the U2OS cell lines. High content analyses also allowed to analyze the number of cells that contained *Lp*-EV. This revealed that an average of 15% of the U2OS cells contained *Lp*-EVs and thus an important background noise is present in *in cellulo* experiments (Supplementary Fig. 3C). For a more in depth analyses, we labelled U2OS cells with CellLight Fluorescent Protein Labelling to visualize Rab5, a marker for early endosomes and Rab7, a marker for late endosomes, and incubated cells with purified DiD-labelled *Lp*-EVs (Fig. 3C). Again, we used automatic confocal microscopy to acquire images of hundreds of living, infected cells every 30 min. We observed that single vesicles originally co-localizing with Rab5-early endosomes (EE) seem to exclusively colocalise with Rab7-late endosomes (LE) at later time points, thus indicating that the *Lp*-EVs might follow the endosomal pathway. When using an HCA approach to analyze *Lp*-EVs in more than 3000 single, infected cells our results indicated that *Lp*-EVs colocalise at early time points slightly more frequently with the EE whereas at later time points (>3.5 h) correlation rates seem to shift more to LE (Fig. 3D). To further investigate the path of *Lp*-EVs within living cells, we followed labelled *Lp*-EVs in single living cells by confocal microscopy in time-lapse experiments and recorded their localization over time. Time-lapse confocal 3D reconstructed movies showed clearly that *Lp*-EVs transit from EE-related to LE-related EVs over time (Movie 2 and Supplementary Fig. 3D). Thus, we propose that about 10% of the total *Lp*-EVs analyzed are transported within the endosomal network towards the ER presumably following a similar pathway as observed for exosome-mediated miRNA transfer between eukaryotic cells[49–51].

To examine whether *Lp*-EVs release their content in the host cell cytosol, we developed a content release assay based on a recent study that followed the delivery of a soluble EV-cargo (HSP70, human homologue of GroEL) within the cytosol of the acceptor cells[42]. This assay was upgraded by taking advantage of split-luciferase complementation system[52]. Briefly, an EV cargo

was tagged with HiBiT (split luciferase 1/2) and isolated EVs were incubated on acceptor cells expressing LgbBit (Split luciferase 2/2) within their cytosol. Luciferase complementation only occurs when EV deliver their cargo into the cytosol of acceptor cells. We had previously shown that the bacterial protein GroEL (*lpp0743*) is present in the *Lp*-EVs. Thus, we tagged GroEL with a HiBiT-tag and in parallel, we transfected THP-1 cells with a LgBiT construct (pCMV-Tag2-LgBiT) under the control of the CMV promoter to express the LgBiT protein in the host cell. If the EV-content is released, Luciferase complementation occurs, and this can be measured. To determine whether the *Lp*-EVs had released their content, we measured luciferase activity *in cellulo* with the Nano-Glo(R) Live Cell Assay System (Promega) after 3 h of incubation with purified *Lp*-EVs containing GroEL-HiBiT. To estimate the total input of *Lp*-EVs containing GroEL-HiBiT, the luciferase activity was quantified after lysis of the *Lp*-EVs using the Nano-Glo (R) HiBiT Lytic detection System (Promega). *Lp*-EVs not containing GroEL-HiBiT were used as negative control. As shown in Fig. 3E, we could detect around 3% luciferase activity of the total GroEL-HiBiT input after 3 h post-infection. This result also corresponds to our co-localization experiments, where after 3 h pi around 5% of *Lp*-EVs co-localized with early or late endosomes, respectively (Fig. 3D). When adding 10 µM of cytochalasin D to the cells, the uptake and/or release was completely abolished (Fig. 3D), suggesting that actin-dependent processes play a crucial role in the endocytosis and/or membrane fusion events.

Strikingly, after the addition of 200 nM Bafilomycin A1, which inhibits endosomal acidification, less than 1% of the total luciferase activity was detected after 3 h pi meaning that less than a third of GroEL-HiBiT proteins were reacting with cytosolic LgBiT protein of the THP-1 cells compared to non-BA1-treated samples. As it was shown recently that Bafilomycin A1 does not change the general uptake of EVs into the host cell[42], this is another hint that the acidification in late endosomes or lysosomes might be an important factor for *Lp*-EV content release.

Taken together, we propose that internalized *Lp*-EVs can follow the endosomal pathway of the host cell leading to transient contacts with the ER ("ER scanning") where the bacterial sRNA could be released ("endosomal or lysosomal escape")[53–57], at least partially dependent on an acidic environment, as previously proposed for eukaryotic EVs[58]. There, the bacterial sRNA like RsmY might be processed and/or is directly loaded into the RISC complex leading to the downregulation of the protein expression of Rig-I and IRAK1, and probably also other target proteins, in a miRNA-like manner.

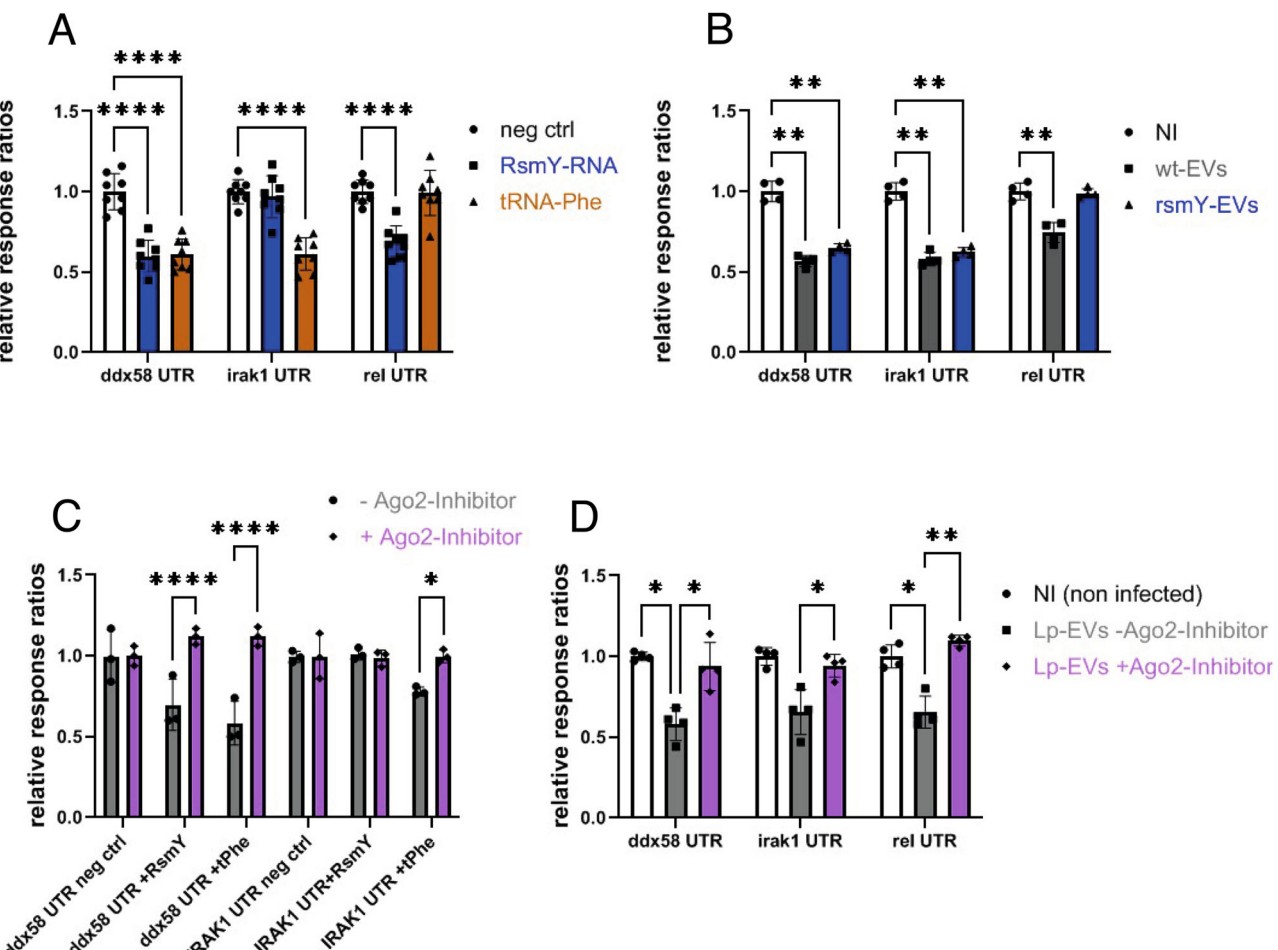

**Fig. 4 RsmY base pairs with the UTR of RIG-I and RsmY activity is modulated by Ago2 inhibition. A** Dual luciferase reporter assay showing luciferase activities of THP-1 cells transfected with random short *Lp*-RNA (ctrlRNA), RsmY, or tRNA-Phe. Results are shown as relative luciferase activity (ratio Firefly/Renilla activity), where luciferase activity of ctrlRNA is defined as 100%. Bars indicate the (mean) SD of eight independent experiments. For statistical analysis, a two-way ANOVA was performed with $p < 0.0001$ (****). Source data provided as Source data file. **B** Dual luciferase reporter assay showing luciferase activities of THP-1 cells in absence (NI) or presence of wt or a *ΔrsmY Lp*-EVs. *Lp*-EVs affect protein expression depending on the presence of the un-translated regions (UTR) of *ddx58* (RIG-I), *irak1*, or *rel* (cRel). Results are shown as relative luciferase activity (ratio Firefly/Renilla activity), luciferase activity of THP-1 cells (NI) was defined as 100%. Bars show the (mean) SD of $n = 7$ independent experiments. Statistical analysis, a two-way ANOVA with $p < 0.05$ significant (*), $p < 0.01$ very significant (**) and $p < 0.001$ extremely significant (***) was performed. Source data provided as Source data file. **C** Relative luciferase activities of THP-1 cells transfected with RsmY or tRNA-Phe. RsmY and tRNA-Phe affect protein expression significantly less in presence of an Ago2-Inhibitor. The bars indicate the (mean) SD of three independent experiments. Statistical analysis, a two-way ANOVA analysis was performed with $p < 0.0001$ (****). Source data provided as Source data file. **D** Dual luciferase reporter assay showing the relative luciferase activities (ratio Firefly/Renilla activity) of THP-1 cells incubated with *Lp*-EVs$+/-$ Ago2-Inhibitor. When inhibiting Ago2, the suppressing effect of *Lp*-EVs on relative luciferase activity depending on the presence of the un-translated regions (UTR) of *ddx58* (RIG-I), *irak1* or *rel* (cRel) is significantly reduced. The bars show the (mean) SD of $n = 4$ independent experiments. Statistical analysis, a two-way ANOVA with values of $p < 0.05$ significant (*), $p < 0.01$ very significant (**) and $p < 0.001$ extremely significant (***) and $p < 0.0001$ (****) was performed. Source data provided as Source data file.

**RsmY targets Rig-I in a miRNA-like manner.** Cells incubated with *Lp*-EVs purified from *L. pneumophila* wt but not those purified from *L. pneumophila* lacking RsmY showed decreased levels of RIG-I protein. Furthermore, RsmY shows some similarity with has-miR144-3p that is predicted to influence RIG-I expression. Thus, we propose that RsmY might act like a human miRNA by base pairing with the UTR of RIG-I and thereby influence its expression. To test this hypothesis, we performed a dual luciferase reporter gene assay, a powerful tool for confirming gene regulation by miRNAs. We introduced the UTR regions of the *ddx58* (RIG-I), *irak1* (IRAK1), or *rel* (cRel) genes downstream of the *luc2* (Firefly) luciferase and a nonspecific (non-silencing), negative ctrlRNA, RsmY, or tRNA-Phe under the control of the H1 promoter. The *hRluc* (Renilla) expression and activity were used as an internal control and the changes in the ratios of

normalized Firefly- *versus* Renilla-luciferase activities at the different conditions were determined by comparing to the negative control. As shown in Fig. 4A, the relative luciferase activity is significantly reduced when Luc2 was fused to the UTR of *ddx58*/ Rig-I and RsmY and tRNA-Phe were present, indicating again that both regulatory RNAs down-regulate RIG-I expression. In contrast, only tRNA-Phe impacts the activity of the IRAK1-UTR, and only RsmY significantly affects the expression of the luciferase gene fused to cRel-UTR. Furthermore, when conducting the same assay with the complementary RNA sequence of RsmY (as-RsmY) and tRNA-Phe as-tRNA-Phe) luciferase activity remained unchanged, highlighting that only the correct strand sequence impacts expression (Supplementary Fig. 4A). Additionally, we undertook the dual luciferase reporter gene assay described above also in primary cells, to rule out the possibility

that the result is due to the cell line used. Indeed, when repeating the above-described experiment in CD14+ cells isolated from human blood we obtained the same result as in THP-1 cells (Supplementary Fig. 4B). These results are also in agreement with the results obtained after RNA transfection (Fig. 1I and Supplementary Fig. 2B), further supporting our results that RsmY interacts with the UTR of the RIG-1, and tRNA-Phe with the UTR of the *irak1* encoding gene and indeed can behave like eukaryotic micro-RNAs. Additionally, we analyzed the effect of *Lp*-EVs purified from the *L. pneumophila* wt or the *L. pneumophila* Δ*rsmY* strain on the different UTRs by measuring the luciferase activity (Fig. 4B). Also, here we observed a clear dependency of the luc2 activity on the presence of *Lp*-EVs and of RsmY similar to what we have seen by western blot (Fig. 1H) supporting the idea that *Lp*-EVs participate in manipulating the immune response.

Argonaute family proteins play a crucial role in RNA-induced silencing complex (RISC). Thus, to further analyze if RsmY acts in a miRNA like manner we investigated whether argonaute-2 (Ago2), the only member with catalytic activity and an essential role within the RISC complex to regulate small RNA guided gene silencing processes[59,60] impacts RsmY and/or tRNA-Phe activity. Indeed, the suppressive effect of RsmY, tRNA-Phe or of *Lp*-EVs on the relative luciferase activity of Luc2 fused to the UTR of *ddx58*/Rig-I or the UTR of *irak1* was significantly reduced when Ago2 inhibitor was added (Fig. 4C, D). This suggests that the presence of a functional Ago2 is necessary for *Lp*-EV RNAs to interact with *ddx58* and *irak1* UTRs *in cellulo* and show that Ago2 and thus probably RISC-mediated silencing are involved in *ddx58* and *irak1* expression during *L. pneumophila* infection. However, secondary effects of Ago2 inhibition may also influence this result, since endogenous human miRNA are described to play an important role in the regulation of the expression of a variety of immune-related proteins, including Rig-I and IRAK1[61] (Supplementary Fig. 4C). Yet, as transfection of RsmY-RNA did not influence luciferase activity of Luc2 fused to the *irak1*-UTR independently of Ago2 inhibition, the observed impact of tRNA-Phe on IRAK1 cannot solely depend on the effect of endogenous has-miRNA-silencing, but it further suggests that indeed *L. pneumophila* RsmY has a significant impact on protein expression in a miRNA-like and Ago2-dependent manner.

To determine if Ago2 plays a direct role in *Lp*-sRNA-mediated gene silencing, we analyzed whether *Lp*-sRNAs and in particular RsmY directly interact with human Ago2 during infection. We infected THP-1 cells with *L. pneumophila* and used has-Ago2 antibodies for Ago2-immunoprecipitation experiments followed by sequencing. To validate our approach, we first analyzed whether hsa-miRNAs known to interact with Ago2 were among the sequences obtained. Indeed, we identified 72 known Ago2-interacting hsa-miRNAs (Supplementary Table 2), but most excitingly we also identified RNAs derived from *L. pneumophila*, in particular, we identified RsmY in two of our three pull downs. Given the fact that only about 50% of the THP-1 cells are infected by *L. pneumophila*, that only 30% of *Lp*-EVs contain RNA (Supplementary Fig. 1C), and that according to our assays only about 3% of the *Lp*-EV's cargo is released in the tested condition (Fig. 3E), the probability to identify *Lp*-derived RNAs in the bulk of human RNAs is very low. Thus, although the results are not statistically significant, our Ago2-CLIP indicated that RsmY seem to directly interact with Ago2 during infection. These results further support our model that RsmY and other *L. pneumophila* RNAs can act in a mi-RNA like manner in the host cell.

Thus, *Lp*-EVs not only follow similar pathways as observed for exosomes, but the bacterial sRNA transported within *Lp*-EVs seem to have a similar mode of action as eukaryotic miRNAs. They can bind to UTRs of target mRNAs leading to a reduced

translation of the corresponding protein and/or to a destabilization of the mRNA and consequently to a downregulation of the protein levels.

**RsmY decreases RIG-I protein expression in the host cell and modulates downstream signalling after uptake of Lp-EVs or during bacterial infection.** To investigate the impact of RsmY contained within *Lp*-EVs on the downstream activation of the RLR and TLR signalling pathway, we analyzed the protein levels of cRel and the phosphorylated forms of p-TBK1, p-IRF3, p-IRF7, p-IκBα, and Co in THP-1 cells incubated with *Lp*-EVs purified either from wt *L. pneumophila* or from the Δ*rsmY*-mutant strain. After 3 h post incubation the protein levels and/or phosphorylation status were compared to that of non-infected cells. As shown before, in cells incubated with the *Lp*-EVs purified from the Δ*rsmY*-strain protein level of Rig-I were significantly increased as compared to the wt-*Lp*-EV incubated cells (Fig. 1H). Furthermore, the amounts of activated (phosphorylated) forms of IRF3, IRF7, and TBK1, which are part of the downstream signalling cascade of the RLR and TLR immune signalling pathways, were significantly increased in the cells incubated with *Lp*-EV from the RsmY-mutant (Fig. 5A and Supplementary Fig. 4D). Similar results were obtained when THP-1 cells were infected with *L. pneumophila* wt and the protein levels of the above-mentioned proteins were compared to the type IV secretion mutant Δ*dotA*, the Δ*rsmY*-mutant, and the Δ*letA* strain after 1, 3, 6, and 8 h of infection (Supplementary Fig. 4E). The LetA mutant strain was also analyzed as the two-component system LetA/LetS regulates the expression of RsmY[62]. As noted above, tRNA-Phe has only a single locus in the *L. pneumophila* genome, hence it was not possible to construct a viable knock-out mutant. Protein levels of RIG-I and cRel decreased significantly in a time-dependent manner in cells infected with the wt but increased in cells infected with Δ*dotA*- and Δ*letA*-mutant strains. Strikingly, in cells infected with the Δ*rsmY*-mutant strain protein level of Rig-I were also significantly increased as compared to the wt-infected cells (Supplementary Fig. 4E). These results further show that RsmY indeed impacts the RLR signalling pathway. In contrast, the levels of the phosphorylated forms of IκBα, a negative regulator of NF-κB, were not significantly different when compared to the wt infected cells, indicating that RsmY, although targeting cRel after transfection, does not have a significant influence on the activation of the initial NF-κB response (Fig. 5A and Supplementary Fig. 4E).

**RsmY contained in Lp-EVs and RsmY alone decrease the IFN-β response of the host cell.** The RLR response, but also TLR signalling via IRAK1 are linked to interferon type I expression and secretion. Thus, RsmY mediated downregulation of RIG-I and the attenuation of the downstream transcription factors should influence IFN-β secretion of the host cells[63]. We thus first analyzed the IFN-β concentration in the supernatants of THP-1 cells infected with the wt *L. pneumophila* or the Δ*rsmY*-mutant strain. We quantified the amount of extracellular IFN-β by performing an ELISA with the supernatants at different time points of the infection. However, no significant differences were found when IFN-β concentrations of cells infected wt *L. pneumophila* were compared to cells infected with the Δ*rsmY*-mutant strain (Supplementary Fig. 5A), suggesting that this approach does not reveal the influence of RsmY on IFN-β secretion as additional factors may also influence IFN-β levels as known for several bacterial and viral infections[63], and these combined effects are measured. Thus, to measure specifically the impact of *Lp*-EVs containing RsmY, we analyzed the extracellular concentration of IFN-ß in the supernatant of THP-1 cells after incubation with *Lp*-

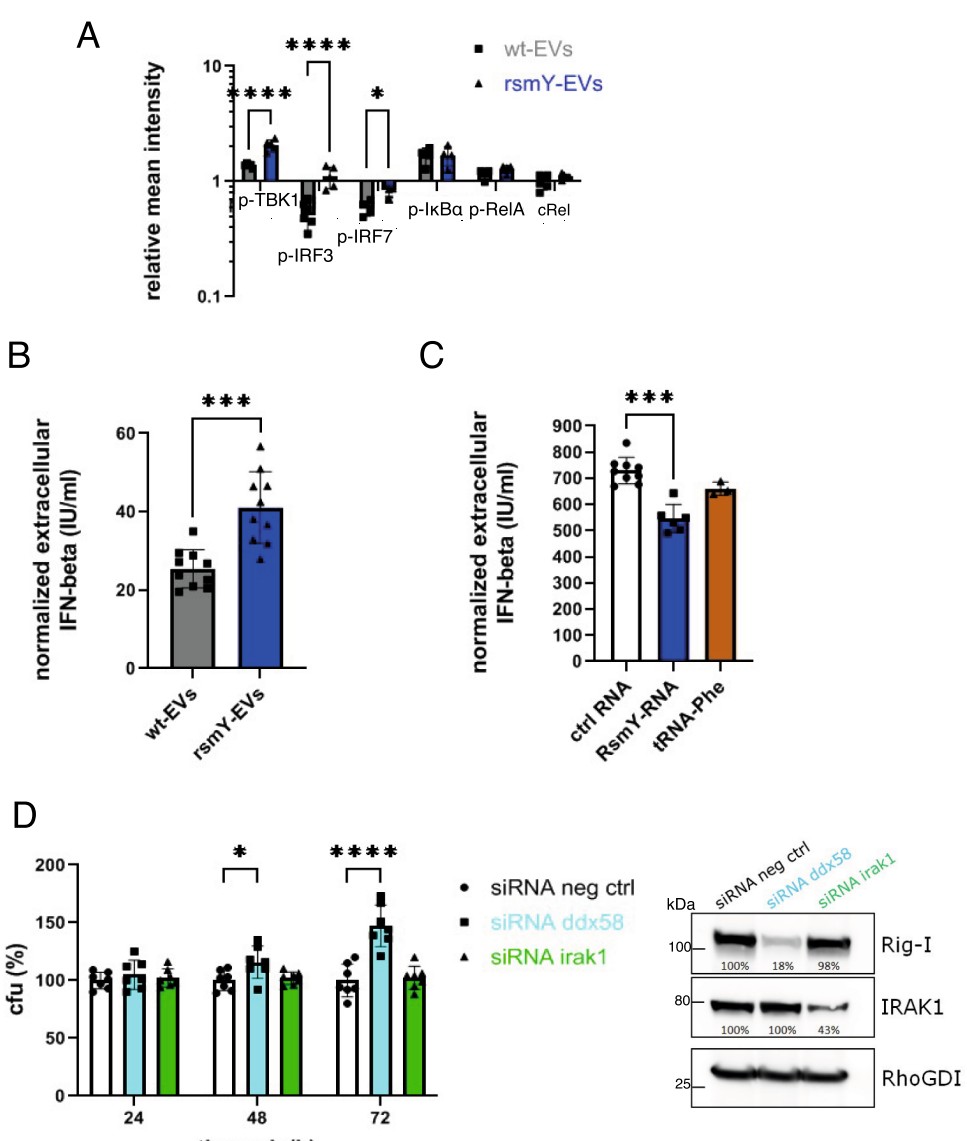

**Fig. 5 RsmY transfection impacts RIG-I expression or when contained in *Lp*-EVs and decreases IFN-ß. A** Quantitative analyses of the protein and phosphorylation levels of key proteins of the RLR and TLR signalling pathway analyzed after 3 h of-incubation of THP-1 cells with wt or the *ΔrsmY Lp*-EVs. wt (grey), *ΔrsmY* (blue). Mean intensities of the proteins, normalized against the value of non-infected cells and the RhoGDI loading control. Bars show the (mean) SD of *n* = 4 independent experiments. For statistical analysis, a two-way ANOVA with values of $p < 0.05$ significant (*), $p < 0.01$ (**), $p < 0.001$ (***) and $p < 0.0001$ (****) was performed. Source data provided as Source data file. **B** Extracellular IFN-β concentration of THP-1 cells incubated for 3 h with wt or *ΔrsmY Lp*-EVs. The bars indicate the (mean) SD of *n* = 10 independent experiments using an unpaired t-test for statistical analysis with two-tailed values of of $p < 0.05$ significant ($p = 0.0002$). **C** THP-1 cells were transfected with in vitro transcribed RsmY, tRNA-Phe RNA, or non-specific control RNA (ctrlRNA). The value for the ctrlRNA comprises the results of two control RNAs, a random short *L. pneumophila* DNA and the anti-sense sequences of RsmY (as-RsmY). Bars show the (mean) SD of *n* = 3 independent experiments using an unpaired t-test for statistical analysis (two-tailed), $p < 0.05$ significant ($p = 0.0007$). In **B** and **C** the IFN-ß concentration of the supernatant was quantified and normalized against the values of the corresponding non-infected/non-transfected experiments. **D** THP-1 cells were transfected with siRNA targeting Rig-I (*ddx58*), IRAK1 (*irak1*), or a scramble control (neg ctrl). The protein expressions of RIG-I and IRAK1 were analyzed at time point T0, before infection with *L. pneumophila*. The graph shows the percentage of colony forming units (% cfu) at 24, 48, and 72 h post-infection relative to the scramble siRNA (neg ctrl). Bars show the (mean) SD of at least *n* = 7 independent experiments. For statistical analysis, a two-way ANOVA with values of $p < 0.05$ significant (*), $p < 0.01$ very significant (**), $p < 0.001$ extremely significant (***) and $p < 0.0001$ (****) was performed. Source data provided as Source data file.

EVs purified either from wt *L. pneumophila* or from the *ΔrsmY* mutant strain. Indeed, as shown in Fig. 5B, internalization of *Lp*-EV containing RsmY (purified from the wt strain) induces less IFN-ß secretion by the host cells, than those infected with *Lp*-EVs from which RsmY is absent (purified from the *ΔrsmY* strain). These results suggest that *Lp*-EVs containing RsmY dampen IFN-β secretion of infected human cells. To further substantiate this finding, we transfected THP-1 cells with RNA to activate RIG-I-

dependent IFN-β secretion. The transfection of RsmY-RNA and to a lesser extent of tRNA-Phe lead to a significant lower IFN-β secretion of the host cells when compared to the nonspecific control RNA (Fig. 5C).

To further investigate the impact of RsmY on the host immune response, we performed RNAseq analyses comparing THP-1 cells incubated with *Lp*-EVs purified from wt bacteria or *Lp*-EVs purified from the RsmY mutant strain at 3 h pi. Our results

revealed that only slight but not significant differences in the host cell transcriptome were present. Indeed, the difference in the IFN-β levels we observed between THP-1 cells treated with *Lp*-EV purified from the wt strain and *Lp*-EVs purified from the Δ*rsmY* strain are apparently not enough to see significant changes in the transcription of Interferon stimulated genes (ISG) at transcript level. However, this is not surprising as accumulating evidence indicates that ISG expression is not solely dependent on IFN-β, but that ISGs can also be up-regulated directly after a pathogen infection independent of IFN-β signalling, thus the network underlying the regulation of the ISG is much more complex[64–66]. Furthermore, RsmY and tRNA-Phe seem to act in concert on the host immune response and a knockout of both, which is unfortunately not possible to achieve, might lead to more important effects on the transcriptional level. Overall, the differences on transcript level between wt *Lp*-EV and Δ*rsmY*-EV treated cells are small and not significant including the transcripts of *ddx58* (Rig-I) and *irak1* suggesting also that post-transcriptional effects may play a more dominant role in the regulation of protein expression by *Lp*-EVs.

Taken together, our results indicate that the absence of RsmY in *Lp*-EVs resulted in an up-regulation of IFN-ß in the supernatant, whereas the transfection of RsmY-RNA lead to a reduced IFN-ß secretion compared to the control RNA, in agreement with our model that RsmY contained within the *Lp*-EVs is biological active in the host cell and plays a role in dampening the immune response of host cells to infection.

**IFN-β levels, Ago2 inhibition, and Rig-I silencing impact replication of L. pneumophila that is modulated by Lp-EVs**. To investigate the influence that IFN-β secretion, partly induced by RsmY, has on infection, we treated THP-1 cells with different concentrations of IFN-β and analyzed the replication phenotype of *L. pneumophila*. We show that increasing concentrations of extracellular IFN-β reduce intracellular replication of *L. pneumophila* in THP-1 cells, whereas high concentrations of IL-1β have no impact (Supplementary Fig. 5B). We then pre-treated the THP-1 cells with *Lp*-EVs either purified from wt bacteria or with *Lp*-EVs purified from the Δ*rsmY* strain. In cells pre-treated with wt *Lp*-EVs we observed a significantly higher replication of *L. pneumophila* than in cells incubated with *Lp*-EVs purified from the Δ*rsmY* strain (Supplementary Fig. 5C). To further analyze the mechanism, we specifically down-regulated the expression of *ddx58* and *irak1* by siRNA-mediated gene silencing. Protein levels of Rig-I or IRAK1, respectively were reduced by 60–80% at the time point of infection compared to scramble transfected control cells (Fig. 5D). After 24 hpi, no significant differences in the replication of *L. pneumophila* were detected, but at later time points (48 and 72 hpi), the number of bacteria in cells where DDX58 (Rig-I) was downregulated by siRNA was increased by up to 50% further confirming that suppression of Rig-I is beneficial for intracellular replication of *L. pneumophila*, as has also been described previously[67]. Additionally, we transfected THP-1 cells with RsmY RNA or its anti-sense sequence (as-RsmY) and infected these cells. *L. pneumophila* replicated significantly better in the cells transfected with RsmY-RNA, similar to what was observed after siRNA knockdown of *ddx58*, again showing that RsmY has a beneficial effect on *L. pneumophila* replication (Supplementary Fig. 5D). Surprisingly, no differences in *L. pneumophila* replication between scramble control and IRAK1 knockdown THP-1 cells were observed suggesting that IRAK 1 has no major influence on intracellular growth of *L. pneumophila* (Fig. 5D). To confirm these results, we pre-treated the THP-1 cells with the IRAK1/4 Inhibitor I (BCI-137). Again, no significant changes in intracellular replication of *L. pneumophila*

were observed (Supplementary Fig. 5E), differently to what was reported previously[39]. In contrast, inhibition of Ago2 in THP-1 cells prior to infection using BCI-137 significantly reduced intracellular replication of *L. pneumophila*, again indicating that a functioning Ago2 is necessary for its optimal proliferation (Supplementary Fig. 5E).

Finally, to further characterize the impact of *Lp*-EVs on the host immune response, we incubated THP-1cells with agonists that mimic pathogen-associated molecular patterns (PAMPs) and measured extracellular IFN-β concentrations after pre-treatment with *Lp*-EVs that were either purified from wt bacteria or from the Δ*rsmY* strain. Indeed, the IFN-β response of certain TLR agonists was dampened after pre-incubation with wt *Lp*-EVs but less with Δ*rsmY*-EVs further pointing to the influence of RsmY on the host immune response. In particular, the IFN-ß response triggered by agonists for TLR1/2/5/6 and TLR8 was significantly reduced when pre-treating the cells with *Lp*-EVs (Supplementary Fig. 6). The IFN-β response to TLR9 agonist CpG instead was even more pronounced after pre-incubation of the THP-1 cells with *Lp*-EVs compared to control experiments, probably due to synergetic effects of multiple ligand stimulations. In contrast, agonist stimulation of TLR3, TLR4, or TLR7 was not affected by *Lp*-EV-treatment, whereas inhibition of Ago2 slightly induced the extracellular IFN-β levels after *Lp*-EV-treatment (Supplementary Fig. 6).

Taken together, these results further support our model that the two sRNAs RsmY and tRNA-Phe transferred in the host cell within *Lp*-EVs, modulate the host innate immune response in an Ago2-dependent and miRNA-like manner for the benefit of intracellular growth of *L. pneumophila*.

## Discussion

The analyses of the RNA content of *L. pneumophila* extracellular vesicles (*Lp*-EVs) and the study of its functional role in infection provides important new insights into host-pathogen communication and trans-kingdom signalling. Our dataset demonstrates that *L. pneumophila*-RNAs, in particular the sRNA RsmY and the tRNA-Phe, can be transported within *Lp*-EVs into the host cell where they are biological active and can modulate the host immune response in a miRNA-like manner.

We found that *Lp*-EVs show a unique RNA profile, different from the RNA content of the bacterial cells from which they derived, as 39 sRNAs are highly enriched in the *Lp*-EVs suggesting that RNA is not randomly incorporated. Indeed, distinct RNA content has also been reported for bacterial EVs shed from *Escherichia coli* or *Pseudomonas aeruginosa*[45,68]. Eukaryotic EVs have also been described to contain distinct microRNA signatures that are proposed as biomarker for diseases[69] and can even contain mycobacterial transcripts when released from infected macrophages[70]. The sRNAs enriched in the *Lp*-EVs share a common feature of EV-RNAs, which is a size of <200 nucleotides and a stable secondary structure for most of them. It was reported for eukaryotic EVs that selectivity of RNA loading might be affected by secondary structures, sequence motifs, or binding to certain proteins[71]. However, secondary structure alone or simply the abundance of certain sRNAs is not sufficient to explain the selective packaging of *e.g.* RsmY in *Lp*-EVs as RsmZ, the most abundant *L. pneumophila* sRNA, exhibits a secondary structure very similar to RsmY[62] but it is predominantly present in the bacterial cell, and not enriched within the *Lp*-EVs. A recent report suggested that the RNA-binding protein Hfq might affect bacterial membranes and might have a role in exporting sRNAs into the periplasmic space and outside of the bacterial cell[72], thus Hfq could play a role in transporting sRNAs of *L. pneumophila* into EVs, but at the moment the mechanism of "RNA selection"

and cargo loading remains unknown and needs further investigation.

To act efficiently, the sRNAs contained in the *Lp*-EVs need to be delivered to the host cell at specific locations. It is tempting to assume that EV-content delivery is an ancient and common secretion mechanism as EVs are produced by organisms from all domains of life, gram-negative and gram-positive bacteria, archaea, and eukaryotes among those fungi and protozoan parasites[31,32,45,51,68,73–75]. We show that *Lp*-EVs are released during infection of the host cell where they seem to cross the LCV and the plasma membrane and enter the sorting of the eukaryotic endosomal pathway, during which they establish transient contacts with the ER. Thus, *Lp*-EV sorting might be comparable to exosome-mediated delivery of miRNAs, where exosomes are sorted into endosomal trafficking circuits that are targeted to scan the ER as a possible site of cargo release[49,76]. Indeed, it has been proposed that the ER and late endosomes are major sites of protein synthesis and could have a central function for RISC complex loading where de novo synthesized target-mRNA becomes associated with miRNA and the Ago2 protein[53–57]. Our data support a model in which bacterial RNA is transported protected from detection and degradation within the *Lp*-EVs that are directed to the ER membranes where the sRNA content may be released. The mechanistic principles underlying such an endosomal sorting and escape of the bacterial RNA from the EVs is still unknown. However, experiments on RNA nanoparticles and viral escape from endosomes reveal that the differing lipid composition of late endosome and lysosome membranes (like lysobisphosphatic acid, or zwitterionic phospholipids) can increase the ability to fuse with EV membranes or can induce pore formation in the endosomal membrane. Furthermore, the lower pH can trigger changes in the conformation and activity of proteins like phospholipases or proteolytic enzymes facilitating the entry in the cytosol[77,78]. Similar mechanisms might play a role in *Lp*-EV cargo release. Interestingly, a proteome analysis of the *Lp*-EV protein content by two-dimensional gel electrophoreses identified several proteases and phospholipases within the *Lp*-EVs that might participate in such an endosomal escape[40]. It was also shown that *Lp*-EVs fuse to reconstituted liposomes composed of different phospholipds and it was proposed that membrane fusion might be a general mechanism of EV-cargo delivery[37]. Thus, we cannot exclude that a certain amount of *Lp*-EVs may also directly fuse with the host membranes and deliver the RNAs into the cytoplasm. However, our data indicate a significant uptake of *Lp*-EVs depending on actin cytoskeleton rearrangements and support their transport through the endosomal pathway into the host cytosol. Such a mechanism would be consistent with the content delivery of EVs derived from mammalian cells that seems to require protein and pH-dependent fusion between EVs and endosomal membranes[42,58]. It is possible that this mechanism and the machinery are conserved through evolution.

Our bioinformatics analyses revealed that the sRNA RsmY and the tRNA-Phe show similarity to human microRNAs, for example, RsmY to miR-144 predicted to act amongst others on RIG-I. Most interestingly, that miR-144 is important in infection was recently shown for influenza, as it impacts viral replication in lung epithelial cells by directly regulating the TRAF6 expression and consequently, affects the IRF7-dependent transcriptional network negatively[79]. Thus, our finding was intriguing and suggested that the bacterial sRNAs might act like miRNAs in the host cell during infection. Indeed, we show that two of these sRNAs, RsmY and tRNA-Phe, can base pair with the UTR of eukaryotic target genes in a miRNA-like manner significantly attenuating the protein expression of at least three proteins that are important players in the RLR and TLR response, RIG-I, cRel, and IRAK1. It

was reported previously that *Lp*-EVs affect the infectivity of *L. pneumophila* by suppressing IRAK1[39]. Here we identify tRNA-Phe as a new player implicated in the downregulation of IRAK1 expression and reveal that RIG-I is an additional target.

RIG-I is a key sensor that upon recognizing foreign RNA in the host cell mediates the transcriptional induction of type I interferons[46]. Similarly, our analyses of the secretion of INF-β after transfection of RsmY or tRNA-Phe into host cells or incubation of wt or RsmY depleted *Lp*-EVs showed decreased INF-β secretion when RsmY and to a lesser extent also tRNA-Phe were present. This uncovers that translocation of *L. pneumophila* sRNAs contained in EVs leads to an attenuation of the host immune response and aids the survival and replication of this pathogen during infection. Thus, direct miRNA-like regulation of the expression of key sensors and regulators of the innate immune response is a striking feature of *L. pneumophila* host-pathogen communication.

Remarkably, the identified "bacterial miRNAs" have a dual function, not only can they act as trans-kingdom signalling molecules, but they are also important in the bacterial cell. RsmY is a regulatory RNA implicated in regulating the life cycle of *L. pneumophila*, and tRNA-Phe is involved in protein synthesis. Whether *L. pneumophila* sRNAs or that of other bacteria embedded in EVs impact eukaryotic cells in diverse ways and if certain common strategies are used to interfere with the host immune response need to be investigated. Indeed, RNA-based trans-kingdom signalling might be a more general mechanism employed by bacteria for establishing virulence as well as symbiosis and other interactions occurring between bacterial and eukaryotic cells.

## Methods

**Cell lines, bacterial strains, and infection assays**. hMDM (THP-1 cell line purchased from ATTC/TIB-202™) were cultivated in RPMI 1640+GlutaMAX medium (Life Technologies) and human bone osteosarcoma epithelial cells stably expressing Sec61b-GFP for ER labelling (U2OS-Sec61b) published previously[18], in DMEM + GlutaMAX (Life Technologies), both supplemented with 10% heat-inactivated FBS (Life Technologies). The cells were incubated at 37 °C with 5% $CO_2$ in a humidified atmosphere. Human monocyte-derived macrophages (hMDM) or CD14 + cells were isolated and differentiated as previously described[27]. L. pneumophila strain Paris and its derivatives were grown on ACES-buffered charcoal yeast-extract (BCYE) medium at 37 °C and *E. coli* was grown in Luria-Bertani (LB) broth and agar. For knock out and complementation constructs, the corresponding antibiotic was added with following concentrations: apramycin 15 μg/ml, kanamycin 15 μg/ml, or chloramphenicol 10 μg/ml. Mutant strains ΔletA (lpp2699), ΔrsmY and ΔdotA (lpp2740) were constructed previously[27,62]. For infection, L. pneumophila or knock out mutant strains from above were grown until post-exponential phase (OD4.2) in *N*-(2-acetamido)-2-aminoethanesulfonic acid (ACES)-buffered yeast extract broth (BYE). If indicated bacteria were labelled with Vybrant DiD-dye (Thermo Scientific) for 30 min, and carefully washed five-times in PBS. THP-1 and U2OS cells were infected with an MOI of 50, hMDM cells with an MOI of 10 as published previously[80].

**Purification, characterization, and incubation of *Lp*-EVs with host cells**. L. pneumophila strains were grown in BYE until post-exponential phase (OD4.2). Bacterial cells were harvested by centrifugation at 5.000 × g, 4 °C for 15 min and the pellet was frozen at −80 °C for further use. The supernatant was twice passed through a 0.2 μm PES membrane Stericup® Quick Release (Millipore) and re-centrifuged for 15 min at 15.000 × g, 4 °C. The supernatant was treated with RNaseA/T1 (Thermo Scientific) at a final concentration of 2 μg/ml RNaseA for 1 h at 37 °C followed by centrifugation at 150.000 × g for 2 h at 4 °C to pellet the *Lp*-EVs. The *Lp*-EV-pellet was washed, re-centrifuged, and resuspended in PBS. For incubation experiments with U2OS or hMDM for FISH, the purified *Lp*-EVs were labelled with Vybrant DiD-dye (Thermo Scientific), 30 min at 37 °C, and subsequently cleaned-up using Exosome Spin Columns (MW3000, Thermo Scientific) for removal of unincorporated dye. Human cells were incubated with *Lp*-EVs at an MOI of 10 (according to flow cytometry dye-labelled events). The values obtained by flow cytometry are much lower than those obtained with the ZetaView analysis, in which we measure a concentration of *Lp*-EVs about 1000 times higher as compared to the flow cytometry data. As we adjusted our MOI in all experiments to the conventional flow cytometry data, we indicate throughout the manuscript that the MOI was estimated according to conventional flow cytometry. It also needs to be noted, that we analyzed how many cells indeed take up of *Lp*-

EVs to estimate the amounts of *Lp*-EVs impacting the host cell. We used high content image analyses to analyze the number of EVs detectable in infected U2OS cells. We observed on average only about 10% of cells that contained detectable EVs. Thus, even with the relative high amount of EVs used according to ZetaView quantification the cell cultures were not saturated with *Lp*-EVs but a high background of noninfected cells that moderate the measurable output is present. Also, the amount of EVs we used did not lead to cell death of U2OS or hMDM cells, even after 17 h pi (Supplementary Fig. 3C). All experiments were performed at least in $n = 3$ independent biological replicates.

**Transmission electron microscopy.** *Lp*-EVs purified as described above, were resuspended in ultrapure water, and visualized by negative staining TEM and cryo-EM. For negative staining TEM, 4 µl of the *Lp*-EVs were incubated for one minute on a glow discharged carbon grid (Agar Scientific) and contrasted with a 2% uranyl acetate solution. Subsequently, the grids were viewed on a Tecnai T12 microscope (Thermo Scientific) and images were obtained using an Eagle camera (Thermo Scientific). For cryo-EM imaging, 3 µl of the *Lp*- EVs (correspond roughly to 8 × $10^5$) were placed on a glow discharged Lacey grid (Agar Scientific) and cryofixed at −180 °C in liquid ethane using an EMGP (Leica). Grids were imaged on a Tecnai F20 using a Falcon II direct detector (Thermo Fisher) operating at 200 kV.

**Flow cytometry experiments.** For flow cytometry experiments, the EVs were labelled with Vybrant DiO or DiD cell dye (Life Technologies) and Syto®RNASelect (Molecular Probes), respectively, according to the manufacturer's instruction. Experiment standardization was done using Megamix-Plus SSC beads (BioCytex). Megamix-Plus SSC is a mix of fluorescent particles of varied diameter (0.16, 0.2, 0.24, 0.5 µm), using SSC a size-related parameter to standardize the setting of the cytometer to study biological vesicles within a constant size region. For analysis, the MACSQuant flow cytometer (Miltenyi Biotec) was used with changing parameters for optimized Megamix-Plus SSC standard separation (Supplementary Fig. 1). However, due to small size of the vesicles, with diameters smaller than the wavelength of the lasers, it makes it quite challenging to analyze and to distinguish them from the background noise. Therefore, for our experiments with Syto®RNA-Select we reduced the SSC voltage and adapted the SSC threshold to reduce the background noise. Thus, mainly large vesicles and EV-agglomerates were detected, which frequently appear during the purification process. Flow cytometry data are submitted to flow-repository public database (https://flowrepository.org/) accession numbers FR-FCM-Z2XL and FR-FCM-Z2XM.

**Floatation assays.** For the floatation assays, we proceeded as previously described[42,43]. Briefly, *Lp*-EVs labelled with Vybrant DiD-dye or not and processed through a size exclusion column, were centrifuged at $100,000 \times g$ for 1 h (MLA-50 rotor). Pellets were resuspended in 1 mL 60% sucrose solution and deposited in the bottom of the tube. 1 mL of 30% sucrose solution and 1 mL of PBS were sequentially loaded on top. Samples were centrifuged at $150,000 \times g$ for 16 h at 4 °C (SW55 rotor). The top fraction was removed, and the 30% sucrose fraction (1 mL) was collected and mixed with 6 mL PBS. Samples were centrifuged at $100,000 \times g$ for 1h30 min (MLA 50 rotor), supernatant containing sucrose was removed and pellets were resuspended in 100 µL PBS prior to further analyses of the particle concentration and their size.

**Nanoparticle tracking analysis.** The *L. pneumophila* vesicles were purified in the same way as described above. The quantification, size characterization, and fluorescence detection of the EVs were then performed on the ZetaView® QUATT (Particle Metrix). For the size and concentration measurements, the 448 nm laser in scatter mode was used; for the fluorescent measurement of DiD-dye positive particles, the 640 nm laser with a 660 nm long-pass filter was used. In all panels, a dot represents the average of 11 measurements corresponding to the 11 frames. For the size, each dot corresponds to the average of the median size which permits to describe the distribution, whereas the mean is biased by the aggregates' extreme values. For the concentration, each dot corresponds to the average number of particles detected taking into account the dilution factor. The normality was tested with D'Agostino–Pearson test which was negative for all panels. The data are paired, and a non-parametric test (Wilcoxon test) was used. All statistical analysis were performed with GraphPad Prism version 9.3.0 for Windows.

**RNA isolation and RNAseq of *Lp*-EVs or THP-1 cells after uptake of *Lp*-EVs.** *Lp*-EV pellets, purified from a 300 ml liquid culture as described above and the corresponding bacterial pellets were resuspended in Qiazol and the RNA extraction was performed following the instruction of the miRNeasy®Mini kit (Qiagen). RNA samples were digested with Turbo DNase (Thermo Scientific) and the size distribution of the *Lp*-EV-RNA was evaluated with a Bioanalyzer (Agilent Technologies). The *Lp*-EV and bacterial RNA (but not *Lp*-EV-RNA) were rRNA depleted using the RiboZero rRNA Removal Kit for Gram-negative bacteria (Illumina) and metal-catalyzed heat-fragmented to a size around 100–200 nts using an RNA fragmentation kit (Ambion). The bacterial RNA was further processed according to the TruSeq stranded mRNA sample preparation guide of Illumina. Before Illumina Hiseq multiplex sequencing, the quantity was determined with a Qubit 2.0 (Invitrogen) and the quality was checked by Bioanalyzer (Agilent Technologies). *Lp*-EV

RNAseq analysis was done $n = 4$ independent biological experiments. To analyze the transcriptome of THP-1 cells after uptake of *Lp*-EVs, THP-1 cells were incubated with *Lp*-EVs purified from wt or the Δ*rsmY*-mutant at an MOI of 10 (according to flow Cytometer measure), or without (non-infected control). After 3 h, the THP-1 cells were pelleted by centrifugation at $500 \times g$, and RNA was extracted as previously described using the miRNeasy®Mini kit (Qiagen). Subsequently, total RNA was treated with Turbo DNase (Thermo Scientific), and RNAs were further processed according to the TruSeq stranded mRNA sample preparation guide of Illumina. Before Illumina Nextseq 500/550 (Illumina) multiplex sequencing, the quantity was determined by Qubit 2.0 (Invitrogen) and the quality was checked by Bioanalyzer (Agilent Technologies).

**RNAseq data analyses.** For analyzing the RNA content of *Lp*-EVs the single-end reads in FASTQ format generated by Illumina sequencing were processed to remove adapters using Cutadapt software version 1.15[81], Sickle (https://github.com/najoshi/sickle) version 1.33 for trimming of the reads (quality threshold Phred score 20). After adapter removal and trimming, all sequence reads shorter than 20 nucleotides were eliminated. Bowtie 1 version 1.2.2[82] was used with a database containing the sequences of ribosomal RNA (rRNA) operons of *L. pneumophila* strain Paris (NC_006368.1) to map all reads corresponding to the rRNA sequences. All mapped reads were removed and the unmapped reads were aligned to the *L. pneumophila* Paris sequence using Bowtie version 2 2.3.4.3[83]. Only uniquely mapped reads were kept, to avoid that a read contributes for the coverage value at different positions. We build, sorted, and indexed the BAM files for each mapping result using the Samtools suite version 1.3 (https://github.com/samtools/samtools). To count reads overlapping each feature, we used featureCounts version 1.6.3 from the Subreads package[84]. The primary option was used to be sure to count only primary alignments. To perform differential analysis of the four replicates for the two conditions (*Lp*-EVs vs bacterial pellet), we used the package Sartools version 1.3.0 including DESeq2 methods. In order to normalize the counts, we used DESeq2 with the « median » option to compute size factors[85]. The resulting tables containing differentially expressed genes with counts, normalized counts, fold changes (ratio of differential expression), and *P*-values were used and visualized using the Artemis analyzing tool version 16.0.17[86]. The transcriptome of THP-1 cells after uptake of *Lp*-EVs were analyzed as described above.

**mi-RNA analyses.** The sRNA highly enriched in the *Lp*-EVs were compared using the database for human mature miRNAs (has-miR) miRBase Release 22.1: October 2018 (http://www.mirbase.org/search.shtml)[87] and similarities with a *E*-value cut off <12 were considered as significant. The search for targets of the predicted human miRNAs (has-miRs) is based on the microT-CDS algorithm (http://diana.imis.athena-innovation.gr/DianaTools/index.php?r=microT_CDS/index)[88].

**Western blot and interferon IFN-β quantification by ELISA.** THP-1 cells were infected (or not) with the wt or the Δ*letA*, Δ*rsmY* or Δ*dotA* strains as described above. At 1, 3, 6, 8, and 24 h post-infection, the THP-1 cells were harvested by centrifugation. For *Lp*-EV uptake, the *Lp*-EVs were incubated with THP-1 cells for 3 h before harvesting the cells by centrifugation. While the supernatant was kept in −80 °C or directly used for IFN-β quantification, the pellet was resuspended in RIPA buffer (20 mM Tris-HCl pH7.5, 150 mM NaCl, 1 mM EDTA, 1% NP-40, 1% Na-deoxycholate, 0.1% SDS, 5% glycerol) + protease inhibitor (Thermo Scientific) and phosphatase and protease inhibitor (Thermo Scientific) for total protein extraction by sonication. After treatment with benzonase (Sigma) and centrifugation, soluble total protein was quantified and an equal amount from each condition was spiked with loading buffer (4xLB: 200 mM Tris, pH6.8, 8% SDSD, 40% glycerol, 400 mM DTT, 0.01% bromphenol blue), denatured at 80 °C for 10 min and loaded on Criterion TGX Stain-free (4–15%) or Mini-Protean TGX Stain-free (AnykD) gels (Biorad). Separated proteins were transferred on 0.2 µM PVDF membranes using the TransBlot Turbo from Biorad. All antibodies used were from Cell Signalling Technology, except IFN-β (NovusBio) and RhoGDI (Santa Cruz Biotechnology) at a dilution of 1:1000. For detection, WesternBright Sirius HRP Substrate (Diagomics) was used and for stripping the WB Strip-it Buffer (Diagomics). Immunoblots were revealed with the G:Box system (Syngene) and the GeneSnap and GeneTools (Syngene) for analyses and quantification. Intensity values at the different time points were normalized against the equivalent bands of the loading control (RhoGDI) and the corresponding non-infected control experiments. The interferon amount within the supernatant (see above) was quantified with the Human IFN-β DuoSet ELISA Kit (R&D Systems) following the manufacturer's instruction. To analyze the impact of TLR agonist, THP-1 cells were pre-treated or not with purified *Lp*-EV isolated from *L. pneumophila* wt or from the *L. pneumophila* Δ*rsmY* for 3 h as described above. Subsequently, the TLR agonists (Human TLR1-9 Agonist Kit, Invivogen) were added as specified by the manufacturer. After 20 h, the cells were centrifuged (500 × *g*, 5 min) and the IFN-b concentration in the supernatants was analyzed using an IFN-ELISA. Protein levels and IFN-β levels shown are a mean of at least $n = 3$ independent biological replicates using a two-way ANOVA for statistical analysis with $p < 0.05$ significant (*), $p < 0.01$ very significant (**), and $p < 0.001$ extremely statistically significant (***).

**RNA:RNA interaction and RNA transfection experiments**. Template DNA for the T7 RNA polymerase was amplified from wt *L. pneumophila* Paris DNA or THP-1 DNA using the corresponding T7-primers indicated in Supplementary Table 1. In vitro RNA was transcribed form the PCR fragments using the T7-MEGAscript® Kit (Life technologies). For the synthesis of in vitro sRNA of RsmY and tRNA-Phe, 1.5 mM CTP-11-Biotin (Roche) was added to the reaction mix for detection. RNA was purified by using mRNeasy®Mini kit (Qiagen) purification protocol. In a 10 µl interaction assay (140 mM KCl, 25 mM NaCL, 5 mM MgCl₂, 50 mM Tris-HCl, pH7.0), 100 µM of labelled sRNAs or was combined with 300 µM of purified in vitro transcribed RNA of *ddx58*, *crel* or *irak1*, respectively, denatured at 70°V for 5 min and incubated at 37° for 30 min. Loading buffer was added and the samples were separated under non-denaturing conditions on a 6% a Tris-Polyacrylamide gel and blotted to BrightStar®-Plus transfer membranes (Ambion). Membranes were blocked in PBS buffer containing 0.1% Tween-20 and 1% ECL blocking agent (GE Healthcare) for 2 h at RT and, incubated for 1 h in the same buffer including mouse anti-biotin antibody (Invitrogen) at a dilution of 1:2000. After washing and binding of the secondary antibody (anti-mouse IgG-HRP, Cell Signaling) at a dilution of 1:5000, the RNA-bands were visualized with *WesternBright Sirius HRP Substrate (Diagomics)* and imaged with a G:Box (Syngene). Each experiment was done in three independent replicates. RNA transfection was done with in vitro transcribed RNA non-biotinylated RsmY, tRNA-Phe or different ctrlRNAs (either a random, non-specific region of the *L. pneumophila* genome or the complementary sequences of RsmY or tRNA-Phe, Supplementary Table 1) were transfected into THP-1 cells following the protocol of Amaxa 4D-Nucleofector X Unit Transfection using the SG Cell line Kit (Lonza). Transfected cells were pelleted after 3 h and protein extraction and immunoblot analysis were performed as above. Alternatively, THP-1 cells were transfected with in vitro transcribed RNAs using the Lipofectamine RNAiMax (Invitrogen) protocol, and cells were harvested after 24 h for further analysis by western blot or IFNβ ELISA. The results of both approaches were combined, as the result was highly similar. For IFN-β quantification, the supernatant was kept at −80 °C or directly used in the ELISA as described above. Results are from at least three independent biological replicates and for statistical analysis, a two-way ANOVA was performed with values of $p < 0.05$ significant (*), $p < 0.01$ very significant (**), and $p < 0.001$ extremely statistically significant (***).

**Bacterial infection assays**. For bacterial infection, *L. pneumophila* wt was grown until post-exponential growth phase (OD4.2) in broth culture (BYE). Bacteria were resuspended in the adequate culture medium for the respective cell line (see above). Non-differentiated THP-1 were infected with an MOI = 1 as described before[80]. To test the influence of cytokines, human interferon β 1A (Sigma) at concentration of 25, 50, and 100 IU/ml or human interleukin-1b (Sigma) at 20 ng/ml were added to the THP-1 cell cultures and incubated for 24 h before infection with *L. pneumophila*. To analyze the impact of *Lp*-EVs on bacterial infection, purified *Lp*-EVs from *L. pneumophila* wt or the *ΔrsmY* -mutant strain were added to the THP-1 cells at MOI of 10 (according to flow cytometer measure) for 3 h before bacterial infection. Two hours after the bacterial infection the THP-1 cells were carefully washed three times with PBS to remove extracellular bacteria and resuspended in RPMI medium. If mentioned, the same amount of *Lp*-EVs or IFN-β was added as described above and additional time points were taken at 24 h, 48 h, and 72 h. For inhibition experiments, 5 µM IRAK1/4 Inhibitor-1 (Sigma) or 30 µM Ago2-Inhibitor (BCI-137, Sigma) were added to the cell culture medium. The graph represents the cfu/ml at the different time points normalized to T0. SiRNA transfection was performed following the manufacturer's protocol, by using Lipofectamine RNAiMax Reagent (Thermo Fisher Scientific). Silencer select siRNA targeting IRAK-1 (#s323), DDX58/Rig-I (#223614) mRNA, and the corresponding scramble control (SilenceSelect Negative Control siRNA#1) were also purchased from Thermo Fisher Scientific. THP-1 cells were infected with an MOI of 1 and time points were taken at 24 h, 48 h, and 72 h. The graphs represent the % cfu at the different time points. All infection conditions were performed in at least $n = 3$ biological replicates.

**Dual Luciferase assay**. The pmirGLO Dual-Luciferase miRNA vector (Promega) was constructed as shown in Supplementary Fig. 4. The UTR of *ddx58*, *rel*, and *irak1* were amplified and cloned adjacent to the *luc2* gene using MssI/XhoI restriction sites. The respective sRNA was introduced by amplifying the H1 promoter region and the sRNA from the pSuper constructs described before with the H1-sRNA oligo pair (Supplementary Table 1) BamHI restriction sites at the ends, the PCR fragments were cloned into the pmirGLO system. All possible combinations of the vector were transfected into THP-1 cells using the SG Cell Line Kit (Lonza) and luciferase activities were measured with a Synergy2 (BioTek) after 24 h following the manufactures instructions. The resulting ratios of Firefly/Renilla activities were normalized to the corresponding values from the control (ctrlRNA) experiments. When indicated, Ago2-inhibitor (BCI-137, Sigma) was added at a final concentration of 30 µM to the transfected THP-1 cells, and the luciferases activities were measured after 24 h. The data shown in the graph are the mean of four independent biological repeats, each in duplicates, using a two-way ANOVA for statistical analysis with values of $p < 0.05$ significant (*), $p < 0.01$ very significant (**), and $p < 0.001$ extremely statistically significant (***).

**Ago2-CLIPseq**. For Ago2-iummunoprecipitation (Ago2-IP), about $1 \times 10^8$ non-differentiated THP-1 cells were infected with *L. pneumophila* at MOI 50. After 8 h cells were washed, resuspended in PBS, and cross-linked by exposure to 400 mJ/cm² of 254 nm UV irradiation (Bio-Link 254 nm Vilber Lourmat) for two times on a metal block cooled to 4 °C. Cells were pelleted at $500 \times g$ (4 °C), washed in cold 1× PBS. After centrifugation, the pellet was resuspended in 1 ml lysis buffer (50 mM HEPES pH 7.5, 150 mM NaCl, 0.5% Nonidet P-40, 2 mM EDTA, 1 mM DTT + protease inhibitor (Thermo Scientific) + 10 U/ml RNase Out (Thermo Scientific), sonicated and cleared by centrifugation. Dynabeads Protein G (Thermo Scientific) were washed and pre-blocked in lysis buffer + 0.1% BSA and incubated with anti-human Argonaute 2 antibody (Cell Signaling) at a dilution of 1:50 or normal rabbit IgG (Cell Signaling) at a dilution of 1:50 for 2 h at 4 °C; before use, they were washed two times and resuspended in lysis buffer. The cell lysate was pre-cleared with non-antibody bound dynabeads for 30 min at RT. From this supernatant, 200 µl was put aside for total control RNA extraction. The remaining 800 µl was divided in two samples and each was incubated overnight on a rotating platform at 4 °C with Ago2- or IgG-beads, respectively. The beads were washed three-times with ice cold lysis buffer and additional three-times in lysis buffer + 350 mM NaCl. To release the RNA, beads were resuspended in 100 mM Tris-HCl pH 7.5, 50 mM NaCl, 1% SDS, 10 mM EDTA + 100 µg Proteinase K (Thermo Scientific) and incubated for 30 min at 55 °C. RNA was extracted using miRNAeasy Mini Kit (Qiagen). The remaining DNA was digested with Turbo DNAse (Thermo Scientific), and the purified RNA was cleaned with miRNA columns and analyzed by QuBit and Bioanalyzer. RNA libraries of total RNA, Ago2-IP, and IgG-IP were prepared following the NEBNext Small RNA Library Prep Set for Illumina and sequenced with the NextSeq 500/550 v2 (Illumina).

**Analyses of Ago2 CLIP-Seq data**. Single end reads in FASTQ format generated by Illumina sequencing were processed to remove adapters using Cutadapt software version 1.15[73] and Sickle (https://github.com/najoshi/sickle) for trimming of the reads (quality threshold Phred score 20). For each remaining read, the 3'end nucleotide was removed as it is always mismatching during the alignment to the reference sequence. After adapter removal and trimming, all sequence reads shorter than 12 nucleotides were eliminated. We also removed reads longer than 36 nucleotides, to focus on miRNA enrichment. Good quality reads were aligned to a database containing the human genome sequence build GRCh38 plus the *L. pneumophila* strain Paris genome sequence (NC_006368.1). To perform an accurate mapping with this short-read size, we used Bowtie2 version 2.3.4.3 with the following parameters: --end-to-end -D 20 -R 3 -N 0 -L 11 -i S,1,0.50. For peak detection in CLIP samples, we used CLAM peak caller version 1.2.0[89] using only unique mapped reads and IgG samples as control, with a minimal coverage of 10 reads in CLIP samples (--min-clip-cov option). We set the minimal fold change threshold (CLIP vs IgG control) to 3 (--fold-change option).

**Fluorescence in situ hybridization (FISH) and analysis**. hMDMs were incubated with Vybrant DiD-labelled purified *Lp*-EVs as described above ($3 \times 10^4$ cells/well, MOI 10). After 5 h of incubation, cells were washed three-times with PBS and fixed in 4% PFA/PBS for 10 min at RT. After washing and neutralization in 50 mM NH₄Cl/PBS, 5 min at RT, the cells were permeabilised in 0.075% Saponin/1% BSA/PBS for 30 min at RT and washed three-times with PBS. hMDM were hybridized over night at 37 °C with two RsmY-FISH probes (Supplementary Table 1) at a concentration of 150 nM in the buffer: 2×SSC (0.3 M NaCl, 30 mM 3NaCitrate, pH7.0) + 10% formamide + 0.1% yeast tRNA (=1 mg/ml) + 1% RNase inhibitor + 10% dextran sulfate + 0.02% BSA. Subsequently, the cells were washed 2-times at 37 °C for 30 min in wash buffer (0.2×SSC + 10% formamide + 0.1% SDS), rinsed with PBS and incubated with Hoechst 33342 (Life Technologies) for 15 min. After two additional washing steps in PBS, image acquisition of multiple fields per well was performed on an automated confocal microscope (OPERA Phenix, Perkin Elmer) using the 60x objective, excitation lasers at 405, 488, 561, and 640 nm, and emission filters at 450, 540, 600, and 690 nm. Images were transferred to the Columbus Image Data Storage and Analysis System version 2.9.1 (Perkin Elmer) for evaluation. High-Content Analyses (HCA) were performed by using in-house developed HCA scripts in Columbus, shared upon request. Briefly, Hoechst 33342 signal was used to find the nuclei of cells and then segment nuclei and cytoplasm in each cell. Then, DiD signal was used to find *Lp*-EVs in the cytoplasm of hMDMs, and the fluorescence of FISH probes was used to analyze which *Lp*-EVs were positive for RsmY RNA.

**Lp-EV content release characterization: HiBiT/LgBiT secretion assay**. *Lp*-EVs of bacteria expressing GroEL-HiBiT (pMMB) were purified as described before. The Luciferase activity of the total input was quantified using the Nano-Glo (R) HiBiT Lytic detection System (Promega) with a Synergy2 (BioTek) following the manufactures instructions. LgBiT was cloned into the pCMV-Tag 2 vector (Agilent) and the vector was transfected into THP-1 cells using the SG Cell Line Kit (Lonza). After 24 h, transfected LgBiT expressing THP-1 cells were incubated with purified *Lp*-EVs containing the HiBiT-tagged GroEL protein for 3 h, before the luciferase activity was measured using the Nano-Glo (R) Live Cell Assay System (Promega) with a Synergy2 (BioTek) according to the manufactures instructions. The values were normalized relative to the total input of GroEL-HiBiT (Lytic

detection) in %. As negative control, *Lp*-EVs were purified from *L. pneumophila* without GroEL-HiBiT vector and comparable amounts of *Lp*-EVs were added in the assay. When indicated, 10 μM of cyochalasin D (Sigma) or 200 nM of Bafilomycin A1 (Sigma) respectively, were added to THP-1 cell cultures and incubated for 30 min before infection. All experiments were performed in $n > 5$ biological replicates

**Live cell imaging and analysis**. U2OS-Sec61β cells were infected with Vybrant DiD-labelled *L. pneumophila* or incubated with *Lp*-EVs as described above. For mitochondrial or acidic organelle labelling, the cells were stained prior to the infection with MitoTracker Red FM or LysoTracker Red DND-99 (Life Technologies) for 30 min and then washed 3-times with assay medium; nuclear staining was performed with Hoechst 33342 (Life Technologies) for 15 min. Rab5 and Rab7 labelling via CellLight was performed according to the manufacturer's instructions for at least 16 h. In the *Lp*-EV uptake experiments, Cytochalasin D (Sigma) or Dynasore (Sigma) was added to the cell culture medium at a final concentration of 1 and 80 μM, respectively. Live image acquisition of multiple fields per well was performed over time at 37 °C (5% $CO_2$) on an automated confocal microscope (OPERA Phenix, Perkin Elmer) using the 60× objective, excitation lasers at 405, 488, 561, and 640 nm, and emission filters at 450, 540, 600, and 690 nm. For the analysis, the Columbus Image Data Storage and Analysis System (Perkin Elmer) was used. HCA was performed as described above. The DiD signal was used to detect *Lp*-EVs in the cytoplasm of U2OS-Sec61β and the fluorescence of Mitotracker, Lysotracker, Sec61β-GFP, Rab5, and Rab7-CellLight were measured inside each *Lp*-EV in order to analyze whether they co-localized with mitochondria, lysosomes, ER, Rab5, or Rab7 endosomes, respectively. For Z-stack time-lapse 3D videos, U2OS-Sec61β cells were plated in IBIDI dishes and imaged in an Ultraview-VOX spinning-disk confocal microscope (Perkin Elmer) using a 63X objective, excitation lasers at 405, 488, 561, and 640 nm, and emission filters at 450, 540, 600, and 690 nm. Time-lapse 3D videos were analyzed and edited using Volocity software version 6.5.1 (Perkin Elmer).

**Reporting summary**. Further information on research design is available in the Nature Research Reporting Summary linked to this article.

## Data availability
The sequence reads as well as the coverage files of the RNAseq libraries of the *Lp*-EV analyses have been deposited in the NCBI Gene Expression Omnibus (GEO) database[90]. Accession number GSE159109. The RNAseq data from THP-1 cells have been deposited in the NCBI Gene Expression Omnibus (GEO) database. Accession number GSE190376. Flow cytometry data are submitted to flow-repository public database (https://flowrepository.org/) accession numbers FR-FCM-Z2XL and FR-FCM-Z2XM. Source data are provided with this paper.

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

## Acknowledgements

Work in the C.B. laboratory was financed by the grant n° ANR-20-CE15-0021-01-BIOEV, the Institut Pasteur, the Fondation pour la Recherche Médicale (FRM) grant N° EQU201903007847 and the grant n°ANR-10-LABX-62-IBEID. We thank Monica Rolando for critically reading of the manuscript and helpful discussions and Nathalie Aulner and Anne Danckaert, Photonic BioImaging (PBI) UTechS for support. For electron microscopy experiments M. Nilges and the Equipex CACSICE (n° ANR-11-EQPX-0008) provided funding for the Falcon II direct detector. G.L. lab is funded by ANR-20-CE15-0021-02, ANR-19-CE18-0020-03, and ANR-18-IDEX-0001, IdEx Université de Paris. S.B. is a recipient of a PhD fellowship from PSL University. The funders, other than the authors, did not play any role in the study or in the preparation of the article or decision to publish.

## Author contributions

T.S., P.E., S.B., and G.L. performed the experiments, C.R. performed the bioinformatics analyses, GPA performed the electron and cryoTEM experiments, T.S., P.E., and C.B. designed the experiments, T.S. and C.B. wrote the article, C.B. supervised the work and acquired funds.

## Competing interests

The authors declare no competing interests.
