## [Peer Review File · Nature Communications]

Translocated *Legionella pneumophila* small RNAs mimic eukaryotic microRNAs targeting the host immune responseREVIEWER COMMENTS

Reviewer #1 (Remarks to the Author):

In this manuscript, Sahr et al. claim that *Legionella pneumophila* translocate small RNAs via extracellular vesicles to eukaryotic host cells and that these small bacterial RNAs reduce several factors of the RIG-I signalling pathway by a microRNA-like way of action. Trans-kingdom RNA-signalling, specifically sRNAs acting as microRNAs, is of very high relevance for the field of infection biology and has been discussed for several years. However, there are important concerns that need to be addressed, mainly regarding (i) the characterization of extracellular vesicles, (ii) the proof of miRNA-like activity of sRNAs, and finally (iii) the exact interference with the RIG-I pathway.

(i) Concerns regarding the characterization of extracellular vesicles:

- EV purification was performed from late post-exponential phase (OD4.2) (M&M line 518). It has been shown that the growth phase has significant impact on the properties of bacterial EVs (Tashiro Y et al.; Variation of physicochemical properties and cell association activity of membrane vesicles with growth phase in *Pseudomonas aeruginosa*; Appl Environ Microbiol 76(2010), McCaig WD et al.; Production of outer membrane vesicles and outer membrane tubes by *Francisella novicida*; J Bacteriol, 195 (2013)) and the cultivation to extremely late stationary phases would lead to bacterial cell lysis as well as the contamination with the broken membranes and cytoplasmic proteins (Klimentová J & Stulík Jiri, Methods of isolation and purification of outer membrane vesicles from negative bacteria; Microbiological Research 2015).
- The authors used Exosome Spin Columns for purification of EVs after staining with Vybrant DiD and Syto RNA select. It is known that these dyes are causing staining artefacts and larger aggregates even without EVs (Morales-Kastresana, A. et al. Labeling Extracellular Vesicles for Nanoscale Flow Cytometry. Sci Rep 7, 1878 (2017). <https://doi.org/10.1038/s41598-017-01731-2>). They should show that free dye and aggregates are successfully removed from the sample, as they would give signals in flow cytometry. How have the authors proven that RNA select is not binding to RNA on the surface of EVs?
- Moreover, RNA-FISH in immunogold EM would strengthen the claim that small RNAs are transported in the EVs.
- The authors observed differences in their EV population (spherical structures with single and double membranes and tube-shaped vesicles). These different populations should be separated by size-exclusion chromatography or density gradient centrifugation to prove which of the population harbors the RNA content as it has been shown for other gram-negative bacteria that they can contain different protein compositions (McCaig WD et al.; Production of outer membrane vesicles and outer membrane tubes by *Francisella novicida*; J Bacteriol, 195 (2013)).
- The authors used conventional flow cytometry and tried to calibrate with beads (0.1-0.5 μm). Additionally, they stained the EVs with a lipophilic dye and an RNA dye (single dyed!). Conventional FACS does not cover the small size range needed for EV analyses and bead standards do not reflect the physical properties of EVs and are not showing the exact size of the EVs. Another state-of-the-art EV characterization method such as NanoFCM should be applied to characterize the size and differences in the EV preparation observed in CryoEM. Moreover, the authors could not reproduce the EV sizes they found in electron microscopy in flow cytometry (Fig. S1D).
- The authors used conventional flow cytometry to quantify the amount of EVs for stimulation experiments. This is not a valid technique for EV quantification and the numbers are misleading as a conventional flow cytometer does not feature enough SSC resolution to be used for standardized EV counts. The authors should use NTA, tunable resistance pulse sensing or Nano flow cytometry.
- Figure S1 A-D: The authors are talking about Lp-MVs here, did they measure something different as in the rest of the manuscript?

(ii) Concerns regarding the proof of miRNA-like activity of sRNAs:

- Fig. 4: The postulated RsmY and tRNA-Phe target-sites in the mRNA 3'UTRs need to be presented in the figure. In addition, it is common practice to introduce point mutations into the 3' UTR target sites in order to abrogate base pairing between the luciferase mRNA-construct and the micro (or in this case small) RNA. Without this experiment the proof that RsmY and tRNA-Phe act through a

microRNA-like 3' UTR targeting mechanism is not provided. In addition to 3' UTR target-site mutations, compensatory base pair mutations should be inserted into RsmY and tRNA-Phe in order to restore the regulation of mutated luciferase constructs.

- Although the authors claim that RsmY and tRNA-Phe adopt microRNA-like functions, no data are provided regarding the incorporation of these two RNAs into the host microRNA machinery. In addition to Northern blot analysis of RsmY and tRNA-Phe processing fragments in the microRNA size range, Ago2-CLIP needs to be performed to validate the association of RsmY / tRNA-Phe or their derived fragments with the microRNA machinery.
- The abundance of RsmY in the host cell cytoplasm and nucleus during a physiological infection setting remains unclear. What are the RsmY copy numbers in cytosolic versus nuclear fractions after stimulation with a physiological MOI / vesicle amount? Does RsmY omit the nuclear steps of the microRNA processing machinery?
- The authors show a co-localization of lipid dye with the lysosomes (Figure 2A/B). Are the lysosomes acidified? Do they degrade the RNA transported on or in the EVs? Can the bacterial RNA escape the lysosomes and can be found in the cytosol? Cellular fractionation experiments with subsequent isolation of RNA should be performed to prove the presence of sRNA in the host cell cytoplasm. Besides, the tracking by immunofluorescence should be prolonged and include a late timepoint.
- The authors describe that they used different protocols for RNA preparation from bacteria and EVs for sequencing. EV-RNA was isolated using miRNeasy Mini Kit and RNA was DNase digested. They do not comment on the isolation protocol for bacterial (cellular) RNA, but this RNA was additionally rRNA depleted and fragmented. This additional steps can give a bias in sample preparation and does not support the comparability of sequenced RNA.

(iii) Concerns regarding the exact interference with the RIG-I pathway:

- The RsmY-dependent effects on the host immune response are not convincingly presented. In addition to RIG-I, TBK1 and IRF3/7 Western blots an RNA-Seq analysis of gene expression changes in WT versus Δ RsmY Legionella infected / vesicle stimulated cells would be required to portrait the global impact of Legionella-encoded microRNA-like molecules on host gene expression. In addition, interferon ELISAs are required to document the postulated effect of RsmY on RIG-I-induced host responses.
- The authors do not provide proof for RIG-I-dependence of the effects of Legionella RsmY deletion on host immune signalling. Experiments with RIG-I / MAVS deficient cells are required to exclude the possibility that the presented effects depend on additional pattern recognition receptor pathways.
- The authors state that RsmY in EVs is regulating cRel, but they do not show a regulation of cREL on protein level by Legionella infection of EV treatment of the macrophages.
- The authors used the bone osteosarcoma epithelial cell line U2OS cells for parts of their experiments. They should reproduce the experiments in a more physiological cell culture model for Legionella pneumophila infection as osteosarcoma cells might respond to differently to a bacterial infection and the stimulation of PRRs with PAMPs present on or in Legionella EVs. In addition, THP-1 cells were used. These cells are monocytes, unless they are differentiated with phorbol 12-myristate 13-acetate in macrophages. Did the authors use monocyte- or macrophages-like cells? And if so, how have they been differentiated?

Reviewer #2 (Remarks to the Author):

In this work Sahr et al have identified two Legionella sRNA species which are proposed to mimic host miRs to control innate immune responses. Although the original observation is interesting, the work appears rather under-developed, and there are limited data on the physiological implication of the observation, including in primary cells.

1. The manuscript critically lacks data on the impact of the identified mechanism on host innate immune responses and anti-microbial defense. The data presented in Fig 4D-E are based on purified EVs and in THP1 cells, and show only a modest effect. As a minimum, the authors should compare IFN β (and ISG) responses to infection with wt and KO bacteria in THP1 cells and primary

macrophages.

2. Along the same lines, the authors should more globally characterize how the sRNAs affect host cell gene expression through RNAseq analysis.

3. To make sure that the observed effects of bacterial EVs are in fact due to targeting of RIG-I, the authors should generate RIG-I KO THP1 cells and demonstrate that the modulatory effect of the EVs is lost.

4. The work would gain significantly, if induction of type I IFN by a panel of synthetic agonists for TLRs and cytosolic PRRs were evaluated in cells treated with relevante EVs.

5. The functional data do generally not show a very large effect of the EVs/sRNAs (e.g. in Fig 4). Therefore, for the work to have impact, it is essential that the authors show data on the effect of the proposed immunomodulatory RNAs in bacterial growth.

6. The functional data in Fig 4, should be confirmed in primary cells, and ideally also in mice (if the sRNAs also target RIG-I and IRAK1 in mice).

REVIEWER COMMENTS

Reviewer #1 (Remarks to the Author):

In this manuscript, Sahr et al. claim that *Legionella pneumophila* translocate small RNAs via extracellular vesicles to eukaryotic host cells and that these small bacterial RNAs reduce several factors of the RIG-I signalling pathway by a microRNA-like way of action. Trans-kingdom RNA-signalling, specifically sRNAs acting as microRNAs, is of very high relevance for the field of infection biology and has been discussed for several years. However, there are important concerns that need to be addressed, mainly regarding (i) the characterization of extracellular vesicles, (ii) the proof of miRNA-like activity of sRNAs, and finally (iii) the exact interference with the RIG-I pathway.

We thank the reviewers for his/her pertinent comments. We have undertaken many of the suggested experiments, which have indeed improved the manuscript. However, several questions he/she is raising are burning questions in the field, but nobody has been able to answer them yet. We would be thrilled if we could answer all these questions but believe that many laboratories will have to continue research in EVs for years before we will be able to answer everything. We have taken the concerns raised seriously and have tried to do as many of the suggested/needed experiments as possible in addition during a very challenging time (Covid lockdown in France and constant work restrictions at Institut Pasteur in parallel to not being priority of the platforms as priority was given to Covid research) which have improved our manuscript. We hope that the reviewer is satisfied with the many additional experiments conducted that all further support our hypothesis that RsmY acts in a miRNA like manner in the host cell.

(i) Concerns regarding the characterization of extracellular vesicles:

- EV purification was performed from late post-exponential phase (OD4.2) (M&M line 518). It has been shown that the growth phase has significant impact on the properties of bacterial EVs (Tashiro Y et al.; Variation of physiochemical properties and cell association activity of membrane vesicles with growth phase in *Pseudomonas aeruginosa*; Appl Environ Microbiol 76(2010), McCaig WD et al.; Production of outer membrane vesicles and outer membrane tubes by *Francisella novicida*; J Bacteriol, 195 (2013)) and the cultivation to extremely late stationary phases would lead to bacterial cell lysis as well as the contamination with the broken membranes and cytoplasmic proteins (Klimentová J & Stulík Jiri, Methods of isolation and purification of outer membrane vesicles from negative bacteria; Microbiological Research 2015).

We agree that the growth phase has significant impact on bacterial EVs and specifically for *L. pneumophila* the growth phase is very important as it exhibits a biphasic life cycle where virulence is expressed only in post exponential (PE) growth

phase (Molowsky and Swanson, Mol Micro, 2004). Thus, we choose an OD of 4.2 as this is PE growth for *L. pneumophila* and not an extremely late stationary phase. Furthermore, when one looks at the TEM images which were taken from EVs isolated from bacteria grown to OD 4.2 one can clearly see that there is no cell debris. We never observed bacterial lysis, which is expected as OD 4.2 is relatively short after exponential (E)-phase time points. We agree that exponentially grown vesicles might be different, but as we were interested in virulence impact of these vesicles PE phase bacteria were chosen. Indeed, RsmY is highly expressed only in PE phase grown bacteria, thus we are not expecting to find it in EVs purified from bacteria in E phase.

- The authors used Exosome Spin Columns for purification of EVs after staining with Vybrant DiD and Syto RNA select. It is known that these dyes are causing staining artefacts and larger aggregates even without EVs (Morales-Kastresana, A. et al. Labeling Extracellular Vesicles for Nanoscale Flow Cytometry. Sci Rep 7, 1878 (2017). <https://doi.org/10.1038/s41598-017-01731-2>). They should show that free dye and aggregates are successfully removed from the sample, as they would give signals in flow cytometry.

Indeed, we have shown that dye aggregates have successfully been removed. The reviewer might have overseen these results shown in Figure S1B, which shows a control experiment with dye alone. In addition, we now used also a sucrose flotation assay to formally rule out the possibility of contaminants during EV isolation and further characterization. This is now reported in Figure 1

How have the authors proven that RNA select is not binding to RNA on the surface of EVs?

There must have been an oversight, as it was mentioned in the M&M that the *Lp*-EVs have been treated with RNase during purification to degrade extravesicular RNAs.

Lines 689-691 it reads... The supernatant was treated with RNaseA/T1 (Thermo Scientific) at a final concentration of 2µg/ml RNaseA for 1h at 37°C followed by centrifugation at 150.000xg for 2h at 4°C to pellet the *Lp*-EVs. The *Lp*-EV-pellet was washed, re-centrifuged and resuspended in PBS. -

- Moreover, RNA-FISH in immunogold EM would strengthen the claim that small RNAs are transported in the EVs.

We agree with the reviewer that this would be a very appealing method to apply, although we do not see which additional information would be gleaned from it what was not already shown by RsmY FISH. However, we have contacted Pierron Gerard who is specialist for this technique (Soquere and Perron, Methods Mol Biol. 2015;1262:105-18.). After discussing with him this request he also told us that such an experiment would not add any additional information as we have already provided the RsmY FISH results in the paper. Furthermore, it is very likely that it is not possible to apply this technique to RNAs in EVs as a high concentration of RNA in the vesicles is required to be detected by this technique. He successfully applied it for example for HSV1 genome detection in intra- and extra-cytoplasmic virions, which

contain a high concentration of RNA, however, this is not the case for bacterial RNAs in EVs. Thus, as he does not think that this technique is feasible for this question and in addition it would not bring any additional information than FISH which we provided already we did not follow it up further.

- The authors observed differences in their EV population (spherical structures with single and double membranes and tube-shaped vesicles). These different populations should be separated by size-exclusion chromatography or density gradient centrifugation to prove which of the population harbors the RNA content as it has been shown for other gram-negative bacteria that they can contain different protein compositions (McCaig WD et al.; Production of outer membrane vesicles and outer membrane tubes by *Francisella novicida*; J Bacteriol, 195 (2013)).

We agree that these two different EV populations might contain different RNAs and or protein compositions. However, it is not the scope of the paper to define the exact content of each of the EV populations, but to analyse the impact of the RNA present in the EVs on the host cell. We are continuing to analyze the EVs and are planning to define the protein content of the EVs and to try to separate them, but this will be a future analysis and is out of the scope of this article which is already quite long with a very high number of experiments.

- The authors used conventional flow cytometry and tried to calibrate with beads (0.1-0.5 μm). Additionally, they stained the EVs with a lipophilic dye and an RNA dye (single dyed!). Conventional FACS does not cover the small size range needed for EV analyses and bead standards do not reflect the physical properties of EVs and are not showing the exact size of the EVs. Another state-of-the-art EV characterization method such as NanoFCM should be applied to characterize the size and differences in the EV preparation observed in CryoEM. Moreover, the authors could not reproduce the EV sizes they found in electron microscopy in flow cytometry (Fig. S1D).

Thank you for pointing this out, this was a mistake in the calculation which can be clearly seen with respect to the indicated size measure in the figure. Indeed, the vesicle size should have been indicated as 20-200nm in diameter and not 20-100nm as originally stated. This has been corrected

Lines 131-133 it reads now... As shown in **Figure 1A**, the *Lp*-EVs are mostly spherical structures ranging from around 20-200nm in diameter, but also tube-shaped vesicles are present.

As requested, we have also undertaken an extensive Nanoparticle tracking analyses using ZetaView that further characterizes the *Lp*-EV in depth (see below)

- The authors used conventional flow cytometry to quantify the amount of EVs for stimulation experiments. This is not a valid technique for EV quantification and the numbers are misleading as a conventional flow cytometer does not feature enough

SSC resolution to be used for standardized EV counts. The authors should use NTA, tunable resistance pulse sensing or Nano flow cytometry.

As requested by the reviewer we did new experiments and performed the quantification, size characterization and fluorescence detection of the EVs on a ZetaView® QUATT (Particle Metrix) equipment. As we do not have neither the equipment nor the expertise to analyse EVs by NTA, we have contacted our collaborators Gregory Lavieu and Sheryl Bui who undertook these analyses. Thus, we have added them as authors on the revised manuscript. To exclude the presence of aggregates we also added a sucrose floatation step in our *Lp*-EV-isolation procedure. These new results showed that using conventional flow cytometry we had underestimated the *Lp*-EV concentration in our purification, as it was estimated as 10^{11} by NTA analyses compared to 10^8 with conventional flow cytometry. This is now corrected in the text. However, the new experiments confirmed our previous size estimation of the EVs. These new results are added in the text, as a new figure (Figure 1D, E, F) and in M&M. The previous figure 1D, E, F were moved to Figure S1 as Figure S1A, C, D.

Lines 139 -159 it reads now.....” To quantify the amount of purified *Lp*-EVs and estimate their size nanoparticle tracking analyses (NTA) was used after staining the putative *Lp*-EVs with Vybrant™ DiD to stain the membranes of the EVs. Although we used a size filtration column to remove excess of free dye, the presence of aggregated dye within our samples, or the presence of other large protein/lipid aggregates emanating from *L. pneumophila* could not be completely ruled out. Thus, we added a sucrose floatation step to the isolation procedure. We first analyzed the size distribution and the number of particles pre- and post-floatation, through light scattering mode revealing that, the number of particles was moderately decreased after floatation likely due to the three additional ultracentrifugation steps required for the floatation procedure (**Figure 1D, E**). Particles in both samples showed a median size of ~ 130 nm (**Figure 1E**). In addition, we compared particles size and concentration when measured in fluorescence mode to analyze *Lp*-particles labeled with the red-lipophilic dye. Size distribution was similar (**Figure 1 F**, right panel), and ~ 85% of the particles were positive for the membrane dye, consistently with our FACS data and previous studies. These results suggest that most of the *Lp*-derived nanoparticles considered in our study are indeed *Lp*-EVs.

However, we could not use the NTA to measure the percentage of *Lp*-EVs positive for the RNA dye as the set-up of the machine was not compatible with the fluorescent properties of the RNA-dye. Thus, we used conventional flow cytometry analyses, as the NTA results with respect to the size distribution and the quantification were comparable to determine the percentage of *Lp*-EVs that contains RNA molecules (**Figure S1A**).

Lines 732-753 it reads now.....” **Floatation assays.** For the floatation assays, we proceeded as previously described^{68,69}. Briefly, *Lp*-EVs labeled with Vybrant DiD-dye or not and processed through a size exclusion column, were centrifuged at 100,000g for 1 hour (MLA-50 rotor). Pellets were resuspended in 1mL 60% sucrose and deposited in the bottom of the tube. 1mL of 30% sucrose solution and 1mL of PBS were sequentially loaded on top. Samples were centrifuged at 150,000 × *g* for 16 h at

4 °C (SW55 rotor). The top fraction was removed, and the 30% sucrose fraction (1mL) was collected and mixed with 6mL PBS. Samples were centrifuged at 100,000 × *g* for 1h30 min (MLA 50 rotor), supernatant containing sucrose was removed and pellets were resuspended in 100µL PBS prior to further analyses of the particle concentration and their size.

Nanoparticle Tracking Analysis. The quantification, size characterization and fluorescence detection of the EVs were performed on the ZetaView® QUATT (Particle Metrix). For the size and concentration measurements, the 448nm laser in scatter mode was used; for the fluorescent measurement of DiD-dye positive particles, the 640nm laser with a 660nm long-pass filter was used. In all panels, a dot represents the average of 11 measurements corresponding to the 11 frames. For the size, each dot corresponds to the average of the median size which permits to describe the distribution, whereas the mean is biased by the aggregates' extreme values. For the concentration, each dot corresponds to the average number of particles detected taking into account the dilution factor. The normality was tested with D'Agostino-Pearson test which was negative for all panels. The data are paired, and a non-parametric test (Wilcoxon test) was used. All statistical analysis were performed with GraphPad Prism version 9.1.1 for MasOS.

• Figure S1 A-D: The authors are talking about Lp-MVs here, did they measure something different as in the rest of the manuscript?

We apologize for this mistake, as the working name was MVs we had forgotten to change it. We have now corrected it, of course these are also *Lp*-EVs.

(ii) Concerns regarding the proof of miRNA-like activity of sRNAs:

• Fig. 4: The postulated RsmY and tRNA-Phe target-sites in the mRNA 3'UTRs need to be presented in the figure. In addition, it is common practice to introduce point mutations into the 3' UTR target sites in order to abrogate base pairing between the luciferase mRNA-construct and the micro (or in this case small) RNA. Without this experiment the proof that RsmY and tRNA-Phe act through a microRNA-like 3'UTR targeting mechanism is not provided. In addition to 3' UTR target-site mutations, compensatory base pair mutations should be inserted into RsmY and tRNA-Phe in order to restore the regulation of mutated luciferase constructs.

We understand that the reviewer would like to see mutations in the interacting site, however, it is not known how the sRNA present in the *Lp*-EVs are further processed in the host cell. Thus, we cannot ascertain which part is the biological active one. However, we have used the complementary sequences of RsmY (Figure S4A), as control showing that the anti-sense RsmY sequence has no impact on Luciferase activity and intracellular replication of *L. pneumophila*, which proofs that RsmY activity is not random. To further support our observations, confirm that RsmY activity is miRNA like we undertook two additional experiments:

- First, we analyzed the impact of Ago2 that is part of the RISC complex and necessary for miRNA function on the activity of RsmY on the 3'UTR of *ddx58*. We treated THP-1 cells with the Ago2 inhibitor BCI-137 and conducted the dual

luciferase assay. This showed that indeed, a functional Ago2 protein is necessary for RsmY to repress RIG-I activity. This new experiment is added as Figure 4C and D and is described in M&M and in the text.

Lines 415-431 it reads now... “Argonaute family proteins play a crucial role in RNA induced silencing complex (RISC). Thus, to further analyse if RsmY acts in a miRNA like manner we investigated whether argonaute-2 (Ago2), the only member with catalytic activity and an essential role within the RISC complex to regulate small RNA guided gene silencing processes^{54,55} impacts RsmY and/or tRNA-Phe activity. Indeed, the suppressive effect of RsmY, tRNA-Phe or of *Lp*-EVs on the relative luciferase activity of Luc2 fused to the UTR of *ddx58*/Rig-I or the UTR of *irak1* was significantly reduced when Ago2 inhibitor was added (**Figure 4C, D**). This suggests that the presence of a functional Ago2 is necessary for *Lp*-EV RNAs to interact with *ddx58* and *irak1* UTRs *in cellulo* and show that Ago2 and thus probably RISC-mediated silencing are involved in *ddx58* and *irak1* expression during *L. pneumophila* infection. However, secondary effects of Ago2 inhibition may also influence this result, since endogenous human miRNA can also play a role in the regulation of protein expression⁵⁶. Yet, as transfection of RsmY-RNA did not influence luciferase activity of Luc2 fused to the *irak1*-UTR independently of Ago2 inhibition, the observed impact of tRNA-Phe on IRAK1 cannot solely depend on the effect of endogenous has-miRNA-silencing, but it further suggests that indeed *L. pneumophila* RsmY has a significant impact on protein expression in a miRNA-like and Ago2-dependent manner. “

Secondly, we undertook CLIPseq (immunoprecipitation followed by RNA sequencing) using Ago2 antibodies to analyze whether we find RsmY bound to Ago2. We undertook 3 IPs and analyzed whether a) the known human miRNAs that interact with Ago2 are present and b) whether bacterial RNAs, in particular RsmY are bound to Ago2. It needs to be stated that such an analysis is very complex and not sensitive enough for detecting very small amounts of RNAs, and to our knowledge was never done for the question asked here. As only about 50% of the cells are infected with *Lp*-EVs and of those only 30% contain RsmY, and according to our observation and previous publications about exosomes, only 1-3% of the vesicles release the cargo *in cellulo*, the probability to detect RsmY bound to Ago2 is minimal. Thus, we first verified if our IPO has worked by analysing human miRNAs known to interact with Ago2 and then we search for RsmY. Indeed, we identified most of the known Ago2 interacting human miRNAs (Table S2) but also *Legionella* RNAs. Most importantly, RsmY RNA was identified in two of the three experiments. However, given the low amount present as explained above, the results are not statistically significant but still show that Ago2 can bind RsmY. These results are now added in Table S3 and the M&M section (Lines 904-938) and in the text.

Lines 432-446 it reads now... “To determine if Ago2 plays a direct role in *Lp*-sRNA-mediated gene silencing, we analysed whether *Lp*-sRNAs and in particular RsmY directly interact with human Ago2 during infection. We infected THP-1 cells with *L. pneumophila* and used hsa-Ago2 antibodies for Ago2-immunoprecipitation experiments followed by sequencing. To validate our approach, we first analyzed

whether hsa-miRNAs known to interact with Ago2 were among the sequences obtained. Indeed, we identified 72 known Ago2-interacting hsa-miRNAs (**Table S3**), but most excitingly we also identified RNAs derived from *L. pneumophila*, in particular we identified RsmY in two of our three pull downs. Given the fact that only about 50% of the THP-1 cells are infected by *L. pneumophila*, that only 30% of *Lp*-EVs contain RNA (**Figure S1C**), and that in our assays only about 3% of the *Lp*-EVs release their cargo in the tested condition (**Figure 3E**), the probability to identify *Lp*-derived RNAs in the bulk of human RNAs is very low. Thus, although the results are not statistically significant, our Ago2-CLIP indicated that RsmY seem to directly interact with Ago2 during infection. These results further support our model that RsmY and other *L. pneumophila* RNAs can act in a mi-RNA like manner in the host cell.

- Although the authors claim that RsmY and tRNA-Phe adopt microRNA-like functions, no data are provided regarding the incorporation of these two RNAs into the host microRNA machinery. In addition to Northern blot analysis of RsmY and tRNA-Phe processing fragments in the microRNA size range, Ago2-CLIP needs to be performed to validate the association of RsmY / tRNA-Phe or their derived fragments with the microRNA machinery.

It is not possible to do Northern blots, as the concentration of RsmY/tRNA-Phe in the host cell is too little that it could be visualized by Northern blot. It is already very rarely possible to see the protein effectors secreted in the host cytosol, due to the small amounts that are delivered. However, as requested, we have undertaken Ago2-CLIPseq and have shown that we can identify RsmY bound to Ago2. See answer question above and **Lines 432-446** in the text and **Lines 904-938** in M&M section.

- The abundance of RsmY in the host cell cytoplasm and nucleus during a physiological infection setting remains unclear. What are the RsmY copy numbers in cytosolic versus nuclear fractions after stimulation with a physiological MOI / vesicle amount? Does RsmY omit the nuclear steps of the microRNA processing machinery?

Initial processing of pri-miRNA in human cells occurs in the nucleus by the Drosha complex which crops the miRNA into a hairpin-shaped pre-miRNA. However, the processing of the pre-miRNA is typically located in the cytoplasm (DICER and RISC loading). Thus, the bacterial RNA might not be necessarily pri-pre-processed in the nucleus but is probably incorporated in the RISC complex directly in the cytoplasm. However, this analysis is out of the scope of the paper and would be an entirely new study.

- The authors show a co-localization of lipid dye with the lysosomes (Figure 2A/B). Are the lysosomes acidified? Do they degrade the RNA transported on or in the EVs? Can the bacterial RNA escape the lysosomes and can be found in the cytosol? Cellular fractionation experiments with subsequent isolation of RNA should be performed to prove the presence of sRNA in the host cell cytoplasm. Besides, the tracking by immunofluorescence should be prolonged and include a late timepoint.

The reviewer seems to have mixed the figures; it was Figure 3A/B where we showed co-localization of lipid dye labelled *Lp*-EVs with acidic vacuoles, lysotracker labels acidified structures thus lysosome and late endosomes and other acid subcellular structures. However, we have no data which percentage of the RNA might be eventually degraded, but we did not write nor assume that 100% of the RNA is successfully translocated into the host cell. We do not think that cell fractionation is a feasible method as there is too little quantity of bacterial RNA in the cytosol that it would be detected by this method given that only a part of the cells is infected and of these only maximal 30% of the EVs contain RNA. To confirm that the RNA is in the cytosol we have done as described above Ago2-CLIP, that has proofed that RNA and in particular RsmY is indeed present the host cytosol. See answer question above and **Lines 432-446** in the text and **Lines 908-929** in M&M section. Furthermore, we have developed an assay based on the split Luciferase and have used it the first time to analyse content release of bacterial EVs. Indeed, we show that the *Lp*-EVs release their content in the cytosol, but only a low amount (3%), comparable to what is observed also for release from EVs originating from mammalian cells (**Figure 3E**). To determine the importance of acidification for this process, we then used Bafilomycin A1 to inhibit endosomal acidification which showed that content release is dramatically decreased, pointing to the importance of an acidic environment for content release, in agreement with our immunofluorescence tracking results. These results are added in **Figure 3E** and in the text.

Lines 344-373 it reads... To examine whether *Lp*-EVs release their content in the host cell cytosol, we developed a content release assay based on a recent study that followed the delivery of a soluble EV-cargo (HSP70, human homolog of GroEL) within the cytosol of the acceptor cells⁴². This assay was upgraded by taking advantage of split-luciferase complementation system {Somiya, 2021 #4247. Briefly, an EV cargo was tagged with HiBiT (split luciferase 1/2) and isolated EVs were incubated on acceptor cell sexpressing LbBit (Split luciferase 2/2) within their cytosol. Luciferase complementation only occurs when EV deliver their cargo into the cytosol of acceptor cells. We had previously shown that the bacterial protein GroEL (*jpp0743*) is present in the *Lp*-EVs. Thus, we tagged GroEL with a HiBiT-tag and in parallel, we transfected THP-1 cells with a LgBiT construct (pCMV-Tag2-LgBiT) under the control of the CMV promoter to express the LgBiT protein in the host cell. If the EV-content is released, Luciferase complementation occurs, and this can be measured. To determine whether the *Lp*-EVs had released their content, we measured luciferase activity *in cellulo* with the *Nano-Glo(R) Live Cell Assay System (Promega)* after 3h of incubation with purified *Lp*-EVs containing GroEL-HiBiT. To estimate the total input of *Lp*-EVs containing GroEL-HiBiT, the luciferase activity was quantified after lysis of the *Lp*-EVs using the *Nano-Glo (R) HiBiT Lytic detection System (Promega)*. *Lp*-EVs not containing GroEL-HiBiT were used as negative control. As shown in **Figure 3E**, we could detect around 3% luciferase activity of the total GroEL-HiBiT input after 3h post-infection. This result also corresponds to our co-localization experiments, where after 3h pi around 5% of *Lp*-EVs co-localized with early or late endosomes, respectively (**Figure 3D**). When adding 10 μ M of cytochalasin D to the cells, the uptake and/or release was completely abolished

(**Figure 3D**), suggesting that actin-dependent processes play a crucial role in the endocytosis and/or membrane fusion events.

Strikingly, after the addition of 200nM Bafilomycin A1, which inhibits endosomal acidification, less than 1% of the total luciferase activity was detected after 3h pi meaning that less than a third of GroEL-HiBiT proteins were reacting with cytosolic LgBiT protein of the THP-1 cells compared to non BA1-treated samples. As it was shown recently that Bafilomycin A1 does not change the general uptake of EVs into the host cell ⁴², this is another hint that the acidification in late endosomes or lysosomes might be an important factor for *Lp*-EV content release.

As requested, we have prolonged the tracking by immunofluorescence and have done it in hMDMs to show that this happens also in other cells. These new data are now added as **Figure S3B** and added in the text.

Lines 323-326 it reads now... “Similarly, when using hMDM cells and pHrodo to label acidic structures and tracking the *Lp*-EVs within these cells for 17h the *Lp*-EVs clearly co-localize with acidic subcellular structures (**Supplementary Figure S3B**), but overall intensities and differences were less pronounced than in the U2OS cell lines.”

- The authors describe that they used different protocols for RNA preparation from bacteria and EVs for sequencing. EV-RNA was isolated using miRNeasy Mini Kit and RNA was DNase digested. They do not comment on the isolation protocol for bacterial (cellular) RNA, but this RNA was additionally rRNA depleted and fragmented. This additional steps can give a bias in sample preparation and does not support the comparability of sequenced RNA.

There seems to be a misunderstanding, perhaps we did not describe it clearly enough. Bacterial RNA and *Lp*-EV RNAs were purified with the same kit and DNase digest. The only difference was that the EVs were treated routinely with RNase to avoid contamination of extracellular RNA or RNA that stuck to the EVs outside and that the bacterial RNA was rRNA depleted as this needs to be done as one sequences only rRNA when this depletion step is not undertaken. However, to avoid wrong results because of this difference in the purification step, we did not take any rRNA that was identified within the EVs into account and eliminated them from our bioinformatics analyses. We have described our procedure now more in detail in the M&M and hope this is now clearer.

Lines 755-766 it reads now... *Lp*-EV pellets, purified from a 300ml liquid culture as described above and the corresponding bacterial pellets were resuspended in Qiazol and the RNA extraction was performed following the instruction of the miRNeasy®Mini kit (Qiagen). RNA samples were digested with Turbo DNase (Thermo Scientific) and the size distribution of the EV-RNA was evaluated with a Bioanalyzer (Agilent Technologies). The EV and bacterial RNA (but not *Lp*-EV-RNA) was rRNA depleted using the RiboZero rRNA Removal Kit for Gram-negative bacteria (Illumina) and metal-catalysed heat-fragmented to a size around 100-200nts using an RNA fragmentation kit (Ambion). The bacterial RNA was further processed according to the TruSeq stranded mRNA sample preparation guide of Illumina.

Before Illumina HiSeq multiplex sequencing, the quantity was determined with a Qubit 2.0 (Invitrogen) and the quality was checked by Bioanalyzer (Agilent Technologies). *Lp*-EV RNAseq analysis was done n=4 independent biological experiments.

(iii) Concerns regarding the exact interference with the RIG-I pathway:

- The RsmY-dependent effects on the host immune response are not convincingly presented. In addition to RIG-I, TBK1 and IRF3/7 Western blots an RNA-Seq analysis of gene expression changes in WT versus Δ RsmY Legionella infected / vesicle stimulated cells would be required to portrait the global impact of Legionella-encoded microRNA-like molecules on host gene expression.

As requested, we have analyzed by RNAseq the differences in the host immune response of cells infected with *Lp*-EVs purified from the wt strain and *Lp*-EVs purified from the Δ *rsmY* strain. At 3hpi we could see only small and not significant differences in the host response, with a special focus on ISGs and the RLR pathway genes. However, this is not surprising to us, as the host immune response is not regulated only by RsmY but many other factors play a role in infection (e.g. tRNA-Phe, LPS, ...) which all play together to get a robust change in the gene expression program that can be measured by RNAseq. One should also bear in mind, that not all cells are indeed infected with *Lp*-EVs (we estimate about 50% of the cells to be really infected) thus the background of the non-infected cells is high and may also mask more important transcriptional changes. We have added these results in the text and in the M&M section.

Lines 505-521 it reads now... “To further investigate the impact of RsmY on the host immune response, we performed RNAseq analyses comparing THP-1 cells incubated with *Lp*-EVs purified from wt bacteria or *Lp*-EVs purified from the RsmY mutant strain at 3h pi. Our results revealed that only slight but not significant differences in the host cell transcriptome were present. Indeed, the difference in the IFN- β levels we observed between THP-1 cells treated with *Lp*-EV purified from the wt strain and *Lp*-EVs purified from the Δ *rsmY* strain are apparently not enough to see significant changes in the transcription of Interferon stimulated genes (ISG) at transcript level. However, this is not surprising as accumulating evidence indicates that ISG expression is not solely dependent on IFN- β , but that ISGs can also be up-regulated directly after a pathogen infection independent of IFN- β signalling, thus the network underlying the regulation of the ISG is much more complex⁵⁹⁻⁶¹. Furthermore, RsmY and tRNA-Phe seem to act in concert on the host immune response and a knockout of both, which is unfortunately not possible to achieve, might lead to more important effects on the transcriptional level. Overall, the differences on transcript level between wt *Lp*-EV and Δ *rsmY*-EV treated cells are small and not significant including the transcripts of *ddx58* (Rig-I) and *irak1* suggesting also that post-transcriptional effects may play a more dominant role in the regulation of protein expression by *Lp*-EVs.”

In addition, interferon ELISAs are required to document the postulated effect of RsmY on RIG-I-induced host responses.

There seem to be a misunderstanding, we have undertaken interferon ELISAs to measure the effect of RsmY and tRNA-Phe, after incubation with *Lp*-EVs and after RNA transfection. These results are represented in **Figure 4F and 4G**. Moreover, we measured the extracellular IFN- β levels by ELISA also during bacterial infection (Suppl Fig S5A)

As was requested by reviewer 2 we also did experiments with TLR agonists to see the influence on the *Lp*-EVs on their stimulation. In these experiments IFN- β is also measured by ELISA (**Figure S6**)

- The authors do not provide proof for RIG-I-dependence of the effects of Legionella RsmY deletion on host immune signalling. Experiments with RIG-I / MAVS deficient cells are required to exclude the possibility that the presented effects depend on additional pattern recognition receptor pathways.

As requested by the reviewer we tried to obtain RIG-I / MAVS deficient THP-1 cells but did not succeed during this revision period, which was in addition very impacted by a Covid lockdown and work time restrictions. Thus, we could not conduct this experiment. Furthermore, due to the fact that RsmY and tRNA-Phe (and maybe also other potential EV-RNAs) have the same impact on RIG-I an infection of RIG-I knockout cells with an RsmY mutant alone might thus not be sufficient to obtain significant results. However, to answer the request of the reviewer, we analysed the intracellular replication of *L. pneumophila* after knockdown of *ddx58* (Rig-I) and IRAK1 by siRNA. The result again strengthens our observation that RIG-I is important during *L. pneumophila* infection (**new Fig 4H**). Additionally, we performed infection experiments comparing RsmY and as-RsmY RNA transfection (**new Suppl Fig S5D**) showing that *L. pneumophila* grows better when RsmY is present.

Lines 538-549 it reads now... To further analyze the mechanism, we specifically down-regulated the expression of *ddx58* and *irak1* by siRNA-mediated gene silencing. Protein levels of Rig-I or IRAK1, respectively were reduced by 60-80% at the time point of infection compared to scramble transfected control cells (**Figure 4H**). After 24 hpi, no significant differences in the replication of *L. pneumophila* were detected, but at later time points (48 and 72 hpi), the number of bacteria in cells where DDX58 (Rig-I) was downregulated by siRNA was increased by up to 50% further confirming that suppression of Rig-I is beneficial for intracellular replication of *L. pneumophila*. Additionally, we transfected THP-1 cells with RsmY RNA or its anti-sense sequence (as-RsmY) and infected these cells. *L. pneumophila* replicated significantly better in the cells transfected with RsmY-RNA, similar to what was observed after siRNA knockdown of *ddx58*, again showing that RsmY has a beneficial effect on *L. pneumophila* replication **Supplementary Figure S5D**)

Finally, we pre-treated THP-1 cells with *Lp*-EVs purified from the wt or Δ *rsmY* strains (**new Suppl Fig S5C**). This experiment highlights the effect of *Lp*-EVs and in particular of those containing RsmY-RNA on *L. pneumophila* survival and propagation.

Lines 530-538 it reads now...” To investigate the influence that IFN- β secretion, partly induced by RsmY, has on infection, we treated THP-1 cells with different concentrations of IFN- β and analyzed the replication phenotype of *L. pneumophila*. We show that increasing concentrations of extracellular IFN- β reduce intracellular replication of *L. pneumophila* in THP-1 cells, whereas high concentrations of IL-1 β have no impact (**Supplementary Figure S5B**). We then pre-treated the THP-1 cells with *Lp*-EVs either purified from wt bacteria or with *Lp*-EVs purified from the Δ *rsmY* strain. In cells pre-treated with wt *Lp*-EVs we observed a significantly higher replication of *L. pneumophila* than in cells incubated with *Lp*-EVs purified from the Δ *rsmY* strain (**Supplementary Figure S5C**)

- The authors state that RsmY in EVs is regulating cRel, but they do not show a regulation of cREL on protein level by *Legionella* infection of EV treatment of the macrophages.

Indeed, the decrease at protein level was not significant, but as the more sensitive luciferase assay detected a difference, we added it to our results, but we focused mainly on RIG-I and IRAK1 as these two proteins showed differences in all our assays conducted. However, this result is further showing that several factors and several RNAs act on the same pathway.

- The authors used the bone osteosarcoma epithelial cell line U2OS cells for parts of their experiments. They should reproduce the experiments in a more physiological cell culture model for *Legionella pneumophila* infection as osteosarcoma cells might respond to differently to a bacterial infection and the stimulation of PRRs with PAMPs present on or in *Legionella* EVs. In addition, THP-1 cells were used. These cells are monocytes, unless they are differentiated with phorbol 12-myristate 13-acetate in macrophages. Did the authors use monocyte- or macrophages-like cells? And if so, how have they been differentiated?

As requested by the reviewer we have conducted several assays now also in primary macrophages. We have analyzed the impact of *Lp*-EVs on RIG-I and IRAK 1 protein levels, the dual luciferase assay and the pRhodo assay also in hMDMs purified from human blood donors. The results are confirming our results obtained with THP-1 cells. These results are added in the text and as figures.

Lines 218-220 it reads now... This phenotype was also observed when incubating human monocyte derived macrophages (hMDM) with *Lp*-EVs purified from wt *L. pneumophila* or the Δ *rsmY* strain (**Supplementary Figure 2A**).

Line 402-405 it reads now...” Additionally, we undertook the dual luciferase reporter gene assay described above also in primary cells, to rule out that this result is due to the cell line used. Indeed, when repeating the above-described experiment in CD14+ cells isolate from human blood we obtained the same result as in THP-1 cells (**Supplementary Figure S4B**).

Lines 322-326 it reads now... “Similarly, when using hMDM cells and pHrodo to label acidic structures and tracking the *Lp*-EVs within these cells for 17h the *Lp*-EVs clearly co-localize with acidic vacuoles (**Supplementary Figure 3B**), but overall intensities and differences were less pronounced than in the U2OS cell lines.”

Reviewer #2 (Remarks to the Author):

In this work Sahr et al have identified two *Legionella* sRNA species which are proposed to mimic host miRs to control innate immune responses. Although the original observation is interesting, the work appears rather under-developed, and there are limited data on the physiological implication of the observation, including in primary cells.

We thank this reviewer for the comments and have taken them into account and have undertaken additional experiments as requested. However, the main concern of this reviewer seems that the impact of RsmY on INF secretion and the host immune response is small. The experiments undertaken here, further confirmed that RsmY has as significant but small impact, however to us this is not surprising but expected. In addition to the many traditional, bacterial virulence factors like LPS, Flagella, pili, type II secretion system or outer membrane proteins *Legionella* secretes over 330 effectors by its type 4 secretion system Dot/Icm. These many factors and in particular the T4SS effectors work together to subvert the host immune response and are often redundant. This led different groups even to decipher how to understand this redundancy (e.g. O'Connor TJ, Boyd D, Dorer MS, **Isberg RR**. “Aggravating genetic interactions allow a solution to redundancy in a bacterial pathogen.” *Science*. 2012 Dec 14;338(6113):1440). Thus, if one single virulence factor is deleted, it is rare that one can observe strong phenotypes in *Legionella* growth, mostly there is not even a phenotype because of redundancy. Thus, it is not surprising that RsmY has only small impact but as it is significant and reproducible in many different systems, it is one piece in the puzzle of the manifold ways how *Legionella* can subvert the hosts immune response.

1. The manuscript critically lacks data on the impact of the identified mechanism on host innate immune responses and anti-microbial defense. The data presented in Fig 4D-E are based on purified EVs and in THP1 cells, and show only a modest effect. As a minimum, the authors should compare IFN β (and ISG) responses to infection with wt and KO bacteria in THP1 cells and primary macrophages.

There might be an oversight, we have quantified the IFN- β response in wt and RsmY mutant strains in THP-1 cells. This result was and is depicted in **Figure S5A** (former **figure S4E**).

Lines 487-500 it reads... . We quantified the amount of extracellular IFN- β by performing an ELISA with the supernatants at different time points of the infection. However, no significant differences were found when IFN- β concentrations of cells

infected wt *L. pneumophila* were compared to cells infected with the $\Delta rsmY$ -mutant strain (**Supplementary Figure S5A**), suggesting that this approach does not reveal the influence of RsmY on IFN- β secretion as additional factors may also influence IFN- β levels as known for several bacterial and viral infections⁵⁸, and these combined effects are measured. Thus, to measure specifically the impact of *Lp*-EVs containing RsmY, we analysed the extracellular concentration of IFN- β in the supernatant of THP-1 cells after incubation with *Lp*-EVs purified either from wt *L. pneumophila* or from the $\Delta rsmY$ mutant strain. Indeed, as shown in **Figure 4F**, internalization of *Lp*-EV containing RsmY (purified from the wt strain) induces less IFN- β secretion by the host cells, than those infected with *Lp*-EVs from which RsmY is absent (purified from the $\Delta rsmY$ strain). These results suggest that *Lp*-EVs containing RsmY dampen IFN- β secretion of infected human cells.”

To analyze also primary cells, we have analyzed the impact of *Lp*-EVs on RIG-I and IRAK 1 protein levels, the dual luciferase assay and the pHrhodo assay also in primary cells (hMDMs purified from human blood donors). The results are confirming our results obtained with THP-1 cells. These results are added in the text and as figures.

Lines 218-220 it reads now... This phenotype was also observed when incubating human monocyte derived macrophages (hMDM) with *Lp*-EVs purified from wt *L. pneumophila* or the $\Delta rsmY$ strain (**Supplementary Figure 2A**).

Line 402-405 it reads now...” Additionally, we undertook the dual luciferase reporter gene assay described above also in primary cells, to rule out that this result is due to the cell line used. Indeed, when repeating the above-described experiment in CD14+ cells isolate from human blood we obtained the same result as in THP-1 cells (**Supplementary Figure S4B**).”

Lines 322-326 it reads now... “Similarly, when using hMDM cells and pHrodo to label acidic structures and tracking the *Lp*-EVs within these cells for 17h the *Lp*-EVs clearly co-localize with acidic vacuoles (**Supplementary Figure 3B**), but overall intensities and differences were less pronounced than in the U2OS cell lines.”

2. Along the same lines, the authors should more globally characterize how the sRNAs affect host cell gene expression through RNAseq analysis.

As requested, we have analyzed by RNAseq the differences in the host immune response of cells infected with *Lp*-EVs purified from the wt strain and *Lp*-EVs purified from the $\Delta rsmY$ strain. At 3hpi we could see only small and not significant differences in the host response, with a special focus on ISGs and the RLR pathway genes. However, this is not surprising to us, as the host immune response is not regulated only by RsmY but many other factors play a role in infection (e.g. tRNA-Phe, LPS, ...) which all paly together to get a robust change in the gene expression program that can be measured by RNAseq. One should also bear in mind, that not all cells are indeed infected with *Lp*-EVs as we estimate about 50% of the cells to be really infected and only a smaller amount of these vesicles enter the endosomal pathway

and release the cargo (10% and 3%, respectively), comparable to what is also observed for exosomes. Thus, the background of the non-infected cells is high and may also mask more important transcriptional changes. We have added these results in the text and in the M&M section.

Lines 505-521 it reads now... “To further investigate the impact of RsmY on the host immune response, we performed RNAseq analyses comparing THP-1 cells incubated with *Lp*-EVs purified from wt bacteria or *Lp*-EVs purified from the RsmY mutant strain at 3h pi. Our results revealed that only slight but not significant differences in the host cell transcriptome were present. Indeed, the difference in the IFN- β levels we observed between THP-1 cells treated with *Lp*-EV purified from the wt strain and *Lp*-EVs purified from the $\Delta rsmY$ strain are apparently not enough to see significant changes in the transcription of Interferon stimulated genes (ISG) at transcript level. However, this is not surprising as accumulating evidence indicates that ISG expression is not solely dependent on IFN- β , but that ISGs can also be up-regulated directly after a pathogen infection independent of IFN- β signalling, thus the network underlying the regulation of the ISG is much more complex⁵⁹⁻⁶¹. Furthermore, RsmY and tRNA-Phe seem to act in concert on the host immune response and a knockout of both, which is unfortunately not possible to achieve, might lead to more important effects on the transcriptional level. Overall, the differences on transcript level between wt *Lp*-EV and $\Delta rsmY$ -EV treated cells are small and not significant including the transcripts of *ddx58* (Rig-I) and *irak1* suggesting also that post-transcriptional effects may play a more dominant role in the regulation of protein expression by *Lp*-EVs.”

3. To make sure that the observed effects of bacterial EVs are in fact due to targeting of RIG-I, the authors should generate RIG-I KO THP1 cells and demonstrate that the modulatory effect of the EVs is lost.

As requested by the reviewer we tried to obtain RIG-I / MAVS deficient THP-1 cells but did not succeed during this revision period, which was in addition very impacted by a Covid lockdown and work time restrictions. Thus, we could not conduct this experiment. Furthermore, due to the fact that RsmY and tRNA-Phe (and maybe also other potential EV-RNAs) have the same impact on RIG-I an infection of RIG-I knockout cells with an RsmY mutant alone might thus not be sufficient to obtain significant results. However, to answer the request of the reviewer, we analysed the intracellular replication of *L. pneumophila* after knockdown of *ddx58* (Rig-I) and IRAK1 by siRNA. The result again strengthens our observation that RIG-I is important during *L. pneumophila* infection (**new Fig 4H**). Additionally, we performed infection experiments comparing RsmY and as-RsmY RNA transfection (**new Suppl Fig S5D**) showing that *L. pneumophila* grows better when RsmY is present.

Lines 538-549 it reads now...” To further analyze the mechanism, we specifically down-regulated the expression of *ddx58* and *irak1* by siRNA-mediated gene silencing. Protein levels of Rig-I or IRAK1, respectively were reduced by 60-80% at the time point of infection compared to scramble transfected control cells (**Figure 4H**). After 24 hpi, no significant differences in the replication of *L. pneumophila* were

detected, but at later time points (48 and 72 hpi), the number of bacteria in cells where DDX58 (Rig-I) was downregulated by siRNA was increased by up to 50% further confirming that suppression of Rig-I is beneficial for intracellular replication of *L. pneumophila*. Additionally, we transfected THP-1 cells with RsmY RNA or its anti-sense sequence (as-RsmY) and infected these cells. *L. pneumophila* replicated significantly better in the cells transfected with RsmY-RNA, similar to what was observed after siRNA knockdown of *ddx58*, again showing that RsmY has a beneficial effect on *L. pneumophila* replication **Supplementary Figure S5D**)

Finally, we pre-treated THP-1 cells with *Lp*-EVs purified from the wt or Δ *rsmY* strains (**new Suppl Fig S5C**). This experiment highlights the effect of *Lp*-EVs and in particular of those containing RsmY-RNA on *L. pneumophila* survival and propagation.

Lines 530-538 it reads now... To investigate the influence that IFN- β secretion, partly induced by RsmY, has on infection, we treated THP-1 cells with different concentrations of IFN- β and analyzed the replication phenotype of *L. pneumophila*. We show that increasing concentrations of extracellular IFN- β reduce intracellular replication of *L. pneumophila* in THP-1 cells, whereas high concentrations of IL-1 β have no impact (**Supplementary Figure S5B**). We then pre-treated the THP-1 cells with *Lp*-EVs either purified from wt bacteria or with *Lp*-EVs purified from the Δ *rsmY* strain. In cells pre-treated with wt *Lp*-EVs we observed a significantly higher replication of *L. pneumophila* than in cells incubated with *Lp*-EVs purified from the Δ *rsmY* strain (**Supplementary Figure S5C**)

4. The work would gain significantly, if induction of type I IFN by a panel of synthetic agonists for TLRs and cytosolic PRRs were evaluated in cells treated with relevante EVs.

As requested by the reviewer we have undertaken experiments using TLR agonists. We have pretreated THP-1 cells for 3h with or without *Lp*-EVs purified either from *L. pneumophila* wt or from the Δ *rsmY* strain. Subsequently, TLR-related agonists were added and the extracellular IFN- β concentrations were measured by ELISA 20h post incubation showing that particularly TLR signalling pathways depending on IRAK1 like TLR1, TLR2, TLR6 or on RLR (Poly(I:C), ssRNA40) are significantly down-regulated after *Lp*-EV-treatment, whereas pathways that can also be activated via alternative signalling routes e.g. through TRIF/TRAM (TLR3, TLR4), or cGAS-STING (ODN2006) are less affected or can even be stimulated by *Lp*-EVs. *Lp*-EVs purified from the Δ *rsmY* strain, thus lacking RsmY inhibit IFN- β -secretion significantly less than *Lp*-EVs purified from *L. pneumophila* wt strain. Additionally, Ago2-Inhibition significantly increases the IFN-beta secretion of *Lp*-EV-treated THP-1 cells These new results are added as Supplementary figure 6 and in the text.

Lines 558-570 it reads ... Finally, to further characterize the impact of *Lp*-EVs on the host immune response, we incubated THP-1 cells with agonists that mimic pathogen-associated molecular patterns (PAMPs) and measured extracellular IFN- β concentrations after pre-treatment with *Lp*-EVs that were either purified from wt

bacteria or from the $\Delta rsmY$ strain. Indeed, the IFN- β response of certain TLR agonists was dampened after pre-incubation with wt *Lp*-EVs but less with $\Delta rsmY$ -EVs further pointing to the influence of RsmY on the host immune response. In particular the IFN- β response triggered by agonists for TLR1/2/5/6 and TLR8 was significantly reduced when pre-treating the cells with *Lp*-EVs (**Supplementary Figure S6**). The IFN- β response to TLR9 agonist CpG instead was even more pronounced after pre-incubation of the THP-1 cells with *Lp*-EVs compared to control experiments, probably due to synergetic effects of multiple ligand stimulations. In contrast, agonist stimulation of TLR3, TLR4 or TLR7 was not affected by *Lp*-EV-treatment, whereas inhibition of Ago2 slightly induced the extracellular IFN- β levels after *Lp*-EV-treatment (**Supplementary Figure S6**).”

5. The functional data do generally not show a very large effect of the EVs/sRNAs (e.g. in Fig 4). Therefore, for the work to have impact, it is essential that the authors show data on the effect of the proposed immunomodulatory RNAs in bacterial growth.

We have tested the growth of a $\Delta rsmY$ strain in growth in THP-1 cells compared to the wt strain. As seen from the graph below, the difference is very small and there is no real growth defect of the $\Delta rsmY$ strain compared to the wt strain. However, this is not surprising but a result which is well known in the *Legionella* field. Given the over 300 effectors *L. pneumophila* is secreting in addition and all work in concert and are often redundant to manipulate the host response, a big growth defect is rarely observed when knocking them out. Even among the over about 30 effectors analysed to date there are only two or three that have an important impact on growth in our classical growth assays. In addition, the growth assays are not very sensitive and thus we did not expect a big difference in growth when RsmY is missing. We did not add the figure to the manuscript but can do this of course if the reviewer thinks it is important to show.

However, we observed a clear and significant effect on the intracellular growth of *L. pneumophila* when we pre-treated the THP-1 cells with *Lp*-EVs deriving from wt or the $\Delta rsmY$ mutant strain, indicating that a high number of purified vesicles indeed leads to a positive effect on intracellular replication of *L. pneumophila* depending on RsmY (**new suppl Fig S5C**).

6. The functional data in Fig 4, should be confirmed in primary cells, and ideally also in mice (if the sRNAs also target RIG-I and IRAK1 in mice).

As requested, we have confirmed the data of Figure four in primary cells by redoing the dual luciferase assay in CD4+ cells purified from human blood donors. This result is added in figure S4B and in the text.

Line 402-409 it reads now...” Additionally, we undertook the dual luciferase reporter gene assay described above also in primary cells, to rule out the possibility that the result is due to the cell line used. Indeed, when repeating the above-described experiment in CD14+ cells isolated from human blood we obtained the same result as in THP-1 cells (**Supplementary Figure S4B**). These results are also in agreement with the results obtained after RNA transfection (**Figure 1I and Supplementary Figure S2B**), further supporting our results that RsmY interacts with the UTR of the RIG-1, and tRNA-Phe with the UTR of the *irak1* encoding gene and indeed can behave like eukaryotic micro-RNAs.

As to the mouse experiments we do not think that this is necessary nor ethically to be defended. Firstly, Legionella’s natural host are amoeba and phagocytic cells, and this is where we have done all our experiments. Secondly mice are not good hosts for Legionella, except AJ mice that have a mutation in Naip5, and in addition mouse macrophages show a very different immune response to *L. pneumophila* infection than what is observed for human cells. Other mouse strains cannot be infected with wt *L. pneumophila*. RIG-I and IRAK1 knock out mice are only available in BALBc mice, and BALBc mice cannot be infected with wt *L. pneumophila* except when constructing in addition a *flaA* mutant to knock out flagellin that is recognized by the mouse immune system.

REVIEWER COMMENTS

Reviewer #1 (Remarks to the Author):

In this revised manuscript, Sahr et al. claim that *Legionella pneumophila* deliver bacterial sRNAs via outer membrane vesicles (OMV) to eukaryotic host cells and thereby regulate RIG-I signaling in a microRNA like way of action.

As stated before, this would a scientifically most relevant observation.

The authors included new experiments to their manuscript by performing OMV characterization experiments by nano particle tracking analyses (NTA) and included experiments in primary human macrophages, as well as pull down and sequencing experiments.

Considering all new data, the difficulties due to the pandemic, and questions that would be nice to know, but that are probably not necessary to validate the central claim, there are still some concerns: The authors did apply size exclusion experiments with the isolated OMVs (line 733f). They should describe the columns that they used, the column material, the fractions that they used and how they pooled and concentrated them for further analyses. These are critical information for the reader. The authors state that they used vesicles purified with the floatation assay for NTA, but vesicles isolated via differential ultracentrifugation for cryo EM. These are two completely different isolation methods and two different methods for measuring vesicles sizes and cannot be compared. The authors obtain a vesicle preparation with 120 nm in diameter (Fig. 1D right panel), which is not reflected by their initial cryo EM images. Did the authors compare the effect of the much purer vesicles fraction obtained by sucrose floatation and size exclusion chromatography for stimulation experiments? As they obtained different vesicle preparation, the subsequent effect of recipient cells might be different as well. The authors' comment that they initially excluded that the free DiD dye was showing up in FACS experiments and refer to Fig. S1B, but the "DiD alone" (a) shows up in the same gate in the FSC/SSC as the labeled EVs (b). Besides that, the authors stated that "conventional flow cytometry seem to underestimated [...] by 1,000 times" (line 698 following). As they observe little effect with such high amounts of OMVs that they apply to the macrophages (they planned to stimulate with MOI 10, but used 1,000 times more. Line 766 following), the physiological relevance of the described results on the infection outcome is debatable.

The authors state that they did not observe bacterial lysis due to growth to stationary phase, but observe differences in vesicle shapes that they cannot explain. They are using a mixed vesicle population for their stimulation experiments and do not clearly show what is mediating the effect they want to describe.

There are also still concerns about the claimed miRNA-like mode of action that for the sRNA RsmY: The authors state that they do not know the exact mode of action, but compare the bacterial sRNAs with human microRNAs (Table 1). By doing so, they are assuming comparable modes of actions and should show the interaction of bacterial sRNAs with 3'UTRs of human mRNAs. Moreover, they are only using the UTRs of the human mRNAs (Fig. 4A-D)

They performed Ago2 inhibition experiments to show the microRNA like manner of ddx58/RIG-I and IRAK1 inhibition, but these can also be mediated by host microRNAs that are simultaneously inhibited. Moreover, for these two mRNAs, effects by human microRNAs in the context of *Legionella pneumophila* infection are published (ddx58: PMID: 32209695; IRAK1: PMID: 27105429). The authors should discriminate in their OMV stimulation experiments between host miRNA effects and bacterial sRNA effects. As Ago2 is a critical molecule for eukaryotic cells, it might be the case that Sahr et al. observed off-target effects by applying the inhibitor that they would see on any 3'UTR. Why do the authors observe a reduction of luciferase activity (Fig 4B) when they combine ddx58 UTR with rsmY-EVs? As the two neighboring bars show a comparable reduction, this argues for an RsmY-independent effect.

The authors performed Ago2 pulldown and sequencing experiments after *L.pneumophila* infection and found RsmY in their Ago2 pulldown-seq data, but they do not show this data in the manuscript (except for human miRNAs in a supplemental table). Could the authors explain why they performed infection experiments and not OMV stimulation experiments as they aimed to show that this sRNA is transport via extracellular vesicles and is thereby taken up by eukaryotic host cells? There might be a completely different mode of sRNA translocation into the cytosol of an infected host cell as intended by the authors.

The authors performed further tracking experiments with human monocyte derived macrophages, but observed a colocalization with the OMVs and acidic subcellular structures after 17 h post stimulation (Fig S3B). This effect was already visible within the first hour of addition in U2OS cells. How do the authors explain this massive delay in macrophages?

The authors tried to address the concerns regarding the interference with the RIG-I pathway on the host immune response by performing stimulation and sequencing experiments after 3 h. They aimed to reveal differences due to differences in IFN β release, which they measured at the same timepoint (Fig 4F). If the OMV stimulated cells are responding to differences in the released IFN β , one would expect to see them as a functional outcome in different ISGs. But they also performed IFN β ELISA over a longer time period where they do not see an increase in IFN β release by Lp wt and Lp RsmY (Fig. S5A). How do the authors explain these differences in infection and stimulation experiments?

The authors performed knockdown experiments for RIG-I and IRAK-1 and subsequent infection with *Legionella pneumophila*. Such experiments have already been published by others (ddx58: PMID: 32209695; IRAK1: PMID: 27105429). They performed OMV pre-stimulation and infection experiments and observed an increase in Lp replication in wt-OMV pre-treated cells but not with RsmY-lacking OMVs. Cells pre-treated with RsmY-lacking OMV show the same *Legionella* replication as not pretreated (only infected) cells. This does not argue for the importance of RsmY.

The authors performed IFN β pre-stimulation experiments before *Legionella pneumophila* infection, but did not observe an effect with the same dose of IFN β that was induced by OMV stimulation (compare: Fig S5B left – Fig 4F) – 25 IU/mL. They start to see impaired *Legionella* replication with 50 IU/mL, but not with the dose released from the cells upon OMV stimulation. In the light of no changes in cellular transcript levels, the effect of the OMVs as proposed by the authors is not totally reflected by the presented experiments as the OMVs induced IFN β release (Fig 4F), they induced bacterial replication in macrophages (Fig S5C), while IFN β alone reduced *Legionella* replication (Fig S5B).

In general, it is confusing when the authors are writing about *Legionella* infection experiments but referring to OMV stimulation experiments. This should be labeled more precisely in the figures and stated more clearly in the text to prevent misunderstandings.

Reviewer #2 (Remarks to the Author):

This reviewer finds that the work has improved considerably by the revision. However, a number of requested experiments have still not been performed. Notably, the claimed physiological importance of the observed phenomenon is still based solely on data with purified EVs. In my opinion, this weakens the impact of the work.

REVIEWER COMMENTS

Reviewer #1 (Remarks to the Author):

In this manuscript, Sahr et al. claim that *Legionella pneumophila* translocate small RNAs via extracellular vesicles to eukaryotic host cells and that these small bacterial RNAs reduce several factors of the RIG-I signalling pathway by a microRNA-like way of action. Trans-kingdom RNA-signalling, specifically sRNAs acting as microRNAs, is of very high relevance for the field of infection biology and has been discussed for several years. However, there are important concerns that need to be addressed, mainly regarding (i) the characterization of extracellular vesicles, (ii) the proof of miRNA-like activity of sRNAs, and finally (iii) the exact interference with the RIG-I pathway.

We thank the reviewers for his/her pertinent comments. We have undertaken many of the suggested experiments, which have indeed improved the manuscript. However, several questions he/she is raising are burning questions in the field, but nobody has been able to answer them yet. We would be thrilled if we could answer all these questions but believe that many laboratories will have to continue research in EVs for years before we will be able to answer everything. We have taken the concerns raised seriously and have tried to do as many of the suggested/needed experiments as possible in addition during a very challenging time (Covid lockdown in France and constant work restrictions at Institut Pasteur in parallel to not being priority of the platforms as priority was given to Covid research) which have improved our manuscript. We hope that the reviewer is satisfied with the many additional experiments conducted that all further support our hypothesis that RsmY acts in a miRNA like manner in the host cell.

(i) Concerns regarding the characterization of extracellular vesicles:

- EV purification was performed from late post-exponential phase (OD4.2) (M&M line 518). It has been shown that the growth phase has significant impact on the properties of bacterial EVs (Tashiro Y et al.; Variation of physiochemical properties and cell association activity of membrane vesicles with growth phase in *Pseudomonas aeruginosa*; Appl Environ Microbiol 76(2010), McCaig WD et al.; Production of outer membrane vesicles and outer membrane tubes by *Francisella novicida*; J Bacteriol, 195 (2013)) and the cultivation to extremely late stationary phases would lead to bacterial cell lysis as well as the contamination with the broken membranes and cytoplasmic proteins (Klimentová J & Stulík Jiri, Methods of isolation and purification of outer membrane vesicles from negative bacteria; Microbiological Research 2015).

We agree that the growth phase has significant impact on bacterial EVs and specifically for *L. pneumophila* the growth phase is very important as it exhibits a biphasic life cycle where virulence is expressed only in post exponential (PE) growth

phase (Molowsky and Swanson, Mol Micro, 2004). Thus, we choose an OD of 4.2 as this is PE growth for *L. pneumophila* and not an extremely late stationary phase. Furthermore, when one looks at the TEM images which were taken from EVs isolated from bacteria grown to OD 4.2 one can clearly see that there is no cell debris. We never observed bacterial lysis, which is expected as OD 4.2 is relatively short after exponential (E)-phase time points. We agree that exponentially grown vesicles might be different, but as we were interested in virulence impact of these vesicles PE phase bacteria were chosen. Indeed, RsmY is highly expressed only in PE phase grown bacteria, thus we are not expecting to find it in EVs purified from bacteria in E phase.

- The authors used Exosome Spin Columns for purification of EVs after staining with Vybrant DiD and Syto RNA select. It is known that these dyes are causing staining artefacts and larger aggregates even without EVs (Morales-Kastresana, A. et al. Labeling Extracellular Vesicles for Nanoscale Flow Cytometry. Sci Rep 7, 1878 (2017). <https://doi.org/10.1038/s41598-017-01731-2>). They should show that free dye and aggregates are successfully removed from the sample, as they would give signals in flow cytometry.

Indeed, we have shown that dye aggregates have successfully been removed. The reviewer might have overseen these results shown in Figure S1B, which shows a control experiment with dye alone. In addition, we now used also a sucrose flotation assay to formally rule out the possibility of contaminants during EV isolation and further characterization. This is now reported in Figure 1

How have the authors proven that RNA select is not binding to RNA on the surface of EVs?

There must have been an oversight, as it was mentioned in the M&M that the *Lp*-EVs have been treated with RNase during purification to degrade extravesicular RNAs.

Lines 689-691 it reads... The supernatant was treated with RNaseA/T1 (Thermo Scientific) at a final concentration of 2µg/ml RNaseA for 1h at 37°C followed by centrifugation at 150.000xg for 2h at 4°C to pellet the *Lp*-EVs. The *Lp*-EV-pellet was washed, re-centrifuged and resuspended in PBS. -

- Moreover, RNA-FISH in immunogold EM would strengthen the claim that small RNAs are transported in the EVs.

We agree with the reviewer that this would be a very appealing method to apply, although we do not see which additional information would be gleaned from it what was not already shown by RsmY FISH. However, we have contacted Pierron Gerard who is specialist for this technique (Soquere and Perron, Methods Mol Biol. 2015;1262:105-18.). After discussing with him this request he also told us that such an experiment would not add any additional information as we have already provided the RsmY FISH results in the paper. Furthermore, it is very likely that it is not possible to apply this technique to RNAs in EVs as a high concentration of RNA in the vesicles is required to be detected by this technique. He successfully applied it for example for HSV1 genome detection in intra- and extra-cytoplasmic virions, which

contain a high concentration of RNA, however, this is not the case for bacterial RNAs in EVs. Thus, as he does not think that this technique is feasible for this question and in addition it would not bring any additional information than FISH which we provided already we did not follow it up further.

- The authors observed differences in their EV population (spherical structures with single and double membranes and tube-shaped vesicles). These different populations should be separated by size-exclusion chromatography or density gradient centrifugation to prove which of the population harbors the RNA content as it has been shown for other gram-negative bacteria that they can contain different protein compositions (McCaig WD et al.; Production of outer membrane vesicles and outer membrane tubes by *Francisella novicida*; J Bacteriol, 195 (2013)).

We agree that these two different EV populations might contain different RNAs and or protein compositions. However, it is not the scope of the paper to define the exact content of each of the EV populations, but to analyse the impact of the RNA present in the EVs on the host cell. We are continuing to analyze the EVs and are planning to define the protein content of the EVs and to try to separate them, but this will be a future analysis and is out of the scope of this article which is already quite long with a very high number of experiments.

- The authors used conventional flow cytometry and tried to calibrate with beads (0.1-0.5 μm). Additionally, they stained the EVs with a lipophilic dye and an RNA dye (single dyed!). Conventional FACS does not cover the small size range needed for EV analyses and bead standards do not reflect the physical properties of EVs and are not showing the exact size of the EVs. Another state-of-the-art EV characterization method such as NanoFCM should be applied to characterize the size and differences in the EV preparation observed in CryoEM. Moreover, the authors could not reproduce the EV sizes they found in electron microscopy in flow cytometry (Fig. S1D).

Thank you for pointing this out, this was a mistake in the calculation which can be clearly seen with respect to the indicated size measure in the figure. Indeed, the vesicle size should have been indicated as 20-200nm in diameter and not 20-100nm as originally stated. This has been corrected

Lines 131-133 it reads now... As shown in **Figure 1A**, the *Lp*-EVs are mostly spherical structures ranging from around 20-200nm in diameter, but also tube-shaped vesicles are present.

As requested, we have also undertaken an extensive Nanoparticle tracking analyses using ZetaView that further characterizes the *Lp*-EV in depth (see below)

- The authors used conventional flow cytometry to quantify the amount of EVs for stimulation experiments. This is not a valid technique for EV quantification and the numbers are misleading as a conventional flow cytometer does not feature enough

SSC resolution to be used for standardized EV counts. The authors should use NTA, tunable resistance pulse sensing or Nano flow cytometry.

As requested by the reviewer we did new experiments and performed the quantification, size characterization and fluorescence detection of the EVs on a ZetaView® QUATT (Particle Metrix) equipment. As we do not have neither the equipment nor the expertise to analyse EVs by NTA, we have contacted our collaborators Gregory Lavieu and Sheryl Bui who undertook these analyses. Thus, we have added them as authors on the revised manuscript. To exclude the presence of aggregates we also added a sucrose floatation step in our *Lp*-EV-isolation procedure. These new results showed that using conventional flow cytometry we had underestimated the *Lp*-EV concentration in our purification, as it was estimated as 10^{11} by NTA analyses compared to 10^8 with conventional flow cytometry. This is now corrected in the text. However, the new experiments confirmed our previous size estimation of the EVs. These new results are added in the text, as a new figure (Figure 1D, E, F) and in M&M. The previous figure 1D, E, F were moved to Figure S1 as Figure S1A, C, D.

Lines 139 -159 it reads now.....” To quantify the amount of purified *Lp*-EVs and estimate their size nanoparticle tracking analyses (NTA) was used after staining the putative *Lp*-EVs with Vybrant™ DiD to stain the membranes of the EVs. Although we used a size filtration column to remove excess of free dye, the presence of aggregated dye within our samples, or the presence of other large protein/lipid aggregates emanating from *L. pneumophila* could not be completely ruled out. Thus, we added a sucrose floatation step to the isolation procedure. We first analyzed the size distribution and the number of particles pre- and post-floatation, through light scattering mode revealing that, the number of particles was moderately decreased after floatation likely due to the three additional ultracentrifugation steps required for the floatation procedure (**Figure 1D, E**). Particles in both samples showed a median size of ~ 130 nm (**Figure 1E**). In addition, we compared particles size and concentration when measured in fluorescence mode to analyze *Lp*-particles labeled with the red-lipophilic dye. Size distribution was similar (**Figure 1 F**, right panel), and ~ 85% of the particles were positive for the membrane dye, consistently with our FACS data and previous studies. These results suggest that most of the *Lp*-derived nanoparticles considered in our study are indeed *Lp*-EVs.

However, we could not use the NTA to measure the percentage of *Lp*-EVs positive for the RNA dye as the set-up of the machine was not compatible with the fluorescent properties of the RNA-dye. Thus, we used conventional flow cytometry analyses, as the NTA results with respect to the size distribution and the quantification were comparable to determine the percentage of *Lp*-EVs that contains RNA molecules (**Figure S1A**).

Lines 732-753 it reads now.....” **Floatation assays.** For the floatation assays, we proceeded as previously described^{68,69}. Briefly, *Lp*-EVs labeled with Vybrant DiD-dye or not and processed through a size exclusion column, were centrifuged at 100,000g for 1 hour (MLA-50 rotor). Pellets were resuspended in 1mL 60% sucrose and deposited in the bottom of the tube. 1mL of 30% sucrose solution and 1mL of PBS were sequentially loaded on top. Samples were centrifuged at 150,000 × *g* for 16 h at

4 °C (SW55 rotor). The top fraction was removed, and the 30% sucrose fraction (1mL) was collected and mixed with 6mL PBS. Samples were centrifuged at 100,000 × *g* for 1h30 min (MLA 50 rotor), supernatant containing sucrose was removed and pellets were resuspended in 100µL PBS prior to further analyses of the particle concentration and their size.

Nanoparticle Tracking Analysis. The quantification, size characterization and fluorescence detection of the EVs were performed on the ZetaView® QUATT (Particle Metrix). For the size and concentration measurements, the 448nm laser in scatter mode was used; for the fluorescent measurement of DiD-dye positive particles, the 640nm laser with a 660nm long-pass filter was used. In all panels, a dot represents the average of 11 measurements corresponding to the 11 frames. For the size, each dot corresponds to the average of the median size which permits to describe the distribution, whereas the mean is biased by the aggregates' extreme values. For the concentration, each dot corresponds to the average number of particles detected taking into account the dilution factor. The normality was tested with D'Agostino-Pearson test which was negative for all panels. The data are paired, and a non-parametric test (Wilcoxon test) was used. All statistical analysis were performed with GraphPad Prism version 9.1.1 for MasOS.

• Figure S1 A-D: The authors are talking about Lp-MVs here, did they measure something different as in the rest of the manuscript?

We apologize for this mistake, as the working name was MVs we had forgotten to change it. We have now corrected it, of course these are also *Lp*-EVs.

(ii) Concerns regarding the proof of miRNA-like activity of sRNAs:

• Fig. 4: The postulated RsmY and tRNA-Phe target-sites in the mRNA 3'UTRs need to be presented in the figure. In addition, it is common practice to introduce point mutations into the 3' UTR target sites in order to abrogate base pairing between the luciferase mRNA-construct and the micro (or in this case small) RNA. Without this experiment the proof that RsmY and tRNA-Phe act through a microRNA-like 3'UTR targeting mechanism is not provided. In addition to 3' UTR target-site mutations, compensatory base pair mutations should be inserted into RsmY and tRNA-Phe in order to restore the regulation of mutated luciferase constructs.

We understand that the reviewer would like to see mutations in the interacting site, however, it is not known how the sRNA present in the *Lp*-EVs are further processed in the host cell. Thus, we cannot ascertain which part is the biological active one. However, we have used the complementary sequences of RsmY (Figure S4A), as control showing that the anti-sense RsmY sequence has no impact on Luciferase activity and intracellular replication of *L. pneumophila*, which proofs that RsmY activity is not random. To further support our observations, confirm that RsmY activity is miRNA like we undertook two additional experiments:

- First, we analyzed the impact of Ago2 that is part of the RISC complex and necessary for miRNA function on the activity of RsmY on the 3'UTR of *ddx58*. We treated THP-1 cells with the Ago2 inhibitor BCI-137 and conducted the dual

luciferase assay. This showed that indeed, a functional Ago2 protein is necessary for RsmY to repress RIG-I activity. This new experiment is added as Figure 4C and D and is described in M&M and in the text.

Lines 415-431 it reads now... “Argonaute family proteins play a crucial role in RNA induced silencing complex (RISC). Thus, to further analyse if RsmY acts in a miRNA like manner we investigated whether argonaute-2 (Ago2), the only member with catalytic activity and an essential role within the RISC complex to regulate small RNA guided gene silencing processes^{54,55} impacts RsmY and/or tRNA-Phe activity. Indeed, the suppressive effect of RsmY, tRNA-Phe or of *Lp*-EVs on the relative luciferase activity of Luc2 fused to the UTR of *ddx58*/Rig-I or the UTR of *irak1* was significantly reduced when Ago2 inhibitor was added (**Figure 4C, D**). This suggests that the presence of a functional Ago2 is necessary for *Lp*-EV RNAs to interact with *ddx58* and *irak1* UTRs *in cellulo* and show that Ago2 and thus probably RISC-mediated silencing are involved in *ddx58* and *irak1* expression during *L. pneumophila* infection. However, secondary effects of Ago2 inhibition may also influence this result, since endogenous human miRNA can also play a role in the regulation of protein expression⁵⁶. Yet, as transfection of RsmY-RNA did not influence luciferase activity of Luc2 fused to the *irak1*-UTR independently of Ago2 inhibition, the observed impact of tRNA-Phe on IRAK1 cannot solely depend on the effect of endogenous has-miRNA-silencing, but it further suggests that indeed *L. pneumophila* RsmY has a significant impact on protein expression in a miRNA-like and Ago2-dependent manner. “

Secondly, we undertook CLIPseq (immunoprecipitation followed by RNA sequencing) using Ago2 antibodies to analyze whether we find RsmY bound to Ago2. We undertook 3 IPs and analyzed whether a) the known human miRNAs that interact with Ago2 are present and b) whether bacterial RNAs, in particular RsmY are bound to Ago2. It needs to be stated that such an analysis is very complex and not sensitive enough for detecting very small amounts of RNAs, and to our knowledge was never done for the question asked here. As only about 50% of the cells are infected with *Lp*-EVs and of those only 30% contain RsmY, and according to our observation and previous publications about exosomes, only 1-3% of the vesicles release the cargo *in cellulo*, the probability to detect RsmY bound to Ago2 is minimal. Thus, we first verified if our IPO has worked by analysing human miRNAs known to interact with Ago2 and then we search for RsmY. Indeed, we identified most of the known Ago2 interacting human miRNAs (Table S2) but also *Legionella* RNAs. Most importantly, RsmY RNA was identified in two of the three experiments. However, given the low amount present as explained above, the results are not statistically significant but still show that Ago2 can bind RsmY. These results are now added in Table S3 and the M&M section (Lines 904-938) and in the text.

Lines 432-446 it reads now... “To determine if Ago2 plays a direct role in *Lp*-sRNA-mediated gene silencing, we analysed whether *Lp*-sRNAs and in particular RsmY directly interact with human Ago2 during infection. We infected THP-1 cells with *L. pneumophila* and used hsa-Ago2 antibodies for Ago2-immunoprecipitation experiments followed by sequencing. To validate our approach, we first analyzed

whether hsa-miRNAs known to interact with Ago2 were among the sequences obtained. Indeed, we identified 72 known Ago2-interacting hsa-miRNAs (**Table S3**), but most excitingly we also identified RNAs derived from *L. pneumophila*, in particular we identified RsmY in two of our three pull downs. Given the fact that only about 50% of the THP-1 cells are infected by *L. pneumophila*, that only 30% of *Lp*-EVs contain RNA (**Figure S1C**), and that in our assays only about 3% of the *Lp*-EVs release their cargo in the tested condition (**Figure 3E**), the probability to identify *Lp*-derived RNAs in the bulk of human RNAs is very low. Thus, although the results are not statistically significant, our Ago2-CLIP indicated that RsmY seem to directly interact with Ago2 during infection. These results further support our model that RsmY and other *L. pneumophila* RNAs can act in a mi-RNA like manner in the host cell.

- Although the authors claim that RsmY and tRNA-Phe adopt microRNA-like functions, no data are provided regarding the incorporation of these two RNAs into the host microRNA machinery. In addition to Northern blot analysis of RsmY and tRNA-Phe processing fragments in the microRNA size range, Ago2-CLIP needs to be performed to validate the association of RsmY / tRNA-Phe or their derived fragments with the microRNA machinery.

It is not possible to do Northern blots, as the concentration of RsmY/tRNA-Phe in the host cell is too little that it could be visualized by Northern blot. It is already very rarely possible to see the protein effectors secreted in the host cytosol, due to the small amounts that are delivered. However, as requested, we have undertaken Ago2-CLIPseq and have shown that we can identify RsmY bound to Ago2. See answer question above and **Lines 432-446** in the text and **Lines 904-938** in M&M section.

- The abundance of RsmY in the host cell cytoplasm and nucleus during a physiological infection setting remains unclear. What are the RsmY copy numbers in cytosolic versus nuclear fractions after stimulation with a physiological MOI / vesicle amount? Does RsmY omit the nuclear steps of the microRNA processing machinery?

Initial processing of pri-miRNA in human cells occurs in the nucleus by the Drosha complex which crops the miRNA into a hairpin-shaped pre-miRNA. However, the processing of the pre-miRNA is typically located in the cytoplasm (DICER and RISC loading). Thus, the bacterial RNA might not be necessarily pri-pre-processed in the nucleus but is probably incorporated in the RISC complex directly in the cytoplasm. However, this analysis is out of the scope of the paper and would be an entirely new study.

- The authors show a co-localization of lipid dye with the lysosomes (Figure 2A/B). Are the lysosomes acidified? Do they degrade the RNA transported on or in the EVs? Can the bacterial RNA escape the lysosomes and can be found in the cytosol? Cellular fractionation experiments with subsequent isolation of RNA should be performed to prove the presence of sRNA in the host cell cytoplasm. Besides, the tracking by immunofluorescence should be prolonged and include a late timepoint.

The reviewer seems to have mixed the figures; it was Figure 3A/B where we showed co-localization of lipid dye labelled *Lp*-EVs with acidic vacuoles, lysotracker labels acidified structures thus lysosome and late endosomes and other acid subcellular structures. However, we have no data which percentage of the RNA might be eventually degraded, but we did not write nor assume that 100% of the RNA is successfully translocated into the host cell. We do not think that cell fractionation is a feasible method as there is too little quantity of bacterial RNA in the cytosol that it would be detected by this method given that only a part of the cells is infected and of these only maximal 30% of the EVs contain RNA. To confirm that the RNA is in the cytosol we have done as described above Ago2-CLIP, that has proofed that RNA and in particular RsmY is indeed present the host cytosol. See answer question above and **Lines 432-446** in the text and **Lines 908-929** in M&M section. Furthermore, we have developed an assay based on the split Luciferase and have used it the first time to analyse content release of bacterial EVs. Indeed, we show that the *Lp*-EVs release their content in the cytosol, but only a low amount (3%), comparable to what is observed also for release from EVs originating from mammalian cells (**Figure 3E**). To determine the importance of acidification for this process, we then used Bafilomycin A1 to inhibit endosomal acidification which showed that content release is dramatically decreased, pointing to the importance of an acidic environment for content release, in agreement with our immunofluorescence tracking results. These results are added in **Figure 3E** and in the text.

Lines 344-373 it reads... To examine whether *Lp*-EVs release their content in the host cell cytosol, we developed a content release assay based on a recent study that followed the delivery of a soluble EV-cargo (HSP70, human homolog of GroEL) within the cytosol of the acceptor cells⁴². This assay was upgraded by taking advantage of split-luciferase complementation system {Somiya, 2021 #4247. Briefly, an EV cargo was tagged with HiBiT (split luciferase 1/2) and isolated EVs were incubated on acceptor cell sexpressing LbBit (Split luciferase 2/2) within their cytosol. Luciferase complementation only occurs when EV deliver their cargo into the cytosol of acceptor cells. We had previously shown that the bacterial protein GroEL (*jpp0743*) is present in the *Lp*-EVs. Thus, we tagged GroEL with a HiBiT-tag and in parallel, we transfected THP-1 cells with a LgBiT construct (pCMV-Tag2-LgBiT) under the control of the CMV promoter to express the LgBiT protein in the host cell. If the EV-content is released, Luciferase complementation occurs, and this can be measured. To determine whether the *Lp*-EVs had released their content, we measured luciferase activity *in cellulo* with the *Nano-Glo(R) Live Cell Assay System (Promega)* after 3h of incubation with purified *Lp*-EVs containing GroEL-HiBiT. To estimate the total input of *Lp*-EVs containing GroEL-HiBiT, the luciferase activity was quantified after lysis of the *Lp*-EVs using the *Nano-Glo (R) HiBiT Lytic detection System (Promega)*. *Lp*-EVs not containing GroEL-HiBiT were used as negative control. As shown in **Figure 3E**, we could detect around 3% luciferase activity of the total GroEL-HiBiT input after 3h post-infection. This result also corresponds to our co-localization experiments, where after 3h pi around 5% of *Lp*-EVs co-localized with early or late endosomes, respectively (**Figure 3D**). When adding 10 μ M of cytochalasin D to the cells, the uptake and/or release was completely abolished

(**Figure 3D**), suggesting that actin-dependent processes play a crucial role in the endocytosis and/or membrane fusion events.

Strikingly, after the addition of 200nM Bafilomycin A1, which inhibits endosomal acidification, less than 1% of the total luciferase activity was detected after 3h pi meaning that less than a third of GroEL-HiBiT proteins were reacting with cytosolic LgBiT protein of the THP-1 cells compared to non BA1-treated samples. As it was shown recently that Bafilomycin A1 does not change the general uptake of EVs into the host cell ⁴², this is another hint that the acidification in late endosomes or lysosomes might be an important factor for *Lp*-EV content release.

As requested, we have prolonged the tracking by immunofluorescence and have done it in hMDMs to show that this happens also in other cells. These new data are now added as **Figure S3B** and added in the text.

Lines 323-326 it reads now... “Similarly, when using hMDM cells and pHrodo to label acidic structures and tracking the *Lp*-EVs within these cells for 17h the *Lp*-EVs clearly co-localize with acidic subcellular structures (**Supplementary Figure S3B**), but overall intensities and differences were less pronounced than in the U2OS cell lines.”

- The authors describe that they used different protocols for RNA preparation from bacteria and EVs for sequencing. EV-RNA was isolated using miRNeasy Mini Kit and RNA was DNase digested. They do not comment on the isolation protocol for bacterial (cellular) RNA, but this RNA was additionally rRNA depleted and fragmented. This additional steps can give a bias in sample preparation and does not support the comparability of sequenced RNA.

There seems to be a misunderstanding, perhaps we did not describe it clearly enough. Bacterial RNA and *Lp*-EV RNAs were purified with the same kit and DNase digest. The only difference was that the EVs were treated routinely with RNase to avoid contamination of extracellular RNA or RNA that stuck to the EVs outside and that the bacterial RNA was rRNA depleted as this needs to be done as one sequences only rRNA when this depletion step is not undertaken. However, to avoid wrong results because of this difference in the purification step, we did not take any rRNA that was identified within the EVs into account and eliminated them from our bioinformatics analyses. We have described our procedure now more in detail in the M&M and hope this is now clearer.

Lines 755-766 it reads now... *Lp*-EV pellets, purified from a 300ml liquid culture as described above and the corresponding bacterial pellets were resuspended in Qiazol and the RNA extraction was performed following the instruction of the miRNeasy®Mini kit (Qiagen). RNA samples were digested with Turbo DNase (Thermo Scientific) and the size distribution of the EV-RNA was evaluated with a Bioanalyzer (Agilent Technologies). The EV and bacterial RNA (but not *Lp*-EV-RNA) was rRNA depleted using the RiboZero rRNA Removal Kit for Gram-negative bacteria (Illumina) and metal-catalysed heat-fragmented to a size around 100-200nts using an RNA fragmentation kit (Ambion). The bacterial RNA was further processed according to the TruSeq stranded mRNA sample preparation guide of Illumina.

Before Illumina HiSeq multiplex sequencing, the quantity was determined with a Qubit 2.0 (Invitrogen) and the quality was checked by Bioanalyzer (Agilent Technologies). *Lp*-EV RNAseq analysis was done n=4 independent biological experiments.

(iii) Concerns regarding the exact interference with the RIG-I pathway:

- The RsmY-dependent effects on the host immune response are not convincingly presented. In addition to RIG-I, TBK1 and IRF3/7 Western blots an RNA-Seq analysis of gene expression changes in WT versus Δ RsmY Legionella infected / vesicle stimulated cells would be required to portrait the global impact of Legionella-encoded microRNA-like molecules on host gene expression.

As requested, we have analyzed by RNAseq the differences in the host immune response of cells infected with *Lp*-EVs purified from the wt strain and *Lp*-EVs purified from the Δ *rsmY* strain. At 3hpi we could see only small and not significant differences in the host response, with a special focus on ISGs and the RLR pathway genes. However, this is not surprising to us, as the host immune response is not regulated only by RsmY but many other factors play a role in infection (e.g. tRNA-Phe, LPS, ...) which all play together to get a robust change in the gene expression program that can be measured by RNAseq. One should also bear in mind, that not all cells are indeed infected with *Lp*-EVs (we estimate about 50% of the cells to be really infected) thus the background of the non-infected cells is high and may also mask more important transcriptional changes. We have added these results in the text and in the M&M section.

Lines 505-521 it reads now... “To further investigate the impact of RsmY on the host immune response, we performed RNAseq analyses comparing THP-1 cells incubated with *Lp*-EVs purified from wt bacteria or *Lp*-EVs purified from the RsmY mutant strain at 3h pi. Our results revealed that only slight but not significant differences in the host cell transcriptome were present. Indeed, the difference in the IFN- β levels we observed between THP-1 cells treated with *Lp*-EV purified from the wt strain and *Lp*-EVs purified from the Δ *rsmY* strain are apparently not enough to see significant changes in the transcription of Interferon stimulated genes (ISG) at transcript level. However, this is not surprising as accumulating evidence indicates that ISG expression is not solely dependent on IFN- β , but that ISGs can also be up-regulated directly after a pathogen infection independent of IFN- β signalling, thus the network underlying the regulation of the ISG is much more complex⁵⁹⁻⁶¹. Furthermore, RsmY and tRNA-Phe seem to act in concert on the host immune response and a knockout of both, which is unfortunately not possible to achieve, might lead to more important effects on the transcriptional level. Overall, the differences on transcript level between wt *Lp*-EV and Δ *rsmY*-EV treated cells are small and not significant including the transcripts of *ddx58* (Rig-I) and *irak1* suggesting also that post-transcriptional effects may play a more dominant role in the regulation of protein expression by *Lp*-EVs.”

In addition, interferon ELISAs are required to document the postulated effect of RsmY on RIG-I-induced host responses.

There seem to be a misunderstanding, we have undertaken interferon ELISAs to measure the effect of RsmY and tRNA-Phe, after incubation with *Lp*-EVs and after RNA transfection. These results are represented in **Figure 4F and 4G**. Moreover, we measured the extracellular IFN- β levels by ELISA also during bacterial infection (Suppl Fig S5A)

As was requested by reviewer 2 we also did experiments with TLR agonists to see the influence on the *Lp*-EVs on their stimulation. In these experiments IFN- β is also measured by ELISA (**Figure S6**)

- The authors do not provide proof for RIG-I-dependence of the effects of Legionella RsmY deletion on host immune signalling. Experiments with RIG-I / MAVS deficient cells are required to exclude the possibility that the presented effects depend on additional pattern recognition receptor pathways.

As requested by the reviewer we tried to obtain RIG-I / MAVS deficient THP-1 cells but did not succeed during this revision period, which was in addition very impacted by a Covid lockdown and work time restrictions. Thus, we could not conduct this experiment. Furthermore, due to the fact that RsmY and tRNA-Phe (and maybe also other potential EV-RNAs) have the same impact on RIG-I an infection of RIG-I knockout cells with an RsmY mutant alone might thus not be sufficient to obtain significant results. However, to answer the request of the reviewer, we analysed the intracellular replication of *L. pneumophila* after knockdown of *ddx58* (Rig-I) and IRAK1 by siRNA. The result again strengthens our observation that RIG-I is important during *L. pneumophila* infection (**new Fig 4H**). Additionally, we performed infection experiments comparing RsmY and as-RsmY RNA transfection (**new Suppl Fig S5D**) showing that *L. pneumophila* grows better when RsmY is present.

Lines 538-549 it reads now... To further analyze the mechanism, we specifically down-regulated the expression of *ddx58* and *irak1* by siRNA-mediated gene silencing. Protein levels of Rig-I or IRAK1, respectively were reduced by 60-80% at the time point of infection compared to scramble transfected control cells (**Figure 4H**). After 24 hpi, no significant differences in the replication of *L. pneumophila* were detected, but at later time points (48 and 72 hpi), the number of bacteria in cells where DDX58 (Rig-I) was downregulated by siRNA was increased by up to 50% further confirming that suppression of Rig-I is beneficial for intracellular replication of *L. pneumophila*. Additionally, we transfected THP-1 cells with RsmY RNA or its anti-sense sequence (as-RsmY) and infected these cells. *L. pneumophila* replicated significantly better in the cells transfected with RsmY-RNA, similar to what was observed after siRNA knockdown of *ddx58*, again showing that RsmY has a beneficial effect on *L. pneumophila* replication **Supplementary Figure S5D**)

Finally, we pre-treated THP-1 cells with *Lp*-EVs purified from the wt or Δ *rsmY* strains (**new Suppl Fig S5C**). This experiment highlights the effect of *Lp*-EVs and in particular of those containing RsmY-RNA on *L. pneumophila* survival and propagation.

Lines 530-538 it reads now...” To investigate the influence that IFN- β secretion, partly induced by RsmY, has on infection, we treated THP-1 cells with different concentrations of IFN- β and analyzed the replication phenotype of *L. pneumophila*. We show that increasing concentrations of extracellular IFN- β reduce intracellular replication of *L. pneumophila* in THP-1 cells, whereas high concentrations of IL-1 β have no impact (**Supplementary Figure S5B**). We then pre-treated the THP-1 cells with *Lp*-EVs either purified from wt bacteria or with *Lp*-EVs purified from the Δ *rsmY* strain. In cells pre-treated with wt *Lp*-EVs we observed a significantly higher replication of *L. pneumophila* than in cells incubated with *Lp*-EVs purified from the Δ *rsmY* strain (**Supplementary Figure S5C**)

- The authors state that RsmY in EVs is regulating cRel, but they do not show a regulation of cREL on protein level by *Legionella* infection of EV treatment of the macrophages.

Indeed, the decrease at protein level was not significant, but as the more sensitive luciferase assay detected a difference, we added it to our results, but we focused mainly on RIG-I and IRAK1 as these two proteins showed differences in all our assays conducted. However, this result is further showing that several factors and several RNAs act on the same pathway.

- The authors used the bone osteosarcoma epithelial cell line U2OS cells for parts of their experiments. They should reproduce the experiments in a more physiological cell culture model for *Legionella pneumophila* infection as osteosarcoma cells might respond to differently to a bacterial infection and the stimulation of PRRs with PAMPs present on or in *Legionella* EVs. In addition, THP-1 cells were used. These cells are monocytes, unless they are differentiated with phorbol 12-myristate 13-acetate in macrophages. Did the authors use monocyte- or macrophages-like cells? And if so, how have they been differentiated?

As requested by the reviewer we have conducted several assays now also in primary macrophages. We have analyzed the impact of *Lp*-EVs on RIG-I and IRAK 1 protein levels, the dual luciferase assay and the pRhodo assay also in hMDMs purified from human blood donors. The results are confirming our results obtained with THP-1 cells. These results are added in the text and as figures.

Lines 218-220 it reads now... This phenotype was also observed when incubating human monocyte derived macrophages (hMDM) with *Lp*-EVs purified from wt *L. pneumophila* or the Δ *rsmY* strain (**Supplementary Figure 2A**).

Line 402-405 it reads now...” Additionally, we undertook the dual luciferase reporter gene assay described above also in primary cells, to rule out that this result is due to the cell line used. Indeed, when repeating the above-described experiment in CD14+ cells isolate from human blood we obtained the same result as in THP-1 cells (**Supplementary Figure S4B**).”

Lines 322-326 it reads now... “Similarly, when using hMDM cells and pHrodo to label acidic structures and tracking the *Lp*-EVs within these cells for 17h the *Lp*-EVs clearly co-localize with acidic vacuoles (**Supplementary Figure 3B**), but overall intensities and differences were less pronounced than in the U2OS cell lines.”

Reviewer #2 (Remarks to the Author):

In this work Sahr et al have identified two *Legionella* sRNA species which are proposed to mimic host miRs to control innate immune responses. Although the original observation is interesting, the work appears rather under-developed, and there are limited data on the physiological implication of the observation, including in primary cells.

We thank this reviewer for the comments and have taken them into account and have undertaken additional experiments as requested. However, the main concern of this reviewer seems that the impact of RsmY on INF secretion and the host immune response is small. The experiments undertaken here, further confirmed that RsmY has as significant but small impact, however to us this is not surprising but expected. In addition to the many traditional, bacterial virulence factors like LPS, Flagella, pili, type II secretion system or outer membrane proteins *Legionella* secretes over 330 effectors by its type 4 secretion system Dot/Icm. These many factors and in particular the T4SS effectors work together to subvert the host immune response and are often redundant. This led different groups even to decipher how to understand this redundancy (e.g. O'Connor TJ, Boyd D, Dorer MS, **Isberg RR**. “Aggravating genetic interactions allow a solution to redundancy in a bacterial pathogen.” *Science*. 2012 Dec 14;338(6113):1440). Thus, if one single virulence factor is deleted, it is rare that one can observe strong phenotypes in *Legionella* growth, mostly there is not even a phenotype because of redundancy. Thus, it is not surprising that RsmY has only small impact but as it is significant and reproducible in many different systems, it is one piece in the puzzle of the manifold ways how *Legionella* can subvert the hosts immune response.

1. The manuscript critically lacks data on the impact of the identified mechanism on host innate immune responses and anti-microbial defense. The data presented in Fig 4D-E are based on purified EVs and in THP1 cells, and show only a modest effect. As a minimum, the authors should compare IFN β (and ISG) responses to infection with wt and KO bacteria in THP1 cells and primary macrophages.

There might be an oversight, we have quantified the IFN- β response in wt and RsmY mutant strains in THP-1 cells. This result was and is depicted in **Figure S5A** (former **figure S4E**).

Lines 487-500 it reads... . We quantified the amount of extracellular IFN- β by performing an ELISA with the supernatants at different time points of the infection. However, no significant differences were found when IFN- β concentrations of cells

infected wt *L. pneumophila* were compared to cells infected with the $\Delta rsmY$ -mutant strain (**Supplementary Figure S5A**), suggesting that this approach does not reveal the influence of RsmY on IFN- β secretion as additional factors may also influence IFN- β levels as known for several bacterial and viral infections⁵⁸, and these combined effects are measured. Thus, to measure specifically the impact of *Lp*-EVs containing RsmY, we analysed the extracellular concentration of IFN- β in the supernatant of THP-1 cells after incubation with *Lp*-EVs purified either from wt *L. pneumophila* or from the $\Delta rsmY$ mutant strain. Indeed, as shown in **Figure 4F**, internalization of *Lp*-EV containing RsmY (purified from the wt strain) induces less IFN- β secretion by the host cells, than those infected with *Lp*-EVs from which RsmY is absent (purified from the $\Delta rsmY$ strain). These results suggest that *Lp*-EVs containing RsmY dampen IFN- β secretion of infected human cells.”

To analyze also primary cells, we have analyzed the impact of *Lp*-EVs on RIG-I and IRAK 1 protein levels, the dual luciferase assay and the pHrhodo assay also in primary cells (hMDMs purified from human blood donors). The results are confirming our results obtained with THP-1 cells. These results are added in the text and as figures.

Lines 218-220 it reads now... This phenotype was also observed when incubating human monocyte derived macrophages (hMDM) with *Lp*-EVs purified from wt *L. pneumophila* or the $\Delta rsmY$ strain (**Supplementary Figure 2A**).

Line 402-405 it reads now...” Additionally, we undertook the dual luciferase reporter gene assay described above also in primary cells, to rule out that this result is due to the cell line used. Indeed, when repeating the above-described experiment in CD14+ cells isolate from human blood we obtained the same result as in THP-1 cells (**Supplementary Figure S4B**).”

Lines 322-326 it reads now... “Similarly, when using hMDM cells and pHrodo to label acidic structures and tracking the *Lp*-EVs within these cells for 17h the *Lp*-EVs clearly co-localize with acidic vacuoles (**Supplementary Figure 3B**), but overall intensities and differences were less pronounced than in the U2OS cell lines.”

2. Along the same lines, the authors should more globally characterize how the sRNAs affect host cell gene expression through RNAseq analysis.

As requested, we have analyzed by RNAseq the differences in the host immune response of cells infected with *Lp*-EVs purified from the wt strain and *Lp*-EVs purified from the $\Delta rsmY$ strain. At 3hpi we could see only small and not significant differences in the host response, with a special focus on ISGs and the RLR pathway genes. However, this is not surprising to us, as the host immune response is not regulated only by RsmY but many other factors play a role in infection (e.g. tRNA-Phe, LPS, ...) which all paly together to get a robust change in the gene expression program that can be measured by RNAseq. One should also bear in mind, that not all cells are indeed infected with *Lp*-EVs as we estimate about 50% of the cells to be really infected and only a smaller amount of these vesicles enter the endosomal pathway

and release the cargo (10% and 3%, respectively), comparable to what is also observed for exosomes. Thus, the background of the non-infected cells is high and may also mask more important transcriptional changes. We have added these results in the text and in the M&M section.

Lines 505-521 it reads now... “To further investigate the impact of RsmY on the host immune response, we performed RNAseq analyses comparing THP-1 cells incubated with *Lp*-EVs purified from wt bacteria or *Lp*-EVs purified from the RsmY mutant strain at 3h pi. Our results revealed that only slight but not significant differences in the host cell transcriptome were present. Indeed, the difference in the IFN- β levels we observed between THP-1 cells treated with *Lp*-EV purified from the wt strain and *Lp*-EVs purified from the Δ *rsmY* strain are apparently not enough to see significant changes in the transcription of Interferon stimulated genes (ISG) at transcript level. However, this is not surprising as accumulating evidence indicates that ISG expression is not solely dependent on IFN- β , but that ISGs can also be up-regulated directly after a pathogen infection independent of IFN- β signalling, thus the network underlying the regulation of the ISG is much more complex⁵⁹⁻⁶¹. Furthermore, RsmY and tRNA-Phe seem to act in concert on the host immune response and a knockout of both, which is unfortunately not possible to achieve, might lead to more important effects on the transcriptional level. Overall, the differences on transcript level between wt *Lp*-EV and Δ *rsmY*-EV treated cells are small and not significant including the transcripts of *ddx58* (Rig-I) and *irak1* suggesting also that post-transcriptional effects may play a more dominant role in the regulation of protein expression by *Lp*-EVs.”

3. To make sure that the observed effects of bacterial EVs are in fact due to targeting of RIG-I, the authors should generate RIG-I KO THP1 cells and demonstrate that the modulatory effect of the EVs is lost.

As requested by the reviewer we tried to obtain RIG-I / MAVS deficient THP-1 cells but did not succeed during this revision period, which was in addition very impacted by a Covid lockdown and work time restrictions. Thus, we could not conduct this experiment. Furthermore, due to the fact that RsmY and tRNA-Phe (and maybe also other potential EV-RNAs) have the same impact on RIG-I an infection of RIG-I knockout cells with an RsmY mutant alone might thus not be sufficient to obtain significant results. However, to answer the request of the reviewer, we analysed the intracellular replication of *L. pneumophila* after knockdown of *ddx58* (Rig-I) and IRAK1 by siRNA. The result again strengthens our observation that RIG-I is important during *L. pneumophila* infection (**new Fig 4H**). Additionally, we performed infection experiments comparing RsmY and as-RsmY RNA transfection (**new Suppl Fig S5D**) showing that *L. pneumophila* grows better when RsmY is present.

Lines 538-549 it reads now...” To further analyze the mechanism, we specifically down-regulated the expression of *ddx58* and *irak1* by siRNA-mediated gene silencing. Protein levels of Rig-I or IRAK1, respectively were reduced by 60-80% at the time point of infection compared to scramble transfected control cells (**Figure 4H**). After 24 hpi, no significant differences in the replication of *L. pneumophila* were

detected, but at later time points (48 and 72 hpi), the number of bacteria in cells where DDX58 (Rig-I) was downregulated by siRNA was increased by up to 50% further confirming that suppression of Rig-I is beneficial for intracellular replication of *L. pneumophila*. Additionally, we transfected THP-1 cells with RsmY RNA or its anti-sense sequence (as-RsmY) and infected these cells. *L. pneumophila* replicated significantly better in the cells transfected with RsmY-RNA, similar to what was observed after siRNA knockdown of *ddx58*, again showing that RsmY has a beneficial effect on *L. pneumophila* replication **Supplementary Figure S5D**)

Finally, we pre-treated THP-1 cells with *Lp*-EVs purified from the wt or Δ *rsmY* strains (**new Suppl Fig S5C**). This experiment highlights the effect of *Lp*-EVs and in particular of those containing RsmY-RNA on *L. pneumophila* survival and propagation.

Lines 530-538 it reads now... To investigate the influence that IFN- β secretion, partly induced by RsmY, has on infection, we treated THP-1 cells with different concentrations of IFN- β and analyzed the replication phenotype of *L. pneumophila*. We show that increasing concentrations of extracellular IFN- β reduce intracellular replication of *L. pneumophila* in THP-1 cells, whereas high concentrations of IL-1 β have no impact (**Supplementary Figure S5B**). We then pre-treated the THP-1 cells with *Lp*-EVs either purified from wt bacteria or with *Lp*-EVs purified from the Δ *rsmY* strain. In cells pre-treated with wt *Lp*-EVs we observed a significantly higher replication of *L. pneumophila* than in cells incubated with *Lp*-EVs purified from the Δ *rsmY* strain (**Supplementary Figure S5C**)

4. The work would gain significantly, if induction of type I IFN by a panel of synthetic agonists for TLRs and cytosolic PRRs were evaluated in cells treated with relevante EVs.

As requested by the reviewer we have undertaken experiments using TLR agonists. We have pretreated THP-1 cells for 3h with or without *Lp*-EVs purified either from *L. pneumophila* wt or from the Δ *rsmY* strain. Subsequently, TLR-related agonists were added and the extracellular IFN- β concentrations were measured by ELISA 20h post incubation showing that particularly TLR signalling pathways depending on IRAK1 like TLR1, TLR2, TLR6 or on RLR (Poly(I:C), ssRNA40) are significantly down-regulated after *Lp*-EV-treatment, whereas pathways that can also be activated via alternative signalling routes e.g. through TRIF/TRAM (TLR3, TLR4), or cGAS-STING (ODN2006) are less affected or can even be stimulated by *Lp*-EVs. *Lp*-EVs purified from the Δ *rsmY* strain, thus lacking RsmY inhibit IFN- β -secretion significantly less than *Lp*-EVs purified from *L. pneumophila* wt strain. Additionally, Ago2-Inhibition significantly increases the IFN-beta secretion of *Lp*-EV-treated THP-1 cells These new results are added as Supplementary figure 6 and in the text.

Lines 558-570 it reads ... Finally, to further characterize the impact of *Lp*-EVs on the host immune response, we incubated THP-1 cells with agonists that mimic pathogen-associated molecular patterns (PAMPs) and measured extracellular IFN- β concentrations after pre-treatment with *Lp*-EVs that were either purified from wt

bacteria or from the $\Delta rsmY$ strain. Indeed, the IFN- β response of certain TLR agonists was dampened after pre-incubation with wt *Lp*-EVs but less with $\Delta rsmY$ -EVs further pointing to the influence of RsmY on the host immune response. In particular the IFN- β response triggered by agonists for TLR1/2/5/6 and TLR8 was significantly reduced when pre-treating the cells with *Lp*-EVs (**Supplementary Figure S6**). The IFN- β response to TLR9 agonist CpG instead was even more pronounced after pre-incubation of the THP-1 cells with *Lp*-EVs compared to control experiments, probably due to synergetic effects of multiple ligand stimulations. In contrast, agonist stimulation of TLR3, TLR4 or TLR7 was not affected by *Lp*-EV-treatment, whereas inhibition of Ago2 slightly induced the extracellular IFN- β levels after *Lp*-EV-treatment (**Supplementary Figure S6**).”

5. The functional data do generally not show a very large effect of the EVs/sRNAs (e.g. in Fig 4). Therefore, for the work to have impact, it is essential that the authors show data on the effect of the proposed immunomodulatory RNAs in bacterial growth.

We have tested the growth of a $\Delta rsmY$ strain in growth in THP-1 cells compared to the wt strain. As seen from the graph below, the difference is very small and there is no real growth defect of the $\Delta rsmY$ strain compared to the wt strain. However, this is not surprising but a result which is well known in the *Legionella* field. Given the over 300 effectors *L. pneumophila* is secreting in addition and all work in concert and are often redundant to manipulate the host response, a big growth defect is rarely observed when knocking them out. Even among the over about 30 effectors analysed to date there are only two or three that have an important impact on growth in our classical growth assays. In addition, the growth assays are not very sensitive and thus we did not expect a big difference in growth when RsmY is missing. We did not add the figure to the manuscript but can do this of course if the reviewer thinks it is important to show.

However, we observed a clear and significant effect on the intracellular growth of *L. pneumophila* when we pre-treated the THP-1 cells with *Lp*-EVs deriving from wt or the $\Delta rsmY$ mutant strain, indicating that a high number of purified vesicles indeed leads to a positive effect on intracellular replication of *L. pneumophila* depending on RsmY (**new suppl Fig S5C**).

6. The functional data in Fig 4, should be confirmed in primary cells, and ideally also in mice (if the sRNAs also target RIG-I and IRAK1 in mice).

As requested, we have confirmed the data of Figure four in primary cells by redoing the dual luciferase assay in CD4+ cells purified from human blood donors. This result is added in figure S4B and in the text.

Line 402-409 it reads now...” Additionally, we undertook the dual luciferase reporter gene assay described above also in primary cells, to rule out the possibility that the result is due to the cell line used. Indeed, when repeating the above-described experiment in CD14+ cells isolated from human blood we obtained the same result as in THP-1 cells (**Supplementary Figure S4B**). These results are also in agreement with the results obtained after RNA transfection (**Figure 1I and Supplementary Figure S2B**), further supporting our results that RsmY interacts with the UTR of the RIG-1, and tRNA-Phe with the UTR of the *irak1* encoding gene and indeed can behave like eukaryotic micro-RNAs.

As to the mouse experiments we do not think that this is necessary nor ethically to be defended. Firstly, Legionella’s natural host are amoeba and phagocytic cells, and this is where we have done all our experiments. Secondly mice are not good hosts for Legionella, except AJ mice that have a mutation in Naip5, and in addition mouse macrophages show a very different immune response to *L. pneumophila* infection than what is observed for human cells. Other mouse strains cannot be infected with wt *L. pneumophila*. RIG-I and IRAK1 knock out mice are only available in BALBc mice, and BALBc mice cannot be infected with wt *L. pneumophila* except when constructing in addition a *flaA* mutant to knock out flagellin that is recognized by the mouse immune system.

REVIEWER COMMENTS

Reviewer #1 (Remarks to the Author):

The manuscript by Sahr et al. has benefited significantly from the additional data the authors provided during the revision rounds. However, based on the data provided, the central claim that small RNAs from *Legionella* mimic microRNAs is still not sufficiently supported by experimental data to rule out alternative mRNA targeting mechanisms. The sole inhibition of Ago2, as previously explained in the comments to the authors, is not sufficient to prove a microRNA-like mechanism of action. Therefore, the referee recommends revising the title and other relevant parts of the manuscript to avoid overinterpretation of the results with regard to the claimed microRNA-like mechanism.

Moreover, the referee feels that calling the incubation of cells with vesicles and their potential uptake an “infection” is misleading to potential readers because infections with live bacteria are also performed in the same manuscript. This linguistic clarity would make the article considerably more readable.

Thirdly, Sahr et al. decided to label the stimulation dose “MOI 10” although they discovered during the revision process that the actual vesicle dose was higher by a factor of 1,000. Only changes to the text would be needed to make the information about the actual vesicle dose of 10,000 EVs/cell directly available to any reader to interpret the physiological relevance of the described phenomenon.

The referee wishes to emphasize that regardless of the exact mechanism the data presented is of general interest for the infection biology community. A remaining limitation of the study by Sahr et al. is that the physiological relevance of host mRNA targeting by *Legionella* sRNAs remains unclear.

Reviewer #2 (Remarks to the Author):

This reviewer finds that the authors have provided sufficient data to support the conclusion that *Legionella pneumophila* deliver sRNAs to host cells via OMVs to regulate RIG-I signaling.

Reviewer #1 (Remarks to the Author):

The manuscript by Sahr et al. has benefited significantly from the additional data the authors provided during the revision rounds. However, based on the data provided, the central claim that small RNAs from *Legionella* mimic microRNAs is still not sufficiently supported by experimental data to rule out alternative mRNA targeting mechanisms. The sole inhibition of Ago2, as previously explained in the comments to the authors, is not sufficient to prove a microRNA-like mechanism of action. Therefore, the referee recommends revising the title and other relevant parts of the manuscript to avoid overinterpretation of the results with regard to the claimed microRNA-like mechanism.

We are happy that the reviewer considers that our manuscript has improved significantly. However, we are confused with the statement “the sole inhibition of Ago2 is not sufficient to prove a microRNA-like mechanism of action”. The effect of Ago2 inhibition is not the one and only proof for bacterial miRNA like action that we have provided in this paper but is one of many different experiments that support our conclusion. For example, we have in addition shown the interaction of RsmY and tRNA-Phe with the 3'UTR of the targeted human mRNAs using EMSA (Fig1J), we have undertaken a dual luciferase assays (Fig4A-D), an assay that is typically used for showing miRNA interaction for human cells, etc. Furthermore, we never state in our manuscript that RsmY acts like a miRNA but we clearly state that it acts in a miRNA-like manner.

Given the many different experiments we have undertaken that all support our conclusion and in accordance with Reviewer 2 who states “This reviewer finds that the authors have provided sufficient data to support the conclusion that *Legionella pneumophila* deliver sRNAs to host cells via OMVs to regulate RIG-I signaling” we prefer to leave the miRNA-like mechanism in our manuscript and the title.

Moreover, the referee feels that calling the incubation of cells with vesicles and their potential uptake an “infection” is misleading to potential readers because infections with live bacteria are also performed in the same manuscript. This linguistic clarity would make the article considerably more readable.

According to the reviewers request we have scanned the entire manuscript and indeed sometimes we had written *Lp*-EV infection. We apologize for this mistake. To avoid confusion, we have replaced infection by “incubation with *Lp*-EVs” or “pretreatment with *Lp*-EVs “or uptake of *Lp*-EVs” and have used the term infection only for bacterial infection. All instances where we write about *Lp*-EV incubation are highlighted in yellow throughout the manuscript to indicate that infection was not used or that it was changed.

Thirdly, Sahr et al. decided to label the stimulation dose “MOI 10” although they discovered during the revision process that the actual vesicle dose was higher by a factor of 1,000. Only changes to the text would be needed to make the information about the actual vesicle dose of 10,000 EVs/cell directly available to any reader to interpret the physiological relevance of the described phenomenon.

The reviewer might have overseen that we have already added this information into the text of the manuscript at the second revision step. Thus, this information is readily available for the reader as requested.

Lines 700-714 it reads ... Human cells were incubated with *Lp*-EVs at an MOI of 10 (according to flow cytometry dye-labelled events). The values we obtained by flow cytometry are much lower than those obtained with the ZetaView analysis, in which we measure a concentration of *Lp*-EVs about 1 000 times higher as compared to the flow cytometry data. As we adjusted our MOI in all experiments to the conventional flow cytometry data, we indicate throughout the manuscript that the MOI was estimated according to conventional flow cytometry. It also needs to be noted, that we analysed how many cells indeed take up of *Lp*-EVs to estimate the amounts of *Lp*-EVs impacting the host cell. We used high content image analyses to analyse the number of EVs detectable in infected U2OS cells. We observed on average only about 5-10% of cells that contained detectable EVs. Thus, even with the relative high amount of EVs used according to ZetaView quantification the cell cultures were not saturated with *Lp*-EVs but a high background of noninfected cells that moderate the measurable output is present. Also, the amount of EVs we used did not lead to cell death of U2OS or hMDM cells, even after 17h pi (**Supplementary Figure S3C**).

The referee wishes to emphasize that regardless of the exact mechanism the data presented is of general interest for the infection biology community. A remaining limitation of the study by Sahr et al. is that the physiological relevance of host mRNA targeting by Legionella sRNAs remains unclear.

We thank the reviewer for this comment. However, we are convinced that there is a physiological relevance in our finding. This is further underlined with the estimation of cells that have taken up *Lp*-EVs which are only about 5-10% of all cells analyzed. Given the background of the uninfected cells, the impact the EVs have is important.

Reviewer #2 (Remarks to the Author):

This reviewer finds that the authors have provided sufficient data to support the conclusion that Legionella pneumophila deliver sRNAs to host cells via OMVs to regulate RIG-I signaling.

Thank you for this encouraging and positive evaluation